# ConquerNet: Convolution-Smoothed Quantile ReLU Neural Networks with Minimax Guarantees

## Abstract

Quantile regression is a fundamental tool for distributional learning but poses significant optimization challenges for deep models due to the non-smoothness of the pinball loss. We propose ConquerNet, a class of **con**volution-smoothed **qu**antile **R**eLU neural **net**works, which yield smooth objectives while preserving the underlying quantile structure. We establish general nonasymptotic risk bounds for ConquerNet under mild conditions, providing minimax guarantees over Besov function classes. In numerical studies, the proposed approach improves performance across a broad range of settings, with gains becoming more consistent as sample size grows, especially at the low and high quantiles.

## 1 Introduction

Quantile regression is a widely considered statistical tool for modeling heterogeneous effects and capturing the distributional structure of responses beyond the conditional mean. In many fields such as quantitative finance, survival analysis, and econometrics (Baur & Dimpfl, 2019; Horowitz, 1998; Chernozhukov & Hansen, 2005), quantile regression is used to understand the tail behaviors and provide robust outcomes faced with skewed or heavy-tailed data. Formally, given quantile level $\tau \in (0, 1)$ and i.i.d. samples $\{(\mathbf{x}_i, y_i)\}_{i=1}^n$ from the random vector $(X, Y)$ where $X \in \mathbb{R}^d, Y \in \mathbb{R}$, the conditional $\tau$-quantile function is defined as

$$f_\tau^*(\mathbf{x}_i) = F_{y_i|\mathbf{x}_i}^{-1}(\tau) = \inf\{y : \mathrm{P}(y_i \le y|\mathbf{x}_i) \ge \tau\}. \tag{1}$$

The standard approach estimates the conditional quantile function $f_\tau^*$ by minimizing the empirical quantile loss

$$\hat{f}_\tau = \arg\min_f \sum_{i=1}^n \rho_\tau(y_i - f(\mathbf{x}_i)), \tag{2}$$

where $\rho_\tau(u) = \max\{\tau u, (\tau - 1)u\}$. We refer to Koenker et al. (2017) for an overview of quantile regression models. Although the estimator in (2) is theoretically well-founded, its loss function is non-differentiable. In models with a large parameter space, optimization becomes challenging so that efficient training is difficult. The neural network is a class of such models: even with a fixed input dimension $d$, the number of parameters can be very large. The flexibility of neural networks enables approximation of complex functions and, in theory, achieving minimax-optimal performance over Besov spaces (Suzuki, 2019). The success of neural networks depends heavily on generalization. Gradient-based optimizers such as stochastic gradient descent (SGD) often generalize well (Wu et al., 2022; Dziugaite & Roy, 2017), but sharp minima (Hochreiter & Schmidhuber, 1997) and non-smooth loss surfaces (Huang et al., 2020; Foret et al., 2021) can degrade performance, especially with large-batch training (Keskar et al., 2016). This is because the high sensitivity to parameter changes caused by the sharp minima reduces the stability of optimization and model performance, and thus weakens generalization. Unfortunately, the quantile loss $\rho_\tau(u)$ is non-differentiable at $u = 0$ forming a sharp minima, and its gradient jumps abruptly from $\tau$ to $\tau - 1$ nearby.

To address the difficulties brought by the sharp minima and non-smooth loss functions, smoothing techniques have been developed, as seen in Berrada et al. (2018), which utilizes a smoothed loss function for classification tasks. For the quantile loss function, the convolution-type smoothed quantile loss (Fernandes et al., 2021) is

particularly attractive, as it preserves both convexity and differentiability while retaining the robustness of quantile regression. Existing applications of convolution-type smoothing have been largely confined to linear models, and it remains unclear whether its advantages carry over to more flexible function classes. In this paper, we propose ConquerNet, a novel framework that integrates convolution-type smoothing with neural networks. We demonstrate that this architecture mitigates optimization difficulties, preserves statistical guarantees, and leverages the expressive capacity of deep learning models.

## 1.1 Related work

Our work builds upon smoothed quantile regression, neural networks, and minimax analysis in Besov spaces. For smoothed quantile regression, Horowitz (1998) introduced a kernel-based smoothing approach that alleviates non-differentiability but sacrifices convexity, leading to challenging optimization problems. More recently, Fernandes et al. (2021) proposed the convolution smoothing method, which preserves both convexity and differentiability while gaining smoothness. The convolution-type smoothing framework has since been widely applied in quantile regression, and subsequent works have established its strong statistical guarantees, including minimax optimality and asymptotic as well as nonasymptotic properties; see Kaplan & Sun (2017), Tan et al. (2022), and He et al. (2023). Apart from parametric models, Hu et al. (2025) proposed a local linear convolution-type smoothing estimator for time-varying coefficient models. However, the convolution-type smoothing estimator has primarily been developed in linear models, with limited exploration in nonlinear nonparametric settings. To the best of our knowledge, nonlinear extensions of the convolution-type smoothing estimator remain underdeveloped, particularly in the context of modern neural networks. This motivates our focus, since Suzuki (2019) showed that deep neural networks can achieve minimax rate in Besov spaces where classical linear estimators such as kernel ridge regression, Nadaraya–Watson, or sieve methods are suboptimal.

Alongside these developments, a growing body of work has applied neural networks to quantile regression. Quantile networks have found applications in credit portfolio analysis, transportation, and survival analysis; see Feng et al. (2010), Rodrigues & Pereira (2020), and Pearce et al. (2022). On the theoretical side, statistical guarantees for quantile networks have been investigated, such as error bounds and minimax optimality; see Padilla et al. (2022), Shen et al. (2024), and Shen et al. (2025). Especially for minimax rates, Padilla et al. (2022) established that ReLU networks achieve near-minimax rates for quantile regression in Besov spaces. More recent work has also considered settings with covariate shift and noncrossing constraints (Feng et al., 2024; Shen et al., 2025), but these results remain restricted to Hölder classes. Together, this literature suggests that bridging smoothed quantile regression with the expressive power of neural networks is a promising and unexplored direction.

## 1.2 Contributions

In this paper, we demonstrate that the convolution-type smoothing quantile regression technique can be effectively integrated with neural networks to advance quantile regression. Our study contributes to distributional learning through smoothed quantile objectives, which naturally relate to quantile-based modeling and theoretical guarantees—areas of growing interest within the deep learning community (Suzuki, 2019; Sun et al., 2022; Nishimura & Suzuki, 2024; Kelen et al., 2025). The main contributions are as follows:

(i) On the theoretical side, we first prove that the proposed ConquerNet estimator attains the minimax convergence rate over Besov spaces, up to a logarithmic factor. In addition, we establish general nonasymptotic error bounds that apply without assuming specific smoothness conditions on the target function. These results demonstrate that the combination of convolution-type smoothing and neural networks retains the desirable statistical guarantees of nonparametric quantile estimation while leveraging the expressive capacity of deep learning models.

(ii) On the methodological and empirical side, we propose the ConquerNet architecture, which extends the convolution-type smoothing framework from linear nonparametric models to deep neural networks, thereby enabling its application in highly nonlinear nonparametric settings. Simulation studies are conducted to assess the performance of the proposed method, and the results show improvements over existing quantile networks in terms of both estimation accuracy and computational efficiency.

Together with the theoretical findings, these empirical results highlight the effectiveness of applying convolution-type smoothing to modern neural network architectures.

The rest of the article is organized as follows. We introduce the convolution-type smoothing framework with ReLU networks in Section 2. In Section 3, we present the minimax rate for our estimation in Besov spaces and further develop general upper bounds. Simulation studies and real data analysis are conducted in Section 4. Finally, Section 5 concludes with a discussion of potential extensions and limitations for future research directions. All proofs and additional simulation results are given in the Appendix.

## 2 Preliminaries

Before starting our main result, we specify some notations regarding the ReLU neural networks with layer parameter $L$, node parameter $W$, sparsity constraint $S$, and norm constraint $B$. In specific, we define

$$
\begin{aligned}
\mathcal{I}(L, W, S, B) := \Big\{ &\Big( A^{(L)}\sigma(\cdot) + b^{(L)} \Big) \circ \cdots \circ \Big( A^{(1)}x + b^{(1)} \Big) : A^{(1)} \in \mathbb{R}^{W \times d}, b^{(1)} \in \mathbb{R}^d, \\
&A^{(L)} \in \mathbb{R}^{1 \times W}, b^{(L)} \in \mathbb{R}, A^{(l)} \in \mathbb{R}^{W \times W}, b^{(l)} \in \mathbb{R}^W (1 < l < L), \\
&\sum_{l=1}^{L} \left( \left\| A^{(l)} \right\|_0 + \left\| b^{(l)} \right\|_0 \right) \le S, \max_{l} \left( \left\| A^{(l)} \right\|_\infty \bigvee \left\| b^{(l)} \right\|_\infty \right) \le B \Big\},
\end{aligned}
\tag{3}
$$

as the class of sparse networks with ReLU activation $\sigma(x) = \max\{x, 0\}$, where $\circ$ denotes the composition of functions, $\|A\|_0$ denotes the number of non-zero elements of the matrix $A$, and $\|A\|_\infty$ denotes the maximum of the absolute values of the elements in matrix $A$. In our paper, we also define $\infty$-norm for function $f(\cdot)$ on the compact domain $\mathcal{X}$, $\|f\|_\infty = \sup_{\mathbf{x} \in \mathcal{X}} |f(\mathbf{x})|$. The sparse network (3) has been widely investigated, by for exmaple Suzuki (2019), Schmidt-Hieber (2020), and Padilla et al. (2022). Based on $\mathcal{I}(L, W, S, B)$, our ConquerNet $\hat{f}_h$ is obtained from the ReLU networks by minimizing the convolution-type smoothed quantile loss for $\tau \in (0, 1)$, i.e.,

$$
\begin{aligned}
\hat{f}_h &:= \operatorname*{arg\,min}_{f \in \mathcal{I}(L, W, S, B), \|f\|_\infty \le F} \sum_{i=1}^{n} \ell_h \left( y_i - f(\mathbf{x}_i) \right), \\
\ell_h(u) &:= \int_{-\infty}^{\infty} \rho_\tau(v) K_h(v - u) \mathrm{d}v,
\end{aligned}
\tag{4}
$$

where $K_h(x) = K(x/h)/h$ with bandwidth $h > 0$ and $F > 0$ is a sufficiently large constant providing technical convenience. The kernel function $K(u)$ is required to be a bounded and nonnegative density function such that $\int uK(u)\mathrm{d}u = 0$, $\int K(u)\mathrm{d}u = 1$, and $\int u^2 K(u)\mathrm{d}u = \sigma_K^2$ where $\sigma_K^2 > 0$ is a constant. $K(u)$ can be chosen from commonly used kernel functions including: uniform kernel $K(u) = (1/2)\mathbf{1}(|u| \le 1)$; Gaussian kernel $K(u) = (2\pi)^{-1/2} e^{-u^2/2}$; Epanechnikov kernel $K(u) = (3/4) \left( 1 - u^2 \right) \mathbf{1}(|u| \le 1)$, etc.

For a given neural network $f \in \mathcal{I}(L, W, S, B), \|f\|_\infty \le F$, we define the $\| \cdot \|_\infty$-projection of $f_\tau^*$ onto $\mathcal{I}(L, W, S, B)$ as

$$
f_n := \operatorname*{arg\,min}_{f \in \mathcal{I}(L, W, S, B), \|f\|_\infty \le F} \|f - f_\tau^*\|_\infty,
$$

where $f_\tau^*$ is assumed to be a function that belongs to Besov spaces defined below. Note that the architectural parameters $(L, W, S, B)$ are usually chosen as functions of the sample size $n$ (see Theorem 3.1), and therefore the projection $f_n$ inherits this dependence through the network class $\mathcal{I}(L, W, S, B)$.

**Definition 2.1** (Besov space). *For a function $f \in L^p(\mathcal{X})$ and $p \in (0, \infty]$, denote the $r$-modulus of continuity as*

$$
w_{r,p}(f, t) = \sup_{\|u\|_2 \le t} \|R_u^r(f)\|_p
$$

*where*

$$
R_u^r(f) = \begin{cases} \sum_{j=0}^{r} \frac{r!}{j!(r-j)!} (-1)^{r-j} f(x + uj) & , \ \textit{if } x \in \mathcal{X}, x + ru \in \mathcal{X} \\ 0 & , \ \textit{otherwise.} \end{cases}
$$

For $q \in (0, \infty]$ and $\alpha > 0, r = \lfloor \alpha \rfloor + 1$, we define the Besov space $B_{p,q}^{\alpha}(\mathcal{X})$ as

$$B_{p,q}^{\alpha}(\mathcal{X}) = \left\{ f \in L^p(\mathcal{X}) : \|f\|_{B_{p,q}^{\alpha}(\mathcal{X})} < \infty \right\}$$

where $\|f\|_{B_{p,q}^{\alpha}(\mathcal{X})} = \|f\|_p + |f|_{B_{p,q}^{\alpha}(\mathcal{X})}$ and

$$|f|_{B_{p,q}^{\alpha}(\mathcal{X})} = \begin{cases} \left( \int_0^\infty \left( t^{-\alpha} w_{r,p}(f,t) \right)^q t^{-1} \mathrm{d}t \right)^{\frac{1}{q}} & \text{if } q < \infty, \\ \sup_{t>0} t^{-\alpha} w_{r,p}(f,t) & \text{if } q = \infty. \end{cases}$$

Throughout this paper, we write $\lceil x \rceil$ for the smallest integer greater than or equal to $x$. For any two positive real sequences $a_n$ and $b_n$, we write $a_n \asymp b_n$ if there exist constants $0 < c < C < \infty$ such that $c \leq \liminf_{n \to \infty} a_n/b_n \leq \limsup_{n \to \infty} a_n/b_n \leq C$. We write $a_n \lesssim b_n$ $(a_n \gtrsim b_n)$ if there exists constant $C > 0$ such that $a_n \leq C b_n$ $(Ca_n \geq b_n)$ for all $n$.

We introduce the performance metric in risk and empirical loss norms. For bounded functions $f$ and $g$, we define

$$\Delta_n^2(f,g) := \frac{1}{n} \sum_{i=1}^n D^2(f(\mathbf{x}_i) - g(\mathbf{x}_i)),$$

where $D^2(t) = \min\{|t|, t^2\}$. Furthermore, we define $\Delta^2(f,g) := \mathbb{E}\left( D^2(f(X) - g(X)) \right)$, $\Delta(f,g) := \sqrt{\Delta^2(f,g)}$, $\|f - g\|_{\ell_2} := \sqrt{\mathbb{E}\left( (f(X) - g(X))^2 \right)}$, and $\|f - g\|_n^2 := \frac{1}{n} \sum_{i=1}^n \left( f(\mathbf{x}_i) - g(\mathbf{x}_i) \right)^2$.

## 3 Theory

In this section, we provide statistical guarantees for our ConquerNet. Firstly, we evaluate how well our methodology can estimate the functions in Besov spaces. Our main result in Theorem 3.1 shows that our ConquerNet achieves the minimax error rate with only an additional logarithmic factor. Secondly, we develop a general risk bound on our estimator with respect to an arbitrary architecture of the neural networks, assuming the true quantile function $f_\tau^*(\cdot)$ in a space more general than the Besov spaces, which imposes no smoothness conditions. This general risk bound accommodates a broad class of network architectures and modeling objectives, indicating that our theoretical development extends beyond the minimax analysis and offers an extensible foundation for future methodological advances.

### 3.1 Minimax rate

In this subsection, we derive the convergence rate for our ConquerNet when the quantile function belongs to Besov spaces. The Besov space is a very general function class which plays an important role in fields like statistical learning (Donoho & Johnstone, 1998; Giné & Nickl, 2021; Padilla et al., 2022) and approximation analysis (Temlyakov, 1993; Suzuki, 2019). As defined in Definition 2.1, Besov spaces unify and extend many classical smoothness spaces. In particular, the Sobolev space $W^{\alpha,p}(\mathcal{X})$ coincides with the Besov space $B_{p,p}^{\alpha}(\mathcal{X})$ and the Hölder space $C^{\alpha}(\mathcal{X})$ corresponds to $B_{\infty,\infty}^{\alpha}(\mathcal{X})$. Thus, Besov space provides a more general framework that strictly contains both Sobolev and Hölder spaces that are commonly assumed for smoothness conditions in statistical guarantee analysis (Farrell et al., 2021; Montanelli, 2021; Schmidt-Hieber, 2020).

Based on the function space and network class specification, we impose the following assumptions on the data generation settings.

**Assumption 1.** We assume that $\{(\mathbf{x}_i, y_i)\}_{i=1}^n$ are i.i.d. samples from $(X, Y)$. Write $F_{y_i|\mathbf{x}_i}$ is cumulative distribution function of $y_i$ conditioning on $\mathbf{x}_i$ for $i = 1, \ldots, n$.

**Assumption 2.** There exists a constant $\kappa > 0$ such that for $\delta \in \mathbb{R}$ satisfying $|\delta| \leq \kappa$ we have that, a.s.,

$$\left| F_{y_i|\mathbf{x}_i}\left( f_\tau^*(\mathbf{x}_i) + \delta \right) - F_{y_i|\mathbf{x}_i}\left( f_\tau^*(\mathbf{x}_i) \right) \right| \geq \underline{p}|\delta| \tag{5}$$

for some constant $\underline{p} > 0$. Denote $p_{y_i|\mathbf{x}_i}(\cdot)$ as the probability density function of $y_i$ conditioning on $\mathbf{x}_i$ uniformly for all $i$. We also require that $p_{y_i|\mathbf{x}_i}(\cdot)$ is continuously differentiable and the derivative $p'_{y_i|\mathbf{x}_i}(\cdot)$ satisfies almost surely that

$$|p'_{y_i|\mathbf{x}_i}(f^*_\tau(\mathbf{x}_i) + \delta) - p'_{y_i|\mathbf{x}_i}(f^*_\tau(\mathbf{x}_i))| \leq l_0|\delta|, \tag{6}$$

for some constant $l_0 > 0$ uniformly for all $i$.

**Assumption 3.** Assume that $X$ has a bounded probability density function $g_X(\cdot) \leq c_2, c_2 > 0$ with support in $[-H, H]^d$.

**Assumption 4.** The quantile function satifies $f^*_\tau \in B^s_{p,q}\left([-H, H]^d\right), \|f^*_\tau\|_\infty \leq F$, where for $0 < p, q \leq \infty$, and $0 < s < \infty$ we have $s \geq d/p$. Furthermore, there exists $m \in \mathbb{N}$ such that $0 < s < \min\{m, m - 1 + 1/p\}$. Here, $B^s_{p,q}\left([-H, H]^d\right)$ is a Besov space in $[-H, H]^d$ as in Definition 2.1.

**Remark 3.1.** *Assumption 2 ensures the density around the target quantile is bounded away from zero. When the density around the target quantile is very small, the quantile becomes ill-conditioned, leading to inflated variance and unstable optimization.*

1. Inflated variance. *Standard quantile regression theory (e.g. Koenker & Xiao (2006)) shows that the asymptotic variance involves an inverse-density factor $1/p_{Y|X}(q_\tau(x))$. Hence, near-zero densities amplify the estimation error for any quantile estimator. In our neural network analysis, the constant appearing in (16) of Lemma A.1 which bounds $\Delta^2(f_n, f)$, is proportional to the lower density bound $1/\underline{p}$ where $\underline{p}$ is imposed in Assumption 2. This reflects the same phenomenon: extremely small density around the target quantile leads to an unfavorable constant in the convergence rate.*

2. Optimization instability. *When the density is very small, the derivative of the quantile loss is nearly constant, resulting in weak curvature and slow convergence. Smoothing via the convolution kernel can mitigate this effect to some extent, but it cannot fully eliminate the ill-posedness caused by a vanishing density.*

Condition (5) in Assumption 2 ensures local identifiability, which is a necessary condition that the quantile is estimatable. Similar conditions are widely imposed in quantile regression literature (Pollard, 1991; Belloni & Chernozhukov, 2011; Padilla et al., 2022). Meanwhile, condition (6) requires the conditional density $p_{Y|X}(t)$ to be smooth in a neighborhood of the quantile. Such Lipschitz-type conditions are common in quantile regression (Koenker & Xiao, 2006; Zhou, 2010; He et al., 2023).

Assumptions 3 and 4 are mild and commonly assumed in both quantile regression (Wu & Zhou, 2017; He et al., 2023) and deep learning studies (Padilla et al., 2022; Suzuki, 2019). Assumption 3 now only requires that $g_X(\cdot)$ is bounded on $[-H, H]^d$. The previous global lower bound $c_1$, as assumed in Padilla et al. (2022), is unnecessary for our results because regions with $g_X(\mathbf{x}) = 0$ do not contribute to the $\ell_2$ norm. This simplification does not affect identifiability or our main results, as the required local density condition is already guaranteed by Assumption 2. It is worth pointing out that Assumption 4 requires that the quantile function $f^*_\tau$ belongs to a Besov space, which, as discussed, provides a more general smoothness framework than commonly assumed Sobolev or Hölder spaces, and is flexible enough to allow even discontinuous functions.

**Theorem 3.1.** *Suppose that Assumptions 1-4 hold. Given $N \asymp n^{\frac{d}{2s+d}}$, let $\epsilon \asymp N^{-s/d} + \log^{-1} N$ with $v = sp/d - 1$, suppose $h^2 \lesssim n^{-\frac{s}{2s+d}}$ and the class $\mathcal{I}(L, W, S, B)$ satisfies*

$$L = 3 + 2\left\lceil \log_2\left(\frac{3^{\max\{d,m\}}}{\epsilon c_{d,m}}\right) + 5\right\rceil \lceil \log_2 \max\{d, m\}\rceil,$$

$$W = W_0 N, S = (L-1)W_0^2 N + N, B = O\left(N^{1/v+1/d}\right), \tag{7}$$

*for a constant $c_{d,m} > 0$ that depends on $d$ and $m$. Then there exists a constant $C > 0$ such that*

$$\mathrm{P}\left(\max\left\{\left\|\hat{f}_h - f^*_\tau\right\|^2_{\ell_2}, \left\|\hat{f}_h - f^*_\tau\right\|^2_n\right\} \leq C(\log n)^3 \max\{\delta n^{-1}, n^{\frac{-2s}{2s+d}}\}\right) \geq 1 - e^{-\delta} \log n.$$

**Remark 3.2.** *The expression for $L$ in (7) is not meant to prescribe a single fixed value. Since the quantity $\epsilon$ may be chosen within the asymptotic range $\epsilon \asymp N^{-s/d} + (\log N)^{-1}$ with $N \asymp n^{d/(2s+d)}$, the resulting depth $L$ can vary accordingly. Thus, (7) should be interpreted as describing an admissible range of depths that ensures the minimax rate, rather than requiring $L$ to take a specific exact value. The constant $C$ in the upper bound depends on the regularity assumptions on the conditional density. In particular, $C$ is proportional to $c_2$ in Assumption 3 and inversely proportional to the lower bound $\underline{p}$ in Assumption 2.*

Note that the rate $n^{-\frac{2s}{2s+d}}$ is known to be minimax optimal for function estimation in Besov spaces (Kerkyacharian & Picard, 1992; Donoho & Johnstone, 1998; Suzuki, 2019). Theorem 3.1 explicitly yields the minimax rate $O_p(n^{-\frac{2s}{2s+d}} \log^3 n)$ when $\delta \asymp \log n$ in Besov space up to constants and logarithmic factors. Our estimator attains this minimax rate up to a logarithmic factor $\log^3 n$, implying that no estimator can substantially improve upon the performance of our ConquerNet beyond this logarithmic term. The minimax rate also clarifies the role of the bandwidth parameter $h$. Specifically, the condition $h^2 \lesssim n^{-\frac{s}{2s+d}}$ ensures that the smoothing bias remains negligible, so that the estimator still targets the true quantile function. Conversely, when $h$ is extremely small, the convolution-type smoothed loss function $\ell_h(y_i - f(\mathbf{x}_i))$ effectively reduces to the original quantile loss $\rho_\tau(y_i - f(\mathbf{x}_i))$; see Figure 1. Smaller $h$ leads to a sharper minimum and a smaller smoothing region, which makes the gradients change dramatically in this region. The excessive sharpness induced by very small $h$ will increase sensitivity to parameter changes, which reduces optimization stability and leads to poor generalization.

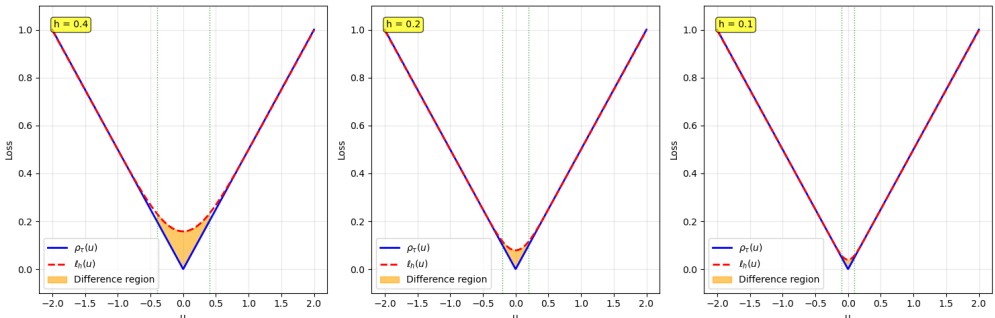

Figure 1: Relationship between loss functions $\rho_\tau(u)$ (solid lines) and $\ell_h(u)$ (dashed lines) for $h = 0.4, 0.2, 0.1$, with $\tau = 0.5$. Smaller $h$ yields smaller difference region between $\rho_\tau(u)$ and $\ell_h(u)$.

Theorem 3.1 is a statistical no-harm result relative to the nonsmoothed quantile ReLU network of Padilla et al. (2022). Under exact empirical optimization, convolution smoothing preserves the near-minimax rate. Its additional benefit concerns the finite-step optimization path. Although smooth losses have shown empirical benefits for DNN training (Berrada et al., 2018), a global iteration-complexity analysis for the full nonconvex problem remains difficult (Boob et al., 2022). We therefore provide a tractable explanation by focusing on the output-layer subproblem and comparing the finite-step empirical optimization. Such output-layer analysis is aligned with widely used perspectives in neural-network methodology and theory literature (Carratino et al., 2018; Jacot et al., 2018; Wei & Khardon, 2024).

Specifically, fix the hidden-layer parameters $\{A^{(l)}, b^{(l)}\}_{l=1}^{L-1}$ in (3). Define $\mathbf{z}_i^{(l)} = \sigma\left(A^{(l)}\mathbf{z}_i^{(l-1)} + b^{(l)}\right)$, $l = 1, \dots, L-1$ with $\mathbf{z}_i^{(0)} = \mathbf{x}_i$, then $f_{A^{(L)}, b^{(L)}}(\mathbf{x}_i) = A^{(L)}\mathbf{z}_i^{(L-1)} + b^{(L)}$. Let $\Theta_L$ be the compact convex output-layer parameter set. Define the output-layer empirical objective

$$\mathcal{Q}_{n,h}(A,b) = \frac{1}{n}\sum_{i=1}^{n} \ell_h\left(y_i - A\mathbf{z}_i^{(L-1)} - b\right), \tag{8}$$

with the convention $\ell_0 = \rho_\tau$, so that $h = 0$ corresponds to the nonsmoothed quantile neural network. If $(A_T^{(h)}, b_T^{(h)})$ is obtained using at most $T$ projected first-order iterations, its empirical optimization gap is

$$\eta_T(h) = \mathcal{Q}_{n,h}\left(A_T^{(h)}, b_T^{(h)}\right) - \min_{(A,b)\in\Theta_L} \mathcal{Q}_{n,h}(A,b). \tag{9}$$

Proposition A.7 shows that $\eta_T(0) = O(T^{-1/2})$, $\eta_T(h) = O\left(\min\left\{T^{-1/2}, h^{-1}T^{-1}\right\}\right)$. Therefore, a sufficient iteration budget for an $\varepsilon$-accurate output-layer solution improves from $O(\varepsilon^{-2})$ to $O(\min\{\varepsilon^{-2}, h^{-1}\varepsilon^{-1}\})$. Together with Theorem 3.1, this shows that, under $h^2 \lesssim n^{-s/(2s+d)}$, ConquerNet reduces the finite-step optimization budget while preserving the near-minimax statistical rate.

## 3.2 General upper bound

While the minimax analysis in the previous subsection relies on structural assumptions on the neural networks and target function, it is also of interest to study performance guarantees in a more general setting without such restrictions. To this end, we provide a general upper bound for our ConquerNet estimator without imposing constraints on the width, depth, magnitude of parameters, and sparsity of the network class, or on the smoothness assumptions (such as belonging to a Besov space) on the target quantile function $f_\tau^*$.

Consider a general neural network function with ReLU activation $f \in \mathcal{F}(P, U, L)$ denoted as

$$\mathcal{F}(P, U, L) := \{f : \mathbb{R}^{p_0} \to \mathbb{R}^{p_{L+1}}, \quad \mathbf{x} \mapsto f(\mathbf{x}) = W_L \sigma_{\mathbf{v}_L} \circ W_{L-1} \sigma_{\mathbf{v}_{L-1}} \circ \cdots \circ W_1 \sigma_{\mathbf{v}_1} \circ W_0 \mathbf{x}\}, \qquad (10)$$

where $W_i$ is a $p_{i+1} \times p_i$ weight matrix, $\sigma_{\mathbf{v}_i}$ is a shifted activation function with shifting vector $\mathbf{v}_i = (v_{i,1}, ..., v_{i,p_i})^\top \in \mathbb{R}^{p_i}$, i.e.,

$$\sigma_{\mathbf{v}_i} \begin{pmatrix} a_1 \\ \vdots \\ a_{p_i} \end{pmatrix} = \begin{pmatrix} \sigma\left(a_1 - v_{i,1}\right) \\ \vdots \\ \sigma\left(a_{p_i} - v_{i,p_i}\right) \end{pmatrix}.$$

Notice that the number of parameters is $P = \sum_{l=0}^{L}(p_{l+1}p_l + p_{l+1})$ with $p_0 = d, p_{L+1} = 1$, the number of nodes is $U = \sum_{l=1}^{L} p_l$, and the number of layers is $L$. For $f \in \mathcal{F}(P, U, L)$, we redefine its ConquerNet estimator

$$\hat{f}_h := \underset{f \in \mathcal{F}(P,U,L), \|f\|_\infty \leq F}{\arg\min} \sum_{i=1}^{n} \ell_h\left(y_i - f(\mathbf{x}_i)\right), \qquad (11)$$

and the approximation error is defined as

$$err_1 := \mathbb{E}\left[\frac{1}{n}\sum_{i=1}^{n} \ell_h\left(y_i - f_n\left(\mathbf{x}_i\right)\right) - \frac{1}{n}\sum_{i=1}^{n} \ell_h\left(y_i - f_\tau^*\left(\mathbf{x}_i\right)\right)\right] \qquad (12)$$

where

$$f_n := \underset{f \in \mathcal{F}(P,U,L), \|f\|_\infty \leq F}{\arg\min} \mathbb{E}\left[\sum_{i=1}^{n} \ell_h\left(y_i - f\left(\mathbf{x}_i\right)\right)\right]. \qquad (13)$$

**Theorem 3.2.** *Suppose that Assumptions 1-2 hold and $n \geq CLP\log(U)$ for a sufficiently large $C > 0$. Then with $c_1 > 0$ a constant, $\hat{f}_h$ defined in (11) satisfies*

$$\mathbb{E}\left[\Delta_n^2\left(\hat{f}_h, f_\tau^*\right)\right] \leq c_1\left[F\left(\frac{LP\log U \cdot \log n}{n}\right)^{1/2} + h^2 \mathbb{E}\|\hat{f}_h - f_\tau^*\|_{n,1} + err_1\right], \qquad (14)$$

*where $\|f - g\|_{n,1} = \frac{1}{n}\sum_{i=1}^{n}|f(\mathbf{x}_i) - g(\mathbf{x}_i)|$. Furthermore, if $h^2 = o\left(\sqrt{\mathbb{E}\|\hat{f}_h - f_\tau^*\|_n^2}\right)$, it also holds that*

$$\mathbb{E}\left[\left\|\hat{f}_h - f_\tau^*\right\|_n^2\right] \leq 2c_1 \max\{1, F\}\left[F\left(\frac{LP\log U \cdot \log n}{n}\right)^{1/2} + err_1\right]. \qquad (15)$$

**Remark 3.3.** *Our general risk bound in Theorem 3.2 is quite flexible and does not rely on a very rigid architecture, but rather on general capacity measures (e.g., network class size, sparsity, norm bounds) and approximation properties. Therefore, in principle, it can accommodate more complex architectures, including residual (skip-connected) MLPs, as long as we can control the relevant complexity measures. However, to*

*the best of our knowledge, there is no existing minimax-rate analysis for quantile regression specifically with residual-based (ResNet/skip-connection) networks. The most closely related work is the minimax-rate analysis for deep ReLU networks under quantile loss by Padilla et al. (2022), which considers plain feedforward (non-skip) ReLU networks. Given the lack of minimax theory for residual architectures in quantile regression, we focused on the simpler network class to establish minimax optimality for our proposed ConquerNet method, and leave for the possible extension to MLPs with skip connections as a promising future work.*

Theorem 3.2 drops out Assumptions 3-4 in Theorem 3.1 and yields a general error bound that depends on parameters $P, U, L$, the approximation error $err_1$, and the sample size $n$. As long as $h^2 = o(\sqrt{\mathbb{E}\|\hat{f}_h - f_\tau^*\|_n^2})$, the risk bound in (15) remains essentially unaffected, implying that taking $h$ sufficiently small does not deteriorate the estimator's performance. However, from an optimization perspective, a very small $h$ causes the smoothed loss $\ell_h(\cdot)$ to behave almost identically to the original quantile loss $\rho_\tau(\cdot)$; see Figure 1, which raises sharp minima concerns discussed in Section 1.

## 4 Empirical study

In this section, we empirically evaluate the proposed ConquerNet through comprehensive simulation studies and a real data application. We use the convolution-type smoothed quantile losses defined in (4) by three different kernel functions: Gaussian, uniform, and Epanechnikov kernels. We compare them against the baseline quantile ReLU network in Padilla et al. (2022) under different sample sizes and quantile levels. The simulation results show that ConquerNet's MSE gains become more consistent as sample size grows, with this transition especially sharp at the fixed low and high quantile levels; ConquerNet also requires less training time. We also discuss the impact of different choices of the bandwidth $h$ and propose a data-driven rule for bandwidth selection, which shows the stability of our ConquerNet. We apply the ConquerNet to the BMI (body mass index) dataset and get better performance compared to the baseline model. In addition, we explore the loss landscape plots of the two networks, which provide an intuitive way to understand the advantage of the ConquerNet during optimization. Due to the page limit, we only present part of our experimental results in this section. See Appendix B for extra experiments, including the tables of MAE (mean absolute error), joint estimation of multiple quantile levels under non-crossing constraints, residual-based networks, and the plot of MSEs by sample sizes.

We have made efforts to ensure reproducibility of our results. The experimental setup, including simulation designs, model configurations, and hyperparameter choices, is described in this section and Appendix B. Random seeds are set to ensure reproducibility of the experiment. Upon publication, we will release the full implementation, including training scripts and simulation setups, in a GitHub repository. An anonymous version of the GitHub repository is available during the review, see `https://anonymous.4open.science/r/conquernn-F625/`.

### 4.1 Scenario settings

Under the comparable number of parameters, we consider two networks with different shapes. One has 5 hidden layers of 70 nodes each, denoted by Model A, and the other one has 10 hidden layers of 50 nodes each, denoted by Model B. We consider both smooth and piecewise continuous quantile functions with heavy-tailed noises. Specifically, the data for the simulation are generated by the following mechanism,

$$y_i = g(\mathbf{x}_i) + \varepsilon_i, \quad i = 1, \ldots, n,$$

where $g(\cdot) : [0,1]^d \to \mathbb{R}$, $\{\mathbf{x}_i\}_{i=1}^n$ are independently sampled from uniform distribution on $[0,1]^d$ and $d$ is the dimension of $\mathbf{x}_i$. We consider the following 3 scenarios of data generation:

**Scenario 1 (S1).** We set $d = 2$, $\mathbf{z} \in [0,1]^2$, $g(\mathbf{z}) = \cos(2\pi z_1^2) + \sin\left(\sqrt{z_1^2 + 2z_2} + 2\right)$, and $\varepsilon_i = \|\mathbf{x}_i - (1,0)^\top\| t_i/2$, where $t_i \stackrel{\text{iid}}{\sim} t(2)$ for $i = 1, \ldots, n$ and $t(2)$ is the t-distribution with 2 degrees of freedom.

**Scenario 2 (S2).** We set $d = 5$, $\mathbf{z} \in [0,1]^5$, $g(\mathbf{z}) = \sqrt{z_1 + 2z_2 + z_3 + 2z_4 + z_5}$, and $\varepsilon_i = \sqrt{\mathbf{x}_i^\top \eta} v_i$ for $i = 1, \ldots, n$, where $\eta = (1/2, 0, 1/2, 0, 1/2)^\top$ and $v_i \stackrel{\text{iid}}{\sim} t(3)$, $t(3)$ is the t-distribution with 3 degrees of freedom.

**Scenario 3 (S3).** We set $d = 5$, $z \in [0,1]^5$, $g(\mathbf{z}) = g_2 \circ g_1(\mathbf{z})$, where $g_1(\mathbf{z}) = (z_1 + 3z_2, \cos(2\pi(z_3 + z_4)), z_2 + \sqrt{z_3} + 2z_5)^\top$ and

$$g_2(\mathbf{z}) = \begin{cases} z_1 + \sqrt{z_2^2 + z_3} & \text{if } z_2 < 0, \\ \sqrt{z_1 + z_2} + 0.5z_3 & \text{otherwise,} \end{cases}$$

with $\varepsilon_i \overset{\text{iid}}{\sim} \text{Laplace}(0,2)$ for $i = 1, \ldots, n$.

## 4.2 Experiment details

Our training data set is denoted by $\{(\mathbf{x}_i, y_i)\}_{i=1}^n$ and the test data set is denoted by $\{\tilde{\mathbf{x}}_i\}_{i=1}^n$. For a fixed quantile $\tau \in (0,1)$, we train the baseline network and ConquerNet from (4), and get the trained quantile estimate $\hat{f}_h$. By the data generating mechanism described above, we can calculate the true quantile $f_\tau^*(\tilde{\mathbf{x}}_i) = g(\tilde{\mathbf{x}}_i) + F_{\tilde{\varepsilon}_i | \tilde{\mathbf{x}}_i}^{-1}(\tau)$. Then the MSE of quantile estimator is obtained by $\sum_{i=1}^T (f_\tau^*(\tilde{\mathbf{x}}_i) - \hat{f}_h(\tilde{\mathbf{x}}_i))^2 / T$, which is evaluated by using $\{\tilde{\mathbf{x}}_i\}_{i=1}^T$ of size $T = 10000$. We set training sample sizes $n \in \{1000, 5000, 10000\}$ and quantile levels $\tau \in \{0.05, 0.25, 0.5, 0.75, 0.95\}$. For each experiment setting, we run the experiment 50 times independently and get the MSE results 50 times. Table 1 shows the averaged MSE results over 50 trials under different experiment settings. The bold fonts represent that the results of the ConquerNet are better than those of the baseline models. In addition, to make the representation clearer, we box the baseline result if it outperforms our ConquerNet.

Table 1: Mean squared error (MSE) performances for scenario 1-3, model A and B under different sample sizes, quantile levels, and smoothing kernels. The MSEs are averaged over 50 independent trials.

| | Method | n=1000 | | | | | n=5000 | | | | | n=10000 | | | | |
|---|---|---|---|---|---|---|---|---|---|---|---|---|---|---|---|---|
| | | $\tau{=}0.05$ | $\tau{=}0.25$ | $\tau{=}0.5$ | $\tau{=}0.75$ | $\tau{=}0.95$ | $\tau{=}0.05$ | $\tau{=}0.25$ | $\tau{=}0.5$ | $\tau{=}0.75$ | $\tau{=}0.95$ | $\tau{=}0.05$ | $\tau{=}0.25$ | $\tau{=}0.5$ | $\tau{=}0.75$ | $\tau{=}0.95$ |
| **S1** Model A | Baseline | [0.3820] | 0.0402 | 0.0278 | 0.0374 | 0.3784 | [0.0996] | 0.0120 | 0.0086 | 0.0108 | [0.0939] | [0.0618] | 0.0067 | 0.0047 | 0.0063 | 0.1366 |
| | Gaussian | 0.4354 | **0.0383** | **0.0224** | 0.0382 | 0.5035 | 0.1124 | **0.0087** | **0.0055** | **0.0097** | 0.1081 | 0.0678 | **0.0064** | **0.0034** | **0.0055** | **0.0625** |
| | Uniform | 0.4448 | **0.0385** | **0.0224** | 0.0328 | **0.3721** | 0.1079 | **0.0087** | **0.0059** | 0.0104 | 0.1312 | 0.0789 | **0.0058** | **0.0036** | 0.0057 | 0.0543 |
| | Epanechnikov | 0.6733 | **0.0390** | **0.0222** | 0.0373 | 0.5913 | 0.1251 | **0.0093** | **0.0055** | 0.0095 | 0.1169 | 0.0653 | **0.0061** | **0.0037** | 0.0053 | 0.0665 |
| Model B | Baseline | [0.3842] | [0.0527] | [0.0319] | [0.0475] | [0.4222] | 0.1143 | 0.0149 | 0.0107 | 0.0151 | 0.1277 | 0.1691 | 0.0099 | 0.0066 | 0.0082 | 0.3882 |
| | Gaussian | 0.4158 | 0.0665 | 0.0383 | 0.0633 | 0.5202 | 0.1172 | **0.0144** | **0.0097** | **0.0142** | **0.1066** | **0.1062** | **0.0095** | **0.0055** | 0.0086 | **0.0726** |
| | Uniform | 0.5652 | 0.0612 | 0.0378 | 0.0614 | 0.5685 | **0.1033** | **0.0145** | **0.0091** | 0.0154 | **0.1252** | **0.0974** | **0.0093** | **0.0055** | 0.0080 | **0.0736** |
| | Epanechnikov | 0.4275 | 0.0582 | 0.0383 | 0.0626 | 0.6050 | **0.1115** | **0.0146** | **0.0094** | 0.0145 | **0.1162** | **0.1294** | **0.0089** | **0.0057** | 0.0092 | 0.0663 |
| **S2** Model A | Baseline | 0.8292 | 0.0868 | 0.0619 | 0.0839 | [0.7874] | 0.2752 | [0.0275] | 0.0222 | 0.0308 | 0.2747 | 0.1704 | 0.0205 | 0.0145 | 0.0223 | 0.1670 |
| | Gaussian | **0.7994** | **0.0711** | **0.0587** | **0.0778** | 1.1466 | **0.2202** | 0.0276 | **0.0169** | **0.0273** | 0.2598 | **0.1316** | **0.0176** | **0.0128** | **0.0193** | **0.1537** |
| | Uniform | 0.8892 | **0.0721** | **0.0471** | 0.0964 | 0.8566 | **0.2308** | 0.0286 | **0.0181** | **0.0275** | 0.2586 | **0.1457** | **0.0180** | **0.0128** | **0.0191** | **0.1467** |
| | Epanechnikov | 0.9192 | **0.0787** | **0.0522** | 0.1048 | 0.9074 | **0.2415** | 0.0284 | **0.0177** | **0.0265** | 0.2492 | **0.1430** | **0.0162** | **0.0126** | **0.0191** | **0.1388** |
| Model B | Baseline | 0.4930 | [0.0583] | [0.0493] | 0.0840 | [0.5898] | 0.1732 | 0.0257 | 0.0178 | 0.0300 | 0.1966 | 0.1323 | 0.0202 | 0.0129 | 0.0220 | 0.1358 |
| | Gaussian | 0.7367 | 0.0639 | 0.0500 | **0.0759** | 0.6273 | 0.1800 | **0.0246** | **0.0155** | **0.0245** | 0.2157 | **0.1085** | **0.0160** | **0.0119** | **0.0178** | **0.1144** |
| | Uniform | **0.4515** | 0.0717 | 0.0503 | 0.0935 | 0.6034 | **0.1706** | **0.0216** | **0.0176** | **0.0241** | 0.2239 | **0.1172** | **0.0151** | **0.0118** | **0.0175** | **0.1342** |
| | Epanechnikov | 0.5800 | 0.0723 | 0.0546 | **0.0819** | 0.7484 | 0.2151 | 0.0263 | **0.0159** | **0.0289** | 0.1802 | **0.1038** | **0.0146** | **0.0107** | **0.0173** | **0.1302** |
| **S3** Model A | Baseline | 3.0766 | 0.7564 | 0.5350 | 0.7407 | [2.8969] | 1.2870 | 0.3854 | 0.2146 | 0.3387 | 1.3495 | 0.9232 | 0.2420 | 0.1459 | 0.2413 | 0.9960 |
| | Gaussian | **2.8841** | 0.7776 | **0.4788** | **0.7056** | 3.4222 | 1.3139 | **0.3379** | **0.1993** | **0.3269** | **1.1897** | **0.8395** | **0.2040** | **0.1295** | **0.2145** | **0.7477** |
| | Uniform | 3.3730 | 0.7735 | **0.4876** | 0.7832 | 3.3576 | **1.1577** | **0.3449** | **0.1912** | **0.3082** | **1.2392** | **0.8790** | **0.2064** | **0.1300** | **0.2146** | **0.7824** |
| | Epanechnikov | 3.4479 | **0.7308** | **0.4679** | 0.8153 | 3.3156 | **1.1229** | **0.3615** | **0.1980** | **0.3112** | **1.3312** | **0.8230** | **0.2129** | **0.1349** | **0.2206** | **0.7998** |
| Model B | Baseline | 2.8166 | [0.7558] | 0.5175 | 0.7665 | 2.2839 | 1.0061 | [0.3596] | [0.2196] | 0.3551 | 1.0990 | 0.7786 | [0.2306] | 0.1391 | 0.2380 | 0.7249 |
| | Gaussian | **2.3193** | 0.8405 | **0.5142** | **0.7543** | 2.6520 | 1.0602 | 0.4263 | 0.2304 | **0.3525** | **1.0681** | 0.8256 | 0.2518 | 0.1416 | 0.2444 | 0.7536 |
| | Uniform | **2.1946** | 0.8316 | 0.5193 | **0.7352** | 2.5765 | 1.2056 | 0.4038 | 0.2320 | **0.3373** | **1.0528** | **0.7167** | 0.2409 | **0.1372** | 0.2493 | **0.6920** |
| | Epanechnikov | 2.9702 | 0.9008 | **0.4892** | 0.8702 | **2.1331** | **1.0053** | 0.3750 | 0.2297 | **0.3473** | **1.0895** | **0.7502** | 0.2454 | **0.1390** | **0.2315** | **0.6683** |

Table 1 first shows a strong sample-size dependence in relative MSE performance. Pooling the three kernels, three scenarios, and two architectures, the number of favorable comparisons increases from 33/90 at $n = 1000$ to 65/90 at $n = 5000$ and 77/90 at $n = 10000$. At the fixed low ($\tau = 0.05$) and high ($\tau = 0.95$) quantiles, the corresponding counts increase more sharply, from 7/36 to 22/36 and 31/36. Thus, the tail advantage is not present at small $n$ but becomes highly consistent as the sample size grows. A sample size of 1000 is inadequate for model training in our settings and is much smaller than the number of network parameters. This pattern is consistent with the discontinuous gradient of the original quantile loss when residuals cross zero, whereas convolution smoothing yields a smoother optimization objective. Second, Scenario 3 has discontinuity points in the function $g(\cdot)$, while $g(\cdot)$ is smooth in Scenario 1 and Scenario 2. The MSE results are consistent with Theorem 3.1 in the sense that MSE is smaller when the smoothness $s$ increases. Third, Model B has more hidden layers than Model A, while the number of parameters is close. Meanwhile, $d = 2$ in Scenario 1, $d = 5$ in Scenario 2 and 3. The MSE results show that when $d$ is small, shallow networks are more suitable. In

contrast, for $d = 5$, deep networks perform better, which confirms the layer condition for the minimax rate in (7) of Theorem 3.1 to some extent. Beyond the mean MSEs in Table 1, we additionally assess the standard errors and quantile calibration using the empirical-coverage absolute bias. In addition to the original baseline, we also add two complementary quantile-regression baselines to the simulations in Scenarios S1–S3. These additional tables and detailed discussion are provided in Appendix B.

Table 1 is intentionally a comprehensive performance comparison rather than a one-factor ablation: it summarizes the overall smoothing effect across scenarios, architectures, sample sizes, quantile levels, and kernels. This comparison may also reflect auxiliary choices, including the kernel, bandwidth, stopping rule, network architecture, and training protocol. We therefore provide a series of supplementary experiments in Appendix B to isolate the smoothing effect from these factors. Due to the page limit, we cannot present all of these results in the main text. Instead, the following summary identifies the comparisons used to examine each factor.

| Effect examined | Evidence tables |
|---|---|
| Overall smoothing effect and uncertainty | Table 1; Tables 5 and 6 |
| Kernel choice | Across all experiments |
| Bandwidth selection and sensitivity | Tables 7 and 8 |
| Stopping rule | Tables 3 and 4 |
| Tuning protocol | Tables 12, 13, and 15 |
| Network architecture and implementation | Tables 2, 9, 10, 11, 12, 13, and 15 |

### 4.3 Choice of bandwidth $h$

The bandwidth $h$ should be chosen properly based on the theoretical results in Section 3 and the experiments. As shown in Figure 1, the difference between smoothed loss $\ell_h(u)$ and quantile loss $\rho_\tau(u)$ grows larger with $h$ increases. Furthermore, as $h$ increases, the smoothed area becomes larger and the gradients become smaller, which affects the efficiency of SGD. On the other hand, too small $h$ reduces to the original quantile loss and increases the sharpness of the minima, which tends to result in poor generalization. Therefore, a proper choice of bandwidth is necessary. By Theorem 3.1, we accept a bigger $h$ when the sample size $n$ is small and a smaller $h$ when $n$ is big. For example in Table 1, we take $h = 0.01/0.005/0.001$ for $n = 1000/5000/10000$ in Scenario 2, Model A. We find that the performance remains outstanding with a wide range of bandwidth $h$ for $n = 10000$, see Table 7 in Appendix B, which shows the stability of the results for different bandwidth choices.

We also make it clear to the data-driven rules for bandwidth selection. In detail, we propose the K-fold cross-validation algorithm for the bandwidth selection. For a candidate list of bandwidths, $h \in \{0.001, 0.005, 0.01, 0.05, 0.1\}$ for example, we train the models on the training set and calculate the pinball loss on the validation set for K times. Select the bandwidth with the minimum mean validation loss. The complete MSE results are shown in Table 8 in Appendix B. By the cross-validation algorithm, our ConquerNet still outperform the baseline models, especially for large sample sizes, which shows the stability of our method.

### 4.4 Real data analysis

We evaluate ConquerNet on two real datasets: BMI and California Housing. The data sources, dataset details, and preprocessing procedures are provided in Appendix B. For both datasets, we use the same single-quantile training protocol at $\tau \in \{0.05, 0.25, 0.5, 0.75, 0.95\}$. Table 2 reports the pinball losses and 95% paired-bootstrap confidence intervals for the reduction in pinball loss achieved by ConquerNet relative to the baseline on the test data sets. On BMI, ConquerNet has a lower point estimate in 23/30 comparisons; at the 5% significance level, eight comparisons favor ConquerNet and two favor the baseline. On California Housing, ConquerNet has a lower point estimate in 10/15 comparisons, and all 10 improvements are significant at the 5% level. Because comparison-by-comparison results may raise multiple-testing concerns, we additionally report a pooled estimand for each dataset. The pooled absolute reductions are 0.00228 (95% CI $[0.00106, 0.00350]$, $p$-value$= 0.00029$) for BMI and 0.004425 (95% CI $[0.003419, 0.005413]$, $p$-value$= 0.00002$) for California Housing.

Additional quantile-regression diagnostics are reported in Appendix B, demonstrating our ConquerNet achieves better performance in most cases.

Table 2: Pinball loss performance for BMI and California Housing datasets. Each ConquerNet entry reports the test pinball loss, with the pointwise 95% basic paired-bootstrap confidence interval for the reduction in pinball loss shown underneath in brackets. A lower bound greater than zero indicates a significant reduction. Bold point results indicate a lower loss than the baseline; bold confidence intervals indicate statistically significant improvements.

| Dataset | Subset | Method | $\tau = 0.05$ | $\tau = 0.25$ | $\tau = 0.5$ | $\tau = 0.75$ | $\tau = 0.95$ |
|---|---|---|---|---|---|---|---|
| BMI | Male | Baseline | 0.5231 | 2.0638 | 2.7387 | 2.0534 | 0.5190 |
| | | Gaussian | **0.5221** [-0.00051, 0.00254] | **2.0609** [-0.00161, 0.00717] | **2.7384** [-0.00392, 0.00456] | **2.0513** [-0.00147, 0.00570] | **0.5189** [-0.00062, 0.00074] |
| | | Uniform | **0.5217** **[0.00011, 0.00262]** | **2.0618** [-0.00225, 0.00612] | 2.7403 [-0.00738, 0.00422] | **2.0521** [-0.00417, 0.00682] | 0.5192 [-0.00094, 0.00043] |
| | | Epanechnikov | **0.5218** **[0.00001, 0.00252]** | **2.0636** [-0.00408, 0.00421] | 2.7389 [-0.00473, 0.00438] | **2.0518** [-0.00254, 0.00591] | 0.5205 **[-0.00279, -0.00020]** |
| | Female | Baseline | 0.5218 | 2.0596 | 2.7366 | 2.0555 | 0.5249 |
| | | Gaussian | **0.5202** **[0.00026, 0.00290]** | **2.0446** **[0.00539, 0.02452]** | **2.7323** [-0.00166, 0.01032] | **2.0531** [-0.00078, 0.00567] | 0.5259 [-0.00247, 0.00079] |
| | | Uniform | **0.5211** [-0.00081, 0.00205] | **2.0497** **[0.00176, 0.01806]** | **2.7311** **[0.00182, 0.00910]** | 2.0642 [-0.01513, -0.00221] | 0.5251 [-0.00084, 0.00057] |
| | | Epanechnikov | **0.5205** [-0.00007, 0.00265] | **2.0454** **[0.00587, 0.02247]** | **2.7276** **[0.00205, 0.01589]** | **2.0523** [-0.00028, 0.00671] | **0.5248** [-0.00054, 0.00091] |
| Housing | All | Baseline | 0.0426 | 0.1418 | 0.1879 | 0.1698 | 0.0671 |
| | | Gaussian | 0.0436 [-0.00192, -0.00016] | **0.1336** **[0.00582, 0.01067]** | **0.1823** **[0.00338, 0.00779]** | **0.1630** **[0.00409, 0.00951]** | **0.0645** **[0.00071, 0.00429]** |
| | | Uniform | 0.0434 [-0.00182, -0.00002] | **0.1327** **[0.00721, 0.01096]** | **0.1815** **[0.00439, 0.00841]** | **0.1590** **[0.00802, 0.01358]** | 0.0680 [-0.00292, 0.00105] |
| | | Epanechnikov | 0.0438 [-0.00219, -0.00013] | **0.1391** **[0.00093, 0.00449]** | **0.1782** **[0.00709, 0.01236]** | **0.1609** **[0.00617, 0.01165]** | 0.0676 [-0.00233, 0.00132] |

### 4.5 Plots of loss landscape

To express the benefit of the ConquerNet more intuitively, we tried to plot the loss landscape of the baseline network and the ConquerNet based on Li et al. (2018), which are shown in Figure 2. The networks have the same structures, consisting of 20 layers, each with 35 nodes. We generated two random directions and ensured the models shared the same directions. Then we used the filter-wise normalization method in Li et al. (2018) and calculated the loss surfaces. The loss landscape plot shows that the baseline network (see the subplot on the left) has at least three local minima. In contrast, the ConquerNet in the right subplot has one minimum, which implies that the ConquerNet improves the training dynamics.

## 5 Discussion

Our smoothing principle can potentially be extended beyond quantile regression to other distributional objectives such as CRPS and Wasserstein distances. As discussed in Berrisch & Ziel (2023), CRPS can be expressed as an integral of quantile losses over $\tau \in [0, 1]$. Therefore, one can consider the smoothed objective

$$\min_{f_\tau \in \mathcal{F}(P, U, L),\, \tau \in [0,1]} \frac{1}{n} \sum_{i=1}^{n} \int_0^1 \ell_h(y_i - f_\tau(x_i)) \, \mathrm{d}\tau,$$

where $\ell_h$ is our convolution-type smoothed quantile loss.

Regarding the potential extension to Wasserstein-based objectives, consider the empirical Wasserstein-$p$ distance between the predictive CDF $F_f(\cdot|x_i)$ ($F_f^{-1}(\tau|x_i) = f_\tau(x_i)$) and the target CDF $G(\cdot|x_i)$

$$W_p(F_f, G) = \sum_{i=1}^{n} \left( \int_0^1 \left| f_\tau(x_i) - G^{-1}(\tau|x_i) \right|^p d\tau \right)^{1/p}.$$

It is important to note that directly minimizing a Wasserstein distance is not straightforward in our conditional quantile regression, since the target $G^{-1}(\tau|x_i)$ is unknown. One alternative is to use the empirical distribution

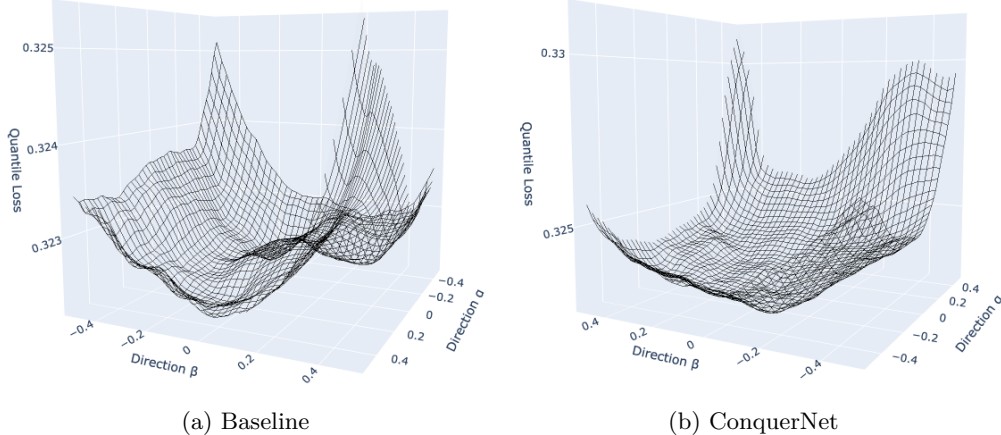

(a) Baseline          (b) ConquerNet

Figure 2: Plot of 3D loss landscape of scenario 2, $\tau = 0.5$. The subplot on the left represents the baseline model, and the right subplot represents the ConquerNet smoothed by the Gaussian kernel with $h = 0.1$. Both networks consist of 20 layers, each with 35 nodes.

$G_n$ as the target, i.e.,

$$\hat{W}_p(F_f, G_n) = \sum_{i=1}^{n} \left( \int_0^1 \left| f_\tau(x_i) - G_n^{-1}(\tau|x_i) \right|^p d\tau \right)^{1/p},$$

where

$$G_n^{-1}(\tau|x_i) = \inf \left\{ y : \frac{1}{|\mathcal{N}_i|} \sum_{j \in \mathcal{N}_i} \mathbf{1}_{\{y_j \leq y\}} \geq \tau \right\}, \quad \mathcal{N}_i = \{j : x_j = x_i\}.$$

However, this requires repeated observations $(y_j, x_j)$ at the same covariate $x_j = x_i$. Such a setting arises in reinforcement learning (Dabney et al., 2018), where multiple returns $y_i$ for a fixed state $x_i$ can be observed under different rollouts. In Bayesian settings, Zhang et al. (2020) chain quantile regression with Wasserstein-based objectives for posterior inference. Some existing literature provides partial connections between the quantile loss and Wasserstein objectives. Lheritier & Bondoux (2022) discussed that optimization based on the quantile loss can be interpreted as a 1-Wasserstein projection in certain settings, while Yang & Wang (2024) introduces a Wasserstein-improved objective for composite quantile regression. While a direct application is beyond the scope of our current work, these studies highlight potential directions for future research.

ConquerNet preserves the near-minimax statistical rate of the nonsmoothed quantile network and improves the finite-step optimization guarantee for the output-layer subproblem, but it does not attain a faster minimax statistical rate or establish globally faster optimization for the joint training of all network layers. The empirical gains are not uniform across all finite-sample settings, but they become increasingly consistent as sample size grows, especially at the fixed low and high quantiles. The method is therefore primarily intended for sufficiently complex quantile networks with sufficiently large data that optimization constitutes a material bottleneck; the present results do not imply universal superiority over simpler models or all existing quantile-regression methods. Moreover, the theoretical guarantees are pointwise for a fixed $\tau \in (0, 1)$ and do not cover extreme-quantile asymptotics with $\tau \to 0$ or 1. Establishing full-network optimization guarantees and extending the theory to drifting extreme quantiles remain important directions for future work.

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

## A  Proofs

We first state several definitions to develop our Empirical process theorems and auxiliary lemmas. Define the empirical loss function as

$$\hat{M}_n(f) = \sum_{i=1}^n \hat{M}_{n,i}(f), \quad \hat{M}_{n,i}(f) = \frac{1}{n}(\ell_h(y_i - f(\mathbf{x}_i)) - \ell_h(y_i - f_n(\mathbf{x}_i))),$$

and we set

$$M_n(f) = \mathbb{E}\left\{\frac{1}{n}\sum_{i=1}^n [\ell_h(y_i - f(\mathbf{x}_i)) - \ell_h(y_i - f_n(\mathbf{x}_i))]\right\}.$$

For $\epsilon > 0$ and a metric $\mathrm{dist}(\cdot, \cdot)$ on the class of functions $\mathcal{F}$, we define the covering number $\mathcal{N}(\epsilon, \mathcal{F}, \mathrm{dist}(\cdot, \cdot))$ as the minimum number of balls of the form $\{g : \mathrm{dist}(g, f) \le \epsilon\}$, with $f \in \mathcal{F}$, needed to cover $\mathcal{F}$ (see Definition 2.2 in Sen (2018) for details). We write $\tilde{\mathcal{I}}(L, W, S, B) = \{f : f \in \mathcal{I}(L, W, S, B), \|f\|_\infty \le F\}$. For simplicity, we consider $[0, 1]^d$ instead of $[-H, H]^d$ in Assumptions 3-4.

### A.1  Auxiliary Lemmas

**Lemma A.1.** *Suppose that $\|f_n - f_\tau^*\|_\infty \le c$ for a small enough constant $c$ we have*

$$\Delta^2(f_n, f) \le C\left[\mathbb{E}\left(\ell_h(Y - f(X)) - \ell_h(Y - f_n(X))\right) + (\|f_n - f_\tau^*\|_\infty + h^2)\Delta(f, f_n)\sqrt{F}\right]. \tag{16}$$

*and*

$$\|f - f_n\|_{\ell_2}^2 \le C\max\{1, F\}\left[\mathbb{E}\left(\ell_h(Y - f(X)) - \ell_h(Y - f_n(X))\right) + (\|f_n - f_\tau^*\|_\infty + h^2)\|f - f_n\|_{\ell_2}\sqrt{F}\right], \tag{17}$$

*for any $f \in \tilde{\mathcal{I}}(L, W, S, B)$ and for some constant $C > 0$.*

*Proof.* By Knight identity (Knight, 1998),

$\rho_\tau(Y - f(X) + s) - \rho_\tau(Y - f_n(X) + s)$

$= -(f(X) - f_n(X))(\tau - \mathbf{1}\{Y + s \le f_n(X)\}) + \int_0^{f(X) - f_n(X)} [\mathbf{1}\{Y + s \le f_n(X) + z\} - \mathbf{1}\{Y + s \le f_n(X)\}]\,\mathrm{d}z,$

$= -(f(X) - f_n(X))(\tau - \mathbf{1}\{Y + s \le f_\tau^*(X)\}) - (f(X) - f_n(X))(\mathbf{1}\{Y + s \le f_\tau^*(X)\} - \mathbf{1}\{Y + s \le f_n(X)\})$

$+ \int_0^{f(X) - f_n(X)} [\mathbf{1}\{Y + s \le f_n(X) + z\} - \mathbf{1}\{Y + s \le f_n(X)\}]\,\mathrm{d}z. \tag{18}$

By (5) in Assumption 2 and mean value expansion, applying Fubini's theorem and the fact $\int sK_h(s)\mathrm{d}s = 0$, $\int s^2 K_h(s)\mathrm{d}s = \sigma_K^2 h^2$, we have for some constant $\underline{c}, c_\tau > 0$,

$$\mathbb{E}\left[\int K_h(s)\int_0^{f(X) - f_n(X)} \mathbf{1}\{Y + s \le f_n(X) + z\} - \mathbf{1}\{Y + s \le f_n(X)\}\,\mathrm{d}z\mathrm{d}s\bigg| X\right],$$

$$= \mathbb{E}\left[\int K_h(s)\int_0^{f(X) - f_n(X)} F_{Y|X}(f_n(X) + z - s) - F_{Y|X}(f_n(X) - s)\mathrm{d}z\mathrm{d}s\right],$$

$$\ge \mathbb{E}\left[\underline{p}\int_0^{f(X) - f_n(X)} \min\{z, \kappa\}\mathrm{d}z\right] - \underline{c}h^2\mathbb{E}|f(X) - f_n(X)|,$$

$$\ge c_\tau\left(\mathbb{E}\left[D^2(f(X) - f_n(X))\right] - h^2\mathbb{E}|f(X) - f_n(X)|\right), \tag{19}$$

Combining (40), (18), and (19), by Fubini's theorem, we have

$$
\begin{aligned}
&\mathbb{E}\left\{\ell_h(Y - f(X)) - \ell_h(Y - f_n(X))\right\} \\
&= \int K_h(s)\mathbb{E}\left\{\rho_\tau(Y + s - f(X)) - \rho_\tau(Y + s - f_n(X))\right\} \mathrm{d}s, \\
&\geq -C\mathbb{E}\left\{|f(X) - f_n(X)| \cdot (|f_\tau^*(X) - f_n(X)| + h^2)\right\} + c_\tau\left(\mathbb{E}\left[D^2(f(X) - f_n(X))\right]\right), \\
&\geq -C(\|f_\tau^* - f_n\|_\infty + h^2)\sqrt{F\Delta^2(f, f_n)} + c_\tau\Delta^2(f, f_n),
\end{aligned}
\tag{20}
$$

which yields (16). Furthermore, note that $\frac{\|f - f_n\|_{\ell^2}^2}{\max\{F,1\}} \leq \Delta^2(f, f_n) \leq \|f - f_n\|_{\ell^2}^2$, then (17) holds. $\qquad\square$

**Lemma A.2.** *Suppose that $f_n \in \tilde{\mathcal{I}}(L, W, S, B)$ and $\|f_n - f_\tau^*\|_\infty \leq c$ for a sufficiently small constant $c > 0$. The estimator $\hat{f}_h$ defined in (4) satisfies for some constant $C > 0$,*

$$
\Delta^2\left(\hat{f}_h, f_n\right) \leq C\left[M_n(\hat{f}_h) - \hat{M}_n(\hat{f}_h) + (\|f_n - f_\tau^*\|_\infty + h^2)\Delta\left(\hat{f}_h, f_n\right)\sqrt{F}\right].
\tag{21}
$$

*Furthermore,*

$$
\left\|\hat{f}_h - f_n\right\|_{\ell_2}^2 \leq C\max\{1, F\}\left[M_n(\hat{f}_h) - \hat{M}_n(\hat{f}_h) + (\|f_n - f_\tau^*\|_\infty + h^2)\left\|\hat{f}_h - f_n\right\|_{\ell_2}\sqrt{F}\right].
\tag{22}
$$

*Proof.* Since $\hat{f}_h$ in (4) satisfies $\hat{f}_h \in \tilde{\mathcal{I}}(L, W, S, B)$, then by Lemma A.1,

$$
\begin{aligned}
\Delta^2(\hat{f}_h, f_n) &\leq C\left[\mathbb{E}\left(\ell_h(Y - \hat{f}_h(X)) - \ell_h(Y - f_n(X))\right) + \|f_n - f_\tau^*\|_\infty\Delta\left(\hat{f}_h, f_n\right)\sqrt{F} + h^2\right], \\
&\leq C\left[M_n(\hat{f}_h) - \hat{M}_n(\hat{f}_h) + \|f_n - f_\tau^*\|_\infty\Delta\left(\hat{f}_h, f_n\right)\sqrt{F} + h^2\right],
\end{aligned}
\tag{23}
$$

where the last inequality is obtained by the fact $\hat{M}_n(\hat{f}_h) \leq 0$. Similarly, (22) can also hold by Lemma A.1 and the optimality of $\hat{f}_h$.

$\qquad\square$

**Lemma A.3.** *Suppose that*

$$
3\mathbb{E}\left(\sup_{f \in \tilde{\mathcal{I}}(L, W, S, B), \|f - f_n\|_{\ell_2}^2 \leq r^2} \frac{1}{n}\sum_{i=1}^{n}\xi_i\left(f(\mathbf{x}_i) - f_n(\mathbf{x}_i)\right)^2\right) \leq r^2,
\tag{24}
$$

*for $\{\xi_i\}_{i=1}^n$ Rademacher variables independent of $\{(\mathbf{x}_i, y_i)\}_{i=1}^n$, and*

$$
2F\sqrt{\frac{7\gamma}{3n}} \leq r
\tag{25}
$$

*Then with probability at least $1 - e^{-\gamma}$, $\|f - f_n\|_{\ell_2}^2 \leq r^2$ with $f \in \tilde{\mathcal{I}}(L, W, S, B)$ implies*

$$
\|f - f_n\|_n^2 \leq (2r)^2.
$$

*Proof.* Using the fact $(a - b)^2 \leq 2(a^2 + b^2)$, we have for $f \in \tilde{\mathcal{I}}(L, W, S, B)$

$$
\begin{aligned}
\mathrm{Var}\left[(f(X) - f_n(X))^2\right] &\leq \mathbb{E}(f(X) - f_n(X))^4, \\
&\leq 2(F^2 + \|f_n\|_\infty^2)\mathbb{E}(f(X) - f_n(X))^2, \\
&\leq 4F^2\|f - f_n\|_{\ell_2}^2.
\end{aligned}
\tag{26}
$$

Note that $0 \leq (f(\mathbf{x}) - f_n(\mathbf{x}))^2 \leq 4F^2$, then by Theorem 2.1 in Bartlett et al. (2005), for every $\gamma > 0$, with probability at least $1 - e^{-\gamma}$,

$$\sup_{f \in \tilde{\mathcal{I}}(L,W,S,B), \|f-f_n\|_{\ell_2}^2 \leq r^2} \left\{ \|f - f_n\|_n^2 - \|f - f_n\|_{\ell_2}^2 \right\}$$

$$\leq 3\mathbb{E}\left( \sup_{f \in \tilde{\mathcal{I}}(L,W,S,B), \|f-f_n\|_{\ell_2}^2 \leq r^2} \frac{1}{n} \sum_{i=1}^n \xi_i \left( f(X_i) - f_n(X_i) \right)^2 \right) + r \cdot 2F\sqrt{\frac{2\gamma}{n}} + \frac{28F^2\gamma}{3n},$$

$$\leq 3r^2,$$

where the last inequality is obtained by (24) and (25). Then with probability at least $1 - e^{-\gamma}$, $\|f - f_n\|_{\ell_2}^2 \leq r^2$ with $f \in \tilde{\mathcal{I}}(L,W,S,B)$ implies $\|f - f_n\|_n^2 \leq 4r^2$.

$\square$

**Lemma A.4.** *Suppose that $h^2 \lesssim n^{-\frac{s}{2s+d}}$ and $\left\| \hat{f}_h - f_n \right\|_{\ell_2} \leq r_0$, with $r_0$ satisfying (24), (25), and Assumption 4 holds. Also, with the notation of Assumption 3, suppose that for the class $\mathcal{I}(L,W,S,B)$ the parameters are chosen as*

$$L = 3 + 2\left\lceil \log_2 \left( \frac{3^{\max\{d,m\}}}{\epsilon c_{d,m}} \right) + 5 \right\rceil \lceil \log_2 \max\{d,m\} \rceil, \quad W = W_0 N \tag{27}$$

$$S = (L-1)W_0^2 N + N, \quad B = O\left( N^{\left(v^{-1}+d^{-1}\right)(\max\{1,(d/p-s)_+\})} \right) \tag{28}$$

*for a constant $c_{d,m}$ that depends on $d$ and $m$, a constant $W_0$, and where $v = (s - \delta)/\delta$,*

$$\delta = \frac{d}{p}, \quad N \asymp n^{\frac{d}{2s+d}}. \tag{29}$$

*Then there exists a universal constant $C_0 > 0$ such that*

$$\left\| \hat{f}_h - f_n \right\|_{\ell_2}^2 \leq C_0 \left[ r_0 F^{5/2} \sqrt{\frac{\gamma}{n}} + \frac{F^{5/2}\gamma}{n} + \right.$$

$$\left. r_0 F \sqrt{\frac{N(\log N)^2}{n}} + r_0 F \sqrt{\frac{N\left[ (\log N)^2 + \log r_0^{-1} + \log n \right]}{n}} + N^{-s/d} r_0 F^{3/2} \right]$$

*with probability at least $1 - 3e^{-\gamma}$, where $N \asymp n^{\frac{d}{2s+d}}$.*

*Proof.* Let

$$\mathcal{G} = \left\{ g : g(\mathbf{x},y) = \ell_h(y - f(\mathbf{x})) - \ell_h(y - f_n(\mathbf{x})), \quad f \in \tilde{\mathcal{I}}(L,W,S,B), \|f - f_n\|_{\ell_2} \leq r_0 \right\}.$$

Then for $\xi_1, \ldots, \xi_n$ independent Rademacher variables independent of $\{(\mathbf{x}_i, y_i)\}_{i=1}^n$, by Theorem 2.1 in Bartlett et al. (2005), with probability at least $1 - 2e^{-\gamma}$, we have that

$$M_n(\hat{f}_h) - \hat{M}_n(\hat{f}_h) + (\|f_n - f_\tau^*\|_\infty + h^2) \left\| \hat{f}_h - f_n \right\|_{\ell_2} \sqrt{F}$$

$$\lesssim \mathbb{E}\left( \sup_{g \in \mathcal{G}} \left| \frac{1}{n} \sum_{i=1}^n \xi_i g(\mathbf{x}_i, y_i) \right| (\mathbf{x}_1, y_1), \ldots, (\mathbf{x}_n, y_n) \right)$$

$$+ 4r_0 F^{3/2} \sqrt{\frac{\gamma}{n}} + \frac{100F^{3/2}\gamma}{3n} + (\|f_n - f_\tau^*\|_\infty + h^2) \left\| \hat{f}_h - f_n \right\|_{\ell_2} F^{1/2}. \tag{30}$$

Denote $\mathbb{E}_\xi$ as the expectation with respect to $\xi_1, \ldots, \xi_n$. Let

$$\varphi_{f,i}(t_i) = \ell_h(y_i - (t_i + f_n(\mathbf{x}_i))) - \ell_h(y_i - f_n(\mathbf{x}_i)),$$

where $t_i = f(\mathbf{x}_i) - f_n(\mathbf{x}_i)$. Note that $\ell_h(\cdot)$ is 1-Lipschitz continuous and $\varphi_{f,i}(0) = 0$, by Talagrand's inequality (Ledoux & Talagrand (2013)), Lemma A.3, with probability at least $1 - e^{-\gamma}$, we have

$$
\mathbb{E}_\xi \left( \sup_{g \in \mathcal{G}} \frac{1}{n} \sum_{i=1}^n \xi_i g(\mathbf{x}_i, y_i) \right) = \mathbb{E}_\xi \left( \sup_{f \in \mathcal{I}(L,W,S,B), \|f\|_\infty \leq F, \|f-f_n\|_{\ell_2} \leq r_0} \frac{1}{n} \sum_{i=1}^n \xi_i \varphi_{f,i}(t_i) \right),
$$

$$
\leq \mathbb{E}_\xi \left( \sup_{f \in \mathcal{I}(L,W,S,B), \|f\|_\infty \leq F, \|f-f_n\|_{\ell_2} \leq r_0} \frac{1}{n} \sum_{i=1}^n \xi_i \left( f(\mathbf{x}_i) - f_n(\mathbf{x}_i) \right) \right),
$$

$$
\leq \mathbb{E}_\xi \left( \sup_{f \in \mathcal{I}(L,W,S,B), \|f\|_\infty \leq F, \|f_n-f\|_n \leq 2r_0} \frac{1}{n} \sum_{i=1}^n \xi_i \left( f(\mathbf{x}_i) - f_n(\mathbf{x}_i) \right) \right). \tag{31}
$$

By Dudley's chaining inequality and arguments in the proof of Theorem 2 in Suzuki (2019), we further have for some constant $C > 0$,

$$
\mathbb{E}_\xi \left( \sup_{f \in \mathcal{I}(L,W,S,B), \|f\|_\infty \leq F, \|f_n-f\|_n \leq 2r_0} \frac{1}{n} \sum_{i=1}^n \xi_i \left( f(\mathbf{x}_i) - f_n(\mathbf{x}_i) \right) \right)
$$

$$
\leq \inf_{0 < \alpha < r_0} \left\{ 4\alpha + \frac{24 r_0}{\sqrt{n}} \sqrt{\log \mathcal{N} \left( \alpha, \tilde{\mathcal{I}}(L,W,S,B), \|\cdot\|_\infty \right)} \right\}
$$

$$
\leq C \inf_{0 < \alpha < r_0} \left\{ \alpha + r_0 \sqrt{\frac{N \log(N) \left[ \log^2 N + \log \alpha^{-1} \right]}{n}} \right\} \tag{32}
$$

By Proposition 1 in Suzuki (2019) and $N \asymp n^{\frac{d}{2s+d}}, h^2 \lesssim n^{-\frac{s}{2s+d}}$, we have $\|f_n - f_\tau^*\|_\infty + h^2 \lesssim N^{-s/d}$. Let

$$
\alpha = r_0 \sqrt{\frac{N(\log N)^2}{n}},
$$

together with Lemma A.2, (32), (30), and (31), we have for some constant $C > 0$,

$$
\left\| \hat{f}_h - f_\tau^* \right\|_{\ell_2}^2 \leq C \left[ 4 r_0 F^{5/2} \sqrt{\frac{\gamma}{n}} + \frac{100 F^{5/2} \gamma}{3n} + \right.
$$

$$
\left. r_0 F \sqrt{\frac{N(\log N)^2}{n}} + r_0 F \sqrt{\frac{N \log(N) \left[ (\log N)^2 + \log r_0^{-1} + \log n \right]}{n}} + r_0 N^{-s/d} F^{3/2} \right]
$$

with probability at least $1 - 3 e^{-\gamma}$. $\qquad\qquad\square$

**Lemma A.5** (Lemma 20 in Padilla et al. (2022)). *Let $r^*$ be defined as*

$$
r^* = \inf \left\{ r > 0 : 3\mathbb{E} \left( \sup_{f \in \tilde{\mathcal{I}}(L,W,S,B), \|f-f_n\|_{\ell_2} \leq s} \frac{1}{n} \sum_{i=1}^n \xi_i \left( f(x_i) - f_n(x_i) \right)^2 \right) < s^2, \forall s \geq r \right\},
$$

*for $\{\xi_i\}_{i=1}^n$ Rademacher variables independent of $\{(x_i, y_i)\}_{i=1}^n$. Then under the conditions (27), (28), and (29) of Lemma A.4,*

$$
r^* \leq \tilde{C} \left[ \sqrt{\frac{N(\log N)^2}{n}} + \sqrt{\frac{N \log(N) \left[ (\log N)^2 + \log n \right]}{n}} \right], \tag{33}
$$

*for a constant $\tilde{C} > 0$ and with $N$ satisfying $N \asymp n^{\frac{d}{2s+d}}$.*

Let $\mathcal{H}$ be a class of functions from $\mathcal{X}$ to $\mathbb{R}$. We define the pseudodimension of $\mathcal{H}$, denoted as $\mathrm{Pdim}(\mathcal{H})$, as the largest integer $m$ for which there exist $(a_1, b_1) \ldots, (a_m, b_m) \in \mathcal{X} \times \mathbb{R}$ such that for all $\eta \in \{0,1\}^m$ there exists $f \in \mathcal{H}$ such that

$$
f(a_i) > b_i \iff \eta_i,
$$

for $i = 1, \ldots, m$.

**Lemma A.6** (Theorem 12 from Padilla et al. (2022))**.** *With the notation from before, for the neural network function class* $\mathcal{F}(P, U, L)$*, we have*

$$\mathrm{Pdim}(\mathcal{F}(P, U, L)) = O(LP \log(U)).$$

## A.2 Proof of Theorem 3.1

*Proof.* Write the ball centered in $f$ of radius $r$

$$\mathrm{B}(f, \|\cdot\|_{\ell_2}, r) = \{g : \|f - g\|_{\ell_2} \le r\}.$$

We divide the space $\tilde{\mathcal{I}}(L, W, S, B) = \{f : f \in \mathcal{I}(L, W, S, B), \|f\|_\infty \le F\}$ into sets of increasing radius

$$\mathrm{B}(f_n, \|\cdot\|_{\ell_2}, \bar{r}), \mathrm{B}(f_n, \|\cdot\|_{\ell_2}, 2\bar{r}) \setminus \mathrm{B}(f_n, \|\cdot\|_{\ell_2}, \bar{r}), \dots, \mathrm{B}(f_n, \|\cdot\|_{\ell_2}, 2^l\bar{r}) \setminus \mathrm{B}(f_n, \|\cdot\|_{\ell_2}, 2^{l-1}\bar{r}),$$

where

$$l = \left\lfloor \log_2 \left( \frac{2F}{\sqrt{\log n/n}} \right) \right\rfloor.$$

If for some $j \le l$,

$$\hat{f}_h \in \mathrm{B}(f_n, \|\cdot\|_{\ell_2}, 2^j\bar{r}),$$

then by Lemma A.4, with probability at least $1 - 3e^{-\gamma}$, we have for some constant $\tilde{C} > 0$,

$$\left\| \hat{f}_h - f_n \right\|_{\ell_2}^2 \le \tilde{C} \left( 2^j \bar{r} F^{5/2} \sqrt{\frac{\gamma}{n}} + \frac{F^{5/2}\gamma}{n} + \right.$$
$$\left. 2^j \bar{r} F \sqrt{\frac{N(\log N)^2}{n}} + 2^j \bar{r} F \sqrt{\frac{N \log(N) \left[ (\log N)^2 + 2 \log n \right]}{n}} + N^{-s/d} 2^j \bar{r} F^{3/2} \right). \tag{34}$$

Recall the $r^*$ defined in Lemma A.5. We set

$$\bar{r} = 8\tilde{C} \left[ F^{5/2} \sqrt{\frac{\gamma}{n}} + F \sqrt{\frac{N(\log N)^2}{n}} + F \sqrt{\frac{N \log(N)[(\log N)^2 + 2 \log n]}{n}} + N^{-s/d} F^{3/2} \right]$$
$$+ 2\sqrt{2\tilde{C}} \cdot \sqrt{\frac{F^{5/2}\gamma}{n}} + r^*. \tag{35}$$

By Lemma A.5 and $N \asymp n^{d/(2s+d)}$, setting $\gamma \asymp \log n$, it follows that $\tilde{\mathcal{I}}(L, W, S, B) \subset \mathrm{B}(f_n, \|\cdot\|_{\ell_2}, 2^l\bar{r})$ when $n$ is sufficiently large. Therefore, $\hat{f}_h \in \mathrm{B}(f_n, \|\cdot\|_{\ell_2}, 2^l\bar{r})$ with probability 1 if $n$ is sufficiently large.

Elementary calculation yields $\bar{r} > r^*$ and for all $0 \le j \le l$,

$$\frac{2^j \bar{r}}{8} \ge \tilde{C} \left[ F^{5/2} \sqrt{\frac{\gamma}{n}} + F \sqrt{\frac{N(\log N)^2}{n}} + F \sqrt{\frac{N \log(N)[(\log N)^2 + 2 \log n]}{n}} + N^{-s/d} F^{3/2} \right], \tag{36}$$

and

$$\frac{2^{2j} \bar{r}^2}{8} \ge \tilde{C} \left( \frac{F^{5/2}\gamma}{n} \right). \tag{37}$$

Combining (36), (37), and (34), we have with probability at least $1 - 3e^{-\gamma}$,

$$\|\hat{f}_h - f_n\|_{\ell_2} \le 2^{j-1}\bar{r}. \tag{38}$$

Now we begin from the first step of our localization procedure. Note that $\bar{r} > r^*$, by Lemmas A.3 and A.4, with probability at least $1 - e^{-\gamma}$, we have that

$$\left\| \hat{f}_h - f_n \right\|_{\ell_2} \le 2^l \bar{r} \quad \text{implies} \quad \left\| \hat{f}_h - f_n \right\|_n \le 2^{l+1}\bar{r}.$$

Then by arguments above, with probability at least $1 - 4e^{-\gamma}$, $\left\|\hat{f}_h - f_n\right\|_{\ell_2} \leq 2^l \bar{r}$ implies that

$$\left\|\hat{f}_h - f_n\right\|_{\ell_2} \leq 2^{l-1}\bar{r}, \quad \text{and} \quad \left\|\hat{f}_h - f_n\right\|_n \leq 2^l \bar{r}.$$

Continue recursively, we arrive at

$$\left\|\hat{f}_h - f_n\right\|_{\ell_2} \leq \bar{r}, \quad \text{and} \quad \left\|\hat{f}_h - f_n\right\|_n \leq 2\bar{r},$$

with probability approaching one.

Specifically, this procedure can be formulated as

$$
\begin{aligned}
\mathrm{P}(\hat{f}_h \in \mathrm{B}(f_n, \|\cdot\|_{\ell_2}, \bar{r})) &= \mathrm{P}(\hat{f}_h \in \mathrm{B}(f_n, \|\cdot\|_{\ell_2}, 2\bar{r})) - \mathrm{P}(\hat{f}_h \in \mathrm{B}(f_n, \|\cdot\|_{\ell_2}, 2\bar{r}) \setminus \mathrm{B}(f_n, \|\cdot\|_{\ell_2}, \bar{r})), \\
&\geq \mathrm{P}(\hat{f}_h \in \mathrm{B}(f_n, \|\cdot\|_{\ell_2}, 2\bar{r})) - 4e^{-\gamma}, \\
&\geq \cdots, \\
&\geq 1 - 4(l+1)e^{-\gamma} = 1 - o(1),
\end{aligned}
\tag{39}
$$

with setting $\gamma \asymp \log n$.

By Lemma A.5 and (35), we have

$$
\begin{aligned}
\bar{r} \leq &8\tilde{C}\left[F^{5/2}\sqrt{\frac{\gamma}{n}} + F\sqrt{\frac{N(\log N)^2}{n}} + F\sqrt{\frac{N\log(N)\left[(\log N)^2 + 2\log n\right]}{n}} + N^{-s/d}F^{3/2}\right] \\
&+ 2\sqrt{2} \cdot \sqrt{\frac{\tilde{C}F^{5/2}\gamma}{n}} + \tilde{C}\left[\sqrt{\frac{N(\log N)^2}{n}} + \sqrt{\frac{N\log(N)\left[(\log N)^2 + \log n\right]}{n}}\right].
\end{aligned}
$$

Then the claim follows by $N \asymp n^{d/(2s+d)}$ and $\|f_n - f_\tau^*\|_\infty \lesssim N^{-s/d}$ in Proposition 1 of Suzuki (2019). □

## A.3 Proof of Theorem 3.2

*Proof.* Throughout this proof, the covariates $\mathbf{x}_1, \ldots, \mathbf{x}_n$ are fixed. For simplicity, we write $f \in \mathcal{F}$ instead of $f \in \mathcal{F}(P, U, L)$ within the proof. Let $\hat{\delta}_i = \hat{f}_h(\mathbf{x}_i) - f_\tau^*(\mathbf{x}_i)$ for $i = 1, \ldots, n$. Note that $\ell_h(u) = \int_{-\infty}^{\infty} \rho_\tau(v)K_h(v-u)\mathrm{d}v = \int_{-\infty}^{\infty} \rho_\tau(u+s)K_h(s)\mathrm{d}s$, then for any $x, y \in \mathbb{R}$,

$$\ell_h(x) - \ell_h(y) = \int_{-\infty}^{\infty} K_h(s)\left(\rho_\tau(x+s) - \rho_\tau(y+s)\right)\mathrm{d}s.\tag{40}$$

By Knight identity (Knight, 1998), for any $\delta \in \mathbb{R}$,

$$
\begin{aligned}
&\rho_\tau(y_i - (f_\tau^*(\mathbf{x}_i) + \hat{\delta}_i) + s) - \rho_\tau(y_i - f_\tau^*(\mathbf{x}_i) + s) \\
&= -\hat{\delta}_i\left(\tau - \mathbf{1}\left\{y_i + s \leq f_\tau^*(\mathbf{x}_i)\right\}\right) + \int_0^{\hat{\delta}_i}\left(\mathbf{1}\{y_i \leq f_\tau^*(\mathbf{x}_i) + z - s\} - \mathbf{1}\{y_i \leq f_\tau^*(\mathbf{x}_i) - s\}\right)\mathrm{d}z.
\end{aligned}
\tag{41}
$$

By Assumption 1, Fubini's theorem and mean value expansion, using the fact $\int sK_h(s)\mathrm{d}s = 0$, $\int s^2 K_h(s)\mathrm{d}s = \sigma_K^2 h^2$, we have

$$
\begin{aligned}
&\mathbb{E}\left[\frac{1}{n}\sum_{i=1}^n \int K_h(s)\int_0^{\hat{\delta}_i}\left(\mathbf{1}\{y_i \leq f_\tau^*(\mathbf{x}_i) + z - s\} - \mathbf{1}\{y_i \leq f_\tau^*(\mathbf{x}_i) - s\}\right)\mathrm{d}z\mathrm{d}s \,\Big|\, \mathbf{x}_1, \ldots, \mathbf{x}_n\right] \\
&= \mathbb{E}\left[\frac{1}{n}\sum_{i=1}^n \int K_h(s)\int_0^{\hat{\delta}_i} F_{y_i|\mathbf{x}_i}(f_\tau^*(\mathbf{x}_i) + z - s) - F_{y_i|\mathbf{x}_i}(f_\tau^*(\mathbf{x}_i) - s)\mathrm{d}z\mathrm{d}s \,\Big|\, \mathbf{x}_1, \ldots, \mathbf{x}_n\right], \\
&\geq \mathbb{E}\left[\frac{1}{n}\sum_{i=1}^n\left(\underline{p}\int_0^{|\hat{\delta}_i|}\min\{z, \kappa\}\mathrm{d}z - \underline{c}|\hat{\delta}_i|h^2\right)\,\Big|\, \mathbf{x}_1, \ldots, \mathbf{x}_n\right],
\end{aligned}
\tag{42}
$$

where the constants $\underline{p}, \underline{c} > 0$ are uniform for all $i = 1, \ldots, n$.

Similarly, we also have

$$\mathbb{E}\left[\frac{1}{n}\sum_{i=1}^{n} -\hat{\delta}_i \int K_h(s)(\tau - \mathbf{1}\{y_i + s \leq f_\tau^*(\mathbf{x}_i)\})\mathrm{d}s\right]$$

$$= \mathbb{E}\left[\frac{1}{n}\sum_{i=1}^{n} -\hat{\delta}_i \int K_h(s)(F_{y_i|\mathbf{x}_i}(f_\tau^*(\mathbf{x}_i)) - F_{y_i|\mathbf{x}_i}(f_\tau^*(\mathbf{x}_i) - s))\mathrm{d}s\right]$$

$$\geq -\underline{c}\mathbb{E}\left(\frac{1}{n}\sum_{i=1}^{n}|\hat{\delta}_i|h^2\right). \tag{43}$$

Combining (40), (41), (42), and (43), for $D_h(t) = \min\{|t|, t^2\} - h^2|t|$, we have

$$\mathbb{E}\left[\frac{1}{n}\sum_{i=1}^{n} D_h^2\left\{f_\tau^*(\mathbf{x}_i) - \hat{f}_h(\mathbf{x}_i)\right\}\right]$$

$$\leq \frac{1}{c_\tau n}\mathbb{E}\left(\sum_{i=1}^{n}\mathbb{E}\left[\ell_h\left\{y_i - f_\tau^*(\mathbf{x}_i) - \hat{\delta}_i\right\}\right] - \sum_{i=1}^{n}\mathbb{E}\left[\ell_h\left\{y_i - f_\tau^*(\mathbf{x}_i)\right\}\right]\right),$$

$$= \frac{1}{c_\tau}\mathbb{E}\left\{M_n(\hat{f}_h)\right\} + O(err_1). \tag{44}$$

Next, we only need to bound $E\left\{M_n(\hat{f}_h)\right\}$. By symmetrization Lemma 10 in **?**, and Talagrand's inequality (Ledoux & Talagrand, 2013) using the fact $\ell_h(\cdot)$ is 1-Lipschitz continuous, for i.i.d. Rademacher variables $\xi_i, i = 1, \ldots, n$,

$$\mathbb{E}\left\{M_n(\hat{f}_h)\right\} \leq \mathbb{E}\left\{\sup_{f\in\mathcal{F}, \|f\|_\infty \leq F}\left[M_n(f) - \hat{M}_n(f)\right]\right\},$$

$$\leq 2\mathbb{E}\left\{\sup_{f\in\mathcal{F}, \|f\|_\infty \leq F}\sum_{i=1}^{n}\xi_i\hat{M}_{n,i}(f)\right\},$$

$$\leq 2\mathbb{E}\left\{\sup_{f\in\mathcal{F}, \|f\|_\infty \leq 2F}\frac{1}{n}\sum_{i=1}^{n}\xi_i f(\mathbf{x}_i)\right\}. \tag{45}$$

By Dudley's theorem and Lemma 4 in Farrell et al. (2021), we further have

$$F\mathbb{E}\left\{\sup_{f\in\mathcal{F}, \|f\|_\infty \leq F}\frac{1}{n}\sum_{i=1}^{n}\xi_i\frac{f(\mathbf{x}_i)}{F}\right\} \leq \frac{CF}{\sqrt{n}}\int_0^2 \sqrt{\log\mathcal{N}\left(\mu, \mathcal{F}/F, \|\cdot\|_n\right)}\mathrm{d}\mu,$$

$$\leq \frac{CF}{\sqrt{n}}\int_0^2\sqrt{\log\left(\left(\frac{2\cdot e\cdot n}{\mu\cdot\mathrm{Pdim}(\mathcal{F})}\right)^{\mathrm{Pdim}(\mathcal{F})}\right)}\mathrm{d}\mu. \tag{46}$$

Combining Lemma A.6, (45) and (46), for some constant $\tilde{C} > 0$, we have

$$\mathbb{E}\left\{M_n(\hat{f}_h)\right\} \leq \tilde{C}F\sqrt{\frac{LP\log U \cdot \log n}{n}}, \tag{47}$$

which shows (14) combining with (44).

For (15), when $h^2 = o(\sqrt{\mathbb{E}\|\hat{f}_h - f_\tau^*\|_n^2})$, by $\sum_i^n |a_i|/n \leq \sqrt{\sum_{i=1}^n a_i^2/n}$ and Jensen's inequality,

$$h^2\mathbb{E}\|\hat{f}_h - f_\tau^*\|_{n,1} \leq h^2\mathbb{E}\sqrt{\|\hat{f}_h - f_\tau^*\|_n^2} \leq h^2\sqrt{\mathbb{E}\|\hat{f}_h - f_\tau^*\|_n^2} = o(\mathbb{E}\|\hat{f}_h - f_\tau^*\|_n^2). \tag{48}$$

Then combining with the fact $\|\hat{f}_h - f_\tau^*\|_n^2 \leq \max\{1, F\}\Delta_n^2(\hat{f}_h, f_\tau^*)$, (15) can also hold. $\qquad\square$

### A.4 Output-layer optimization comparison

Conditional on the fixed hidden-layer parameters $\{A^{(l)}, b^{(l)}\}_{l=1}^{L-1}$, write

$$\vartheta = (A^\top, b)^\top, \qquad \widetilde{\mathbf{z}}_i = \left((\mathbf{z}_i^{(L-1)})^\top, 1\right)^\top,$$

so that $A\mathbf{z}_i^{(L-1)} + b = \vartheta^\top \widetilde{\mathbf{z}}_i$. Write $\mathcal{Q}_{n,h}(\vartheta) = \mathcal{Q}_{n,h}(A, b)$ and

$$D_L = \sup_{\vartheta, \vartheta' \in \Theta_L} \|\vartheta - \vartheta'\|_2, \qquad G_L = \max\{\tau, 1 - \tau\} \max_{1 \le i \le n} \|\widetilde{\mathbf{z}}_i\|_2.$$

For $h > 0$, also define

$$L_{n,h}^{\text{out}} = \frac{\|K\|_\infty}{h} \lambda_{\max} \left( \frac{1}{n} \sum_{i=1}^n \widetilde{\mathbf{z}}_i \widetilde{\mathbf{z}}_i^\top \right).$$

Let $\Pi_{\Theta_L}$ denote the Euclidean projection onto $\Theta_L$. If $D_L = 0$, the optimization gap is identically zero; hence, below we assume $D_L > 0$. For any $h \ge 0$, starting from $\vartheta_1^{(h,\mathrm{s})} \in \Theta_L$, consider the projected subgradient iterations

$$\vartheta_{t+1}^{(h,\mathrm{s})} = \Pi_{\Theta_L} \left( \vartheta_t^{(h,\mathrm{s})} - \frac{D_L}{G_L \sqrt{T}} \mathbf{g}_t^{(h)} \right), \qquad \mathbf{g}_t^{(h)} \in \partial \mathcal{Q}_{n,h}(\vartheta_t^{(h,\mathrm{s})}),$$

for $t = 1, \ldots, T$, and define

$$\overline{\vartheta}_T^{(h,\mathrm{s})} = \frac{1}{T} \sum_{t=1}^T \vartheta_t^{(h,\mathrm{s})}.$$

For $h > 0$, we also consider projected gradient descent

$$\vartheta_{t+1}^{(h,\mathrm{g})} = \Pi_{\Theta_L} \left( \vartheta_t^{(h,\mathrm{g})} - \frac{1}{L_{n,h}^{\text{out}}} \nabla \mathcal{Q}_{n,h}(\vartheta_t^{(h,\mathrm{g})}) \right), \qquad t = 1, \ldots, T.$$

Set $\vartheta_T^{(0)} = \overline{\vartheta}_T^{(0,\mathrm{s})}$. For $h > 0$, select the procedure having the smaller theoretical upper bound

$$\vartheta_T^{(h)} = \begin{cases} \overline{\vartheta}_T^{(h,\mathrm{s})}, & \dfrac{D_L G_L}{\sqrt{T}} \le \dfrac{L_{n,h}^{\text{out}} D_L^2}{2T}, \\[3mm] \vartheta_{T+1}^{(h,\mathrm{g})}, & \dfrac{D_L G_L}{\sqrt{T}} > \dfrac{L_{n,h}^{\text{out}} D_L^2}{2T}. \end{cases}$$

The choice is made before optimization, so only one procedure is run and the total number of projected first-order iterations is at most $T$. Finally, write $\vartheta_T^{(h)} = \left((A_T^{(h)})^\top, b_T^{(h)}\right)^\top$.

**Proposition A.7** (Finite-step output-layer optimization)**.** *Suppose that $\Theta_L$ is nonempty, compact, and convex. Under the above construction, for every $T \ge 1$ and $h > 0$,*

$$\eta_T(0) = O(T^{-1/2}), \qquad \eta_T(h) = O\left(\min\left\{T^{-1/2}, h^{-1}T^{-1}\right\}\right).$$

*Proof.* Conditional on the hidden-layer parameters and the observed data, $\mathcal{Q}_{n,h}$ is a deterministic function of $\vartheta$. Since $\rho_\tau$ is convex and $K$ is a nonnegative density, $\ell_h$ and hence $\mathcal{Q}_{n,h}$ are convex. Moreover, the minimum of $\mathcal{Q}_{n,h}$ over $\Theta_L$ exists because $\Theta_L$ is compact. We first establish the required subgradient and smoothness bounds. For $h > 0$, a change of variables gives $\ell_h(u) = \int \rho_\tau(u + hv)K(v)\,\mathrm{d}v.$. Differentiating under the integral yields $\ell_h'(u) = \tau - \int_{-\infty}^{-u/h} K(v)\,\mathrm{d}v$. It follows that

$$|\ell_h'(u)| \le \max\{\tau, 1 - \tau\}, \qquad |\ell_h'(u) - \ell_h'(v)| \le \frac{\|K\|_\infty}{h}|u - v|. \tag{49}$$

For $h = 0$, every subgradient of $\rho_\tau$ belongs to $[\tau - 1, \tau]$. Therefore, for every $h \geq 0$ and $\mathbf{g} \in \partial \mathcal{Q}_{n,h}(\vartheta)$,

$$\|\mathbf{g}\|_2 \leq \max\{\tau, 1 - \tau\} \frac{1}{n} \sum_{i=1}^{n} \|\widetilde{\mathbf{z}}_i\|_2 \leq G_L. \tag{50}$$

For $h > 0$, the gradient of the empirical objective is

$$\nabla \mathcal{Q}_{n,h}(\vartheta) = -\frac{1}{n} \sum_{i=1}^{n} \ell'_h \left( y_i - \vartheta^\top \widetilde{\mathbf{z}}_i \right) \widetilde{\mathbf{z}}_i.$$

Let $\widehat{\boldsymbol{\Sigma}}_L = \frac{1}{n} \sum_{i=1}^{n} \widetilde{\mathbf{z}}_i \widetilde{\mathbf{z}}_i^\top$. For $\vartheta, \vartheta' \in \Theta_L$ and any unit vector $\mathbf{v}$, (49) and the Cauchy–Schwarz inequality imply

$$\begin{aligned}
&\left| \mathbf{v}^\top \left\{ \nabla \mathcal{Q}_{n,h}(\vartheta) - \nabla \mathcal{Q}_{n,h}(\vartheta') \right\} \right| \\
&\leq \frac{\|K\|_\infty}{h} \frac{1}{n} \sum_{i=1}^{n} \left| \widetilde{\mathbf{z}}_i^\top (\vartheta - \vartheta') \right| \left| \widetilde{\mathbf{z}}_i^\top \mathbf{v} \right| \\
&\leq \frac{\|K\|_\infty}{h} \left\{ (\vartheta - \vartheta')^\top \widehat{\boldsymbol{\Sigma}}_L (\vartheta - \vartheta') \right\}^{1/2} \left( \mathbf{v}^\top \widehat{\boldsymbol{\Sigma}}_L \mathbf{v} \right)^{1/2} \\
&\leq L_{n,h}^{\text{out}} \|\vartheta - \vartheta'\|_2. 
\end{aligned} \tag{51}$$

Taking the supremum over all unit vectors $\mathbf{v}$ shows that

$$\left\| \nabla \mathcal{Q}_{n,h}(\vartheta) - \nabla \mathcal{Q}_{n,h}(\vartheta') \right\|_2 \leq L_{n,h}^{\text{out}} \|\vartheta - \vartheta'\|_2. \tag{52}$$

We next analyze the projected subgradient procedure. Let

$$\vartheta_h^* \in \underset{\vartheta \in \Theta_L}{\arg\min} \, \mathcal{Q}_{n,h}(\vartheta).$$

For any given constant step size $\alpha > 0$ and $h \geq 0$, the nonexpansiveness of the Euclidean projection gives

$$\begin{aligned}
\|\vartheta_{t+1}^{(h,\text{s})} - \vartheta_h^*\|_2^2 &\leq \|\vartheta_t^{(h,\text{s})} - \alpha \mathbf{g}_t^{(h)} - \vartheta_h^*\|_2^2 \\
&\leq \|\vartheta_t^{(h,\text{s})} - \vartheta_h^*\|_2^2 - 2\alpha \left\{ \mathcal{Q}_{n,h}(\vartheta_t^{(h,\text{s})}) - \mathcal{Q}_{n,h}(\vartheta_h^*) \right\} + \alpha^2 G_L^2,
\end{aligned} \tag{53}$$

where the second inequality follows from convexity and (50). Summing over $t = 1, \ldots, T$ and using $\|\vartheta_1^{(h,\text{s})} - \vartheta_h^*\|_2 \leq D_L$ yields

$$\frac{1}{T} \sum_{t=1}^{T} \left\{ \mathcal{Q}_{n,h}(\vartheta_t^{(h,\text{s})}) - \mathcal{Q}_{n,h}(\vartheta_h^*) \right\} \leq \frac{D_L^2}{2\alpha T} + \frac{\alpha G_L^2}{2}.$$

By convexity,

$$\mathcal{Q}_{n,h}(\overline{\vartheta}_T^{(h,\text{s})}) - \mathcal{Q}_{n,h}(\vartheta_h^*) \leq \frac{D_L^2}{2\alpha T} + \frac{\alpha G_L^2}{2}.$$

Taking $\alpha = D_L / (G_L \sqrt{T})$ gives

$$\mathcal{Q}_{n,h}(\overline{\vartheta}_T^{(h,\text{s})}) - \mathcal{Q}_{n,h}(\vartheta_h^*) \leq \frac{D_L G_L}{\sqrt{T}}. \tag{54}$$

It remains to analyze projected gradient descent for $h > 0$. For simplicity, write

$$\mathbf{u}_t = \vartheta_t^{(h,\text{g})}, \qquad L_h = L_{n,h}^{\text{out}}.$$

The optimality condition for the Euclidean projection is

$$\langle \nabla \mathcal{Q}_{n,h}(\mathbf{u}_t) + L_h(\mathbf{u}_{t+1} - \mathbf{u}_t), \vartheta - \mathbf{u}_{t+1} \rangle \geq 0$$

for every $\vartheta \in \Theta_L$. Taking $\vartheta = \vartheta_h^*$ and combining this inequality with the convexity and $L_h$-smoothness of $\mathcal{Q}_{n,h}$ gives

$$\mathcal{Q}_{n,h}(\mathbf{u}_{t+1}) - \mathcal{Q}_{n,h}(\vartheta_h^*) \leq \frac{L_h}{2} \left\{ \|\mathbf{u}_t - \vartheta_h^*\|_2^2 - \|\mathbf{u}_{t+1} - \vartheta_h^*\|_2^2 \right\}. \tag{55}$$

Taking $\vartheta = \mathbf{u}_t$ in the projection optimality condition and using (52) also gives

$$\mathcal{Q}_{n,h}(\mathbf{u}_{t+1}) \leq \mathcal{Q}_{n,h}(\mathbf{u}_t) - \frac{L_h}{2} \|\mathbf{u}_{t+1} - \mathbf{u}_t\|_2^2.$$

Thus, the objective values are nonincreasing. Summing (55) over $t = 1, \ldots, T$ therefore yields

$$T \left\{ \mathcal{Q}_{n,h}(\mathbf{u}_{T+1}) - \mathcal{Q}_{n,h}(\vartheta_h^*) \right\} \leq \frac{L_h}{2} \|\mathbf{u}_1 - \vartheta_h^*\|_2^2 \leq \frac{L_h D_L^2}{2}.$$

Consequently,

$$\mathcal{Q}_{n,h}(\vartheta_{T+1}^{(h,g)}) - \mathcal{Q}_{n,h}(\vartheta_h^*) \leq \frac{L_{n,h}^{\text{out}} D_L^2}{2T}. \tag{56}$$

The bound for $\eta_T(0)$ follows from (54). For $h > 0$, the procedure is selected according to the smaller of (54) and (56), which proves

$$\eta_T(h) \leq \min \left\{ \frac{D_L G_L}{\sqrt{T}}, \frac{L_{n,h}^{\text{out}} D_L^2}{2T} \right\}.$$

Finally, $L_{n,h}^{\text{out}} = O(h^{-1})$ conditional on the fixed hidden-layer features. Solving the two bounds for an optimization gap no greater than $\varepsilon$ gives the stated iteration budgets. □

## B  Additional experiments

We state additional details of the experiment in this section. The whole experiments are implemented in PyTorch (Paszke et al., 2019). We use stochastic gradient descent (SGD) with the Nesterov method of momentum factor 0.9. For each sample size, we keep $1/10$ of the data for validation. We start the learning rate at 0.1, and use scheduler `ReduceLROnPlateau` with factor 0.5 and patience 5 to adjust the learning rate dynamically. We also implement gradient calculation for three different convolution-type smoothed loss functions, see Remark 3.1 in He et al. (2023).

### B.1  Tables

Table 1 and Figures 4–8 show that our ConquerNet obtains better performance and higher training efficiency. The faster training mainly comes from our stopping strategy, that is to stop training if the learning rate is lower than a threshold and the validation loss does not decrease for several consecutive epochs, while the baseline models are trained using all epochs according to the codes provided in Padilla et al. (2022). Therefore, the timing comparison in Figures 4–8 should not be used as isolated evidence that the speedup is solely caused by smoothing. Rather, they should be interpreted together with the accuracy results in Table 1: under the reproduced baseline training protocol, the baseline quantile neural network used a larger computation budget but still did not outperform ConquerNet. Therefore, we also train the ConquerNet without the stop criterion and study the MSE results, which are shown in Table 3. We can conclude that our ConquerNet still outperform the baseline models.

Furthermore, we add a controlled timing experiment (Tabel 4), where the same stopping criterion is applied to both the nonsmoothed baseline and ConquerNet. Under this matched stopping protocol, ConquerNet remains faster: across all scenarios, models, sample sizes, quantile levels, and kernels, ConquerNet is faster in 240 out of 270 kernel-setting comparisons. To account for trial-to-trial variation, we compute the combined Monte Carlo standard error of the reported mean difference as $\text{SE}_\Delta = \{\text{SE}_{\text{base}}^2 + \text{SE}_{\text{ConquerNet}}^2\}^{1/2}$, where the standard errors are computed over 50 trials. Under this criterion, 164 reductions exceed $\text{SE}_\Delta$ and 77

Table 3: MSE performances without stop criterion for scenario 1-3, Model A and B under different sample sizes, quantile levels and smoothing kernels. The MSEs are averaged over 50 independent trials.

| | | Method | n=1000 | | | | | n=5000 | | | | | n=10000 | | | | |
|---|---|---|---|---|---|---|---|---|---|---|---|---|---|---|---|---|---|
| | | | τ=0.05 | τ=0.25 | τ=0.5 | τ=0.75 | τ=0.95 | τ=0.05 | τ=0.25 | τ=0.5 | τ=0.75 | τ=0.95 | τ=0.05 | τ=0.25 | τ=0.5 | τ=0.75 | τ=0.95 |
| S1 | Model A | Baseline | 0.3820 | 0.0402 | 0.0278 | 0.0374 | 0.3784 | 0.0996 | 0.0120 | 0.0086 | 0.0108 | 0.0939 | 0.0618 | 0.0067 | 0.0047 | 0.0063 | 0.1366 |
| | | Gaussian | 0.4271 | **0.0379** | **0.0224** | **0.0374** | 0.5027 | 0.1118 | **0.0088** | **0.0053** | **0.0095** | 0.1081 | 0.0669 | **0.0063** | **0.0033** | **0.0054** | **0.0626** |
| | | Uniform | 0.4413 | **0.0378** | **0.0224** | **0.0326** | **0.3704** | 0.1072 | **0.0087** | **0.0059** | **0.0101** | 0.1312 | 0.0789 | **0.0058** | **0.0035** | **0.0055** | **0.0530** |
| | | Epanechnikov | 0.6718 | **0.0390** | **0.0218** | **0.0371** | 0.5913 | 0.1249 | **0.0091** | **0.0055** | **0.0095** | 0.1153 | 0.0648 | **0.0060** | **0.0037** | **0.0053** | **0.0662** |
| | Model B | Baseline | 0.3842 | 0.0527 | 0.0319 | 0.0475 | 0.4222 | 0.1143 | 0.0149 | 0.0107 | 0.0151 | 0.1277 | 0.1691 | 0.0099 | 0.0066 | 0.0082 | 0.3882 |
| | | Gaussian | 0.4158 | 0.0666 | 0.0383 | 0.0632 | 0.5080 | 0.1172 | **0.0143** | **0.0098** | **0.0141** | 0.1067 | 0.1063 | **0.0094** | **0.0055** | 0.0084 | **0.0729** |
| | | Uniform | 0.5645 | 0.0610 | 0.0376 | 0.0616 | 0.5685 | **0.1034** | **0.0143** | **0.0091** | 0.0153 | 0.1255 | **0.0975** | **0.0093** | **0.0055** | **0.0080** | **0.0738** |
| | | Epanechnikov | 0.4247 | 0.0576 | 0.0383 | 0.0623 | 0.6050 | **0.1109** | **0.0144** | **0.0092** | **0.0146** | 0.1164 | 0.1291 | **0.0089** | **0.0058** | 0.0092 | **0.0662** |
| S2 | Model A | Baseline | 0.8292 | 0.0868 | 0.0619 | 0.0839 | 0.7874 | 0.2752 | 0.0275 | 0.0222 | 0.0308 | 0.2747 | 0.1704 | 0.0205 | 0.0145 | 0.0223 | 0.1670 |
| | | Gaussian | **0.7994** | **0.0711** | **0.0587** | **0.0778** | 1.1450 | **0.2202** | 0.0276 | **0.0169** | **0.0273** | 0.2598 | **0.1316** | **0.0176** | **0.0128** | **0.0193** | **0.1537** |
| | | Uniform | 0.8890 | **0.0722** | **0.0472** | 0.0968 | 0.8566 | **0.2320** | 0.0286 | **0.0181** | **0.0275** | 0.2586 | **0.1457** | **0.0180** | **0.0128** | **0.0191** | **0.1467** |
| | | Epanechnikov | 0.9192 | **0.0787** | **0.0522** | 0.1048 | 0.9065 | **0.2415** | 0.0284 | **0.0177** | **0.0265** | 0.2492 | **0.1430** | **0.0162** | **0.0125** | **0.0191** | **0.1388** |
| | Model B | Baseline | 0.4930 | 0.0583 | 0.0493 | 0.0840 | 0.5898 | 0.1732 | 0.0257 | 0.0178 | 0.0300 | 0.1966 | 0.1323 | 0.0202 | 0.0129 | 0.0220 | 0.1358 |
| | | Gaussian | 0.7345 | 0.0633 | 0.0499 | **0.0759** | 0.6273 | 0.1800 | **0.0246** | **0.0155** | **0.0245** | 0.2157 | **0.1085** | **0.0160** | **0.0119** | **0.0178** | **0.1144** |
| | | Uniform | **0.4506** | 0.0716 | 0.0504 | 0.0935 | 0.6030 | **0.1707** | **0.0216** | **0.0176** | **0.0241** | 0.2243 | **0.1172** | **0.0151** | **0.0118** | **0.0175** | **0.1342** |
| | | Epanechnikov | 0.5800 | 0.0723 | 0.0545 | **0.0806** | 0.7471 | 0.2151 | **0.0263** | **0.0159** | **0.0289** | 0.1792 | **0.1038** | **0.0146** | **0.0107** | **0.0173** | **0.1302** |
| S3 | Model A | Baseline | 3.0766 | 0.7564 | 0.5350 | 0.7407 | 2.8969 | 1.2870 | 0.3854 | 0.2146 | 0.3387 | 1.3495 | 0.9232 | 0.2420 | 0.1459 | 0.2413 | 0.9960 |
| | | Gaussian | **2.9080** | 0.7769 | **0.4784** | **0.7056** | 3.4181 | 1.3139 | **0.3379** | **0.1994** | 0.3229 | **1.1897** | **0.8395** | **0.2038** | **0.1295** | **0.2145** | **0.7477** |
| | | Uniform | 3.3730 | 0.7690 | **0.4876** | 0.7832 | 3.3576 | **1.1593** | **0.3449** | **0.1912** | **0.3088** | 1.2392 | **0.8790** | **0.2064** | **0.1300** | **0.2146** | **0.7824** |
| | | Epanechnikov | 3.4479 | **0.7308** | **0.4679** | 0.8076 | 3.3156 | **1.1229** | **0.3616** | **0.1980** | **0.3099** | 1.3312 | **0.8230** | **0.2123** | **0.1337** | **0.2206** | **0.7998** |
| | Model B | Baseline | 2.8166 | 0.7558 | 0.5175 | 0.7665 | 2.2839 | 1.0061 | 0.3596 | 0.2196 | 0.3551 | 1.0990 | 0.7786 | 0.2306 | 0.1391 | 0.2380 | 0.7249 |
| | | Gaussian | **2.3173** | 0.8406 | **0.5142** | **0.7536** | 2.6542 | 1.0602 | 0.4255 | 0.2302 | **0.3526** | **1.0681** | 0.8244 | 0.2516 | 0.1416 | 0.2443 | 0.7536 |
| | | Uniform | **2.1946** | 0.8319 | 0.5193 | **0.7352** | 2.5765 | 1.2056 | 0.4038 | 0.2292 | **0.3372** | **1.0528** | **0.7167** | 0.2411 | **0.1359** | 0.2493 | **0.6890** |
| | | Epanechnikov | 2.9684 | 0.9008 | **0.4892** | 0.8704 | **2.1302** | **1.0053** | 0.3741 | 0.2297 | **0.3473** | 1.0896 | **0.7474** | 0.2453 | **0.1390** | **0.2315** | **0.6683** |

Table 4: Running-time reduction under matched stopping for scenarios 1–3, Models A and B, sample sizes, quantile levels, and smoothing kernels. Each entry is $\text{Time}_{\text{baseline}} - \text{Time}_{\text{ConquerNet}}$; every positive value indicates that ConquerNet is faster and is shown in bold. The second line gives $\text{SE}_\Delta$ in parentheses.

| Scenario | Model | Kernel | n = 1000 | | | | | n = 5000 | | | | | n = 10000 | | | | |
|---|---|---|---|---|---|---|---|---|---|---|---|---|---|---|---|---|---|
| | | | τ = 0.05 | τ = 0.25 | τ = 0.5 | τ = 0.75 | τ = 0.95 | τ = 0.05 | τ = 0.25 | τ = 0.5 | τ = 0.75 | τ = 0.95 | τ = 0.05 | τ = 0.25 | τ = 0.5 | τ = 0.75 | τ = 0.95 |
| S1 | Model A | Gaussian | **0.12** (0.28) | **0.39** (0.25) | **0.22** (0.25) | **0.27** (0.24) | **0.24** (0.27) | **0.69** (0.52) | **0.94** (0.40) | **0.28** (0.45) | **0.47** (0.52) | **0.20** (0.56) | **0.81** (0.60) | **0.07** (0.49) | **0.88** (0.56) | **1.03** (0.50) | **0.12** (0.70) |
| | | Uniform | **0.10** (0.28) | **0.11** (0.26) | **0.14** (0.22) | **0.36** (0.22) | **0.46** (0.25) | **0.64** (0.50) | **0.65** (0.43) | **0.93** (0.44) | **1.22** (0.52) | **0.11** (0.57) | **1.25** (0.57) | **0.12** (0.49) | **0.56** (0.57) | **1.22** (0.49) | -0.10 (0.67) |
| | | Epanechnikov | **0.09** (0.25) | **0.06** (0.23) | -0.10 (0.26) | **0.04** (0.24) | **0.02** (0.29) | -0.68 (0.47) | **0.45** (0.43) | -0.08 (0.44) | **0.38** (0.51) | -0.09 (0.50) | **0.42** (0.58) | -0.33 (0.49) | -0.05 (0.56) | **0.43** (0.49) | -0.07 (0.71) |
| | Model B | Gaussian | **0.40** (0.45) | **1.23** (0.44) | **0.15** (0.43) | **1.62** (0.36) | **0.92** (0.44) | **2.25** (0.77) | **0.88** (0.77) | **2.34** (0.77) | **1.51** (0.80) | **1.49** (0.85) | **0.44** (1.01) | **2.32** (0.82) | **1.11** (0.87) | **1.74** (0.94) | **0.85** (0.88) |
| | | Uniform | **1.10** (0.43) | **0.85** (0.42) | **0.84** (0.41) | **1.28** (0.35) | **1.16** (0.44) | **1.20** (0.82) | **2.04** (0.81) | **2.01** (0.72) | **2.78** (0.82) | **1.51** (0.82) | **2.02** (1.09) | **1.99** (0.85) | **2.97** (0.89) | **0.84** (0.80) | **0.85** (0.92) |
| | | Epanechnikov | **0.17** (0.43) | **0.43** (0.44) | **0.75** (0.39) | **0.90** (0.33) | **0.92** (0.41) | **1.40** (0.80) | **1.74** (0.75) | **1.98** (0.79) | **1.32** (0.81) | **0.41** (0.80) | **2.03** (1.05) | **1.45** (0.83) | **1.18** (0.88) | **1.16** (0.85) | -0.41 (0.99) |
| S2 | Model A | Gaussian | **0.03** (0.23) | **0.44** (0.22) | **0.31** (0.25) | **0.44** (0.19) | **0.23** (0.22) | **0.28** (0.41) | **1.26** (0.42) | **0.91** (0.43) | **0.68** (0.48) | **0.68** (0.47) | **0.26** (0.44) | **0.14** (0.56) | **0.83** (0.54) | **0.00** (0.63) | -0.11 (0.52) |
| | | Uniform | **0.17** (0.23) | **0.33** (0.27) | **0.23** (0.22) | **0.39** (0.20) | **0.30** (0.21) | -0.17 (0.44) | **0.99** (0.39) | **1.21** (0.49) | **0.49** (0.50) | **0.48** (0.52) | **0.73** (0.40) | **0.49** (0.47) | **0.57** (0.51) | **1.38** (0.51) | **0.22** (0.52) |
| | | Epanechnikov | -0.16 (0.22) | **0.14** (0.23) | -0.27 (0.26) | **0.42** (0.20) | **0.08** (0.24) | -0.11 (0.41) | **0.10** (0.42) | **0.38** (0.45) | -0.07 (0.49) | **0.29** (0.49) | -0.38 (0.47) | **0.22** (0.48) | -0.28 (0.55) | -0.11 (0.56) | **0.19** (0.56) |
| | Model B | Gaussian | **0.88** (0.41) | **0.93** (0.43) | **0.67** (0.36) | **0.58** (0.38) | **0.46** (0.34) | **2.03** (0.67) | **2.40** (0.85) | **1.65** (0.79) | **1.01** (0.87) | **2.58** (0.95) | **1.63** (0.80) | **0.87** (0.82) | **1.60** (1.02) | **1.79** (0.89) | **2.24** (0.95) |
| | | Uniform | **1.25** (0.37) | **1.03** (0.40) | **0.79** (0.39) | **0.71** (0.42) | **0.53** (0.37) | **1.10** (0.73) | **1.36** (0.88) | **2.60** (0.73) | **1.77** (0.76) | **2.36** (0.82) | **1.83** (0.78) | **0.82** (0.82) | **2.21** (0.89) | **0.58** (0.97) | **2.37** (0.94) |
| | | Epanechnikov | **0.70** (0.38) | **0.57** (0.41) | -0.02 (0.43) | **0.17** (0.42) | **0.55** (0.36) | **1.43** (0.69) | **1.77** (0.85) | **1.86** (0.74) | **1.12** (0.78) | **0.82** (0.89) | **0.03** (0.83) | **1.08** (0.88) | **1.80** (0.95) | **0.70** (0.96) | **1.42** (0.99) |
| S3 | Model A | Gaussian | **0.07** (0.23) | **0.14** (0.25) | **0.16** (0.22) | **0.38** (0.26) | **0.08** (0.23) | **0.91** (0.42) | **0.38** (0.50) | **1.25** (0.48) | **1.53** (0.52) | **1.06** (0.53) | **0.47** (0.49) | **0.13** (0.50) | **0.34** (0.52) | **0.58** (0.60) | -0.44 (0.58) |
| | | Uniform | -0.05 (0.27) | **0.56** (0.23) | **0.14** (0.22) | **0.34** (0.24) | **0.43** (0.23) | **0.16** (0.45) | **0.55** (0.53) | **1.38** (0.53) | **1.02** (0.55) | **0.74** (0.56) | **0.02** (0.54) | **0.61** (0.55) | **0.72** (0.50) | **0.33** (0.57) | **0.63** (0.54) |
| | | Epanechnikov | -0.50 (0.27) | **0.14** (0.26) | -0.17 (0.22) | **0.12** (0.26) | **0.24** (0.24) | -0.02 (0.45) | -0.09 (0.53) | **0.46** (0.53) | **0.64** (0.53) | **0.50** (0.52) | -0.12 (0.52) | **0.99** (0.50) | -0.09 (0.53) | **0.10** (0.63) | -0.58 (0.56) |
| | Model B | Gaussian | **0.34** (0.36) | **0.80** (0.44) | **0.40** (0.42) | **0.55** (0.37) | **0.68** (0.31) | **1.42** (0.78) | **2.84** (0.78) | **2.40** (0.89) | **1.83** (0.88) | **2.11** (0.71) | **1.44** (0.88) | **2.91** (0.98) | **1.25** (0.90) | **3.02** (0.98) | **1.33** (0.95) |
| | | Uniform | **0.47** (0.38) | **0.49** (0.43) | **0.53** (0.42) | **0.25** (0.37) | **0.87** (0.31) | **2.15** (0.75) | **1.41** (0.84) | **1.63** (0.92) | **1.48** (0.97) | **1.27** (0.81) | **2.18** (0.84) | **1.56** (0.98) | **1.70** (0.83) | **2.10** (0.97) | **1.56** (0.96) |
| | | Epanechnikov | **0.33** (0.36) | **0.37** (0.47) | -0.23 (0.47) | **0.12** (0.41) | **0.35** (0.28) | **0.56** (0.80) | **1.04** (0.78) | **0.63** (0.96) | **1.58** (0.90) | **0.72** (0.76) | **1.86** (0.91) | **3.46** (0.95) | **1.15** (0.83) | **2.40** (1.00) | **1.28** (0.99) |

exceed $1.96 \times \mathrm{SE}_\Delta$ (indicating significantly strong improvement). Together with the same-stopping-rule MSE comparison already reported in Table 3, this controlled timing comparison supports that the improvement comes from ConquerNet rather than from the stopping strategy alone.

Table 5 reports the complete mean MSE comparison together with the trial-to-trial variability. To assess whether the MSE reductions are large relative to Monte Carlo uncertainty, Table 6 further reports $\mathrm{MSE}_{\text{baseline}} - \mathrm{MSE}_{\text{ConquerNet}}$ together with the combined standard error $\mathrm{SE}_\Delta = \{(\widehat{\sigma}^2_{\text{baseline}} + \widehat{\sigma}^2_{\text{ConquerNet}})/50\}^{1/2}$. Pooled over the three pre-specified kernels, ConquerNet has lower mean MSE in 175 of 270 comparisons; 109 reductions exceed one $\mathrm{SE}_\Delta$ and 47 exceed $1.96\mathrm{SE}_\Delta$. The pattern is similar across the Gaussian, Uniform, and Epanechnikov kernels, which improve 56/90, 59/90, and 60/90 comparisons, respectively. Across sample sizes, the pooled favorable counts increase from 33/90 to 65/90 and 77/90; at the fixed low and high quantiles, they increase more sharply from 7/36 to 22/36 and 31/36. The tail-specific evidence is therefore the transition from mostly unfavorable small-sample results to highly consistent large-sample gains, rather than a higher final proportion than the overall 77/90.

Table 5: Mean squared error (MSE) for scenarios 1–3, Models A and B, sample sizes, quantile levels, and smoothing kernels. The first line gives the mean over 50 independent trials; the second line gives the sample standard error in parentheses.

| | Method | | | $n=1000$ | | | | | $n=5000$ | | | | | $n=10000$ | | |
|---|---|---|---|---|---|---|---|---|---|---|---|---|---|---|---|---|
| | | $\tau=0.05$ | $\tau=0.25$ | $\tau=0.5$ | $\tau=0.75$ | $\tau=0.95$ | $\tau=0.05$ | $\tau=0.25$ | $\tau=0.5$ | $\tau=0.75$ | $\tau=0.95$ | $\tau=0.05$ | $\tau=0.25$ | $\tau=0.5$ | $\tau=0.75$ | $\tau=0.95$ |
| **S1** Model A | Baseline | 0.3820 (0.6477) | 0.0402 (0.0178) | 0.0278 (0.0133) | 0.0374 (0.0168) | 0.3784 (0.3708) | 0.0996 (0.1214) | 0.0120 (0.0037) | 0.0086 (0.0050) | 0.0108 (0.0033) | 0.0939 (0.1467) | 0.0618 (0.0338) | 0.0067 (0.0024) | 0.0047 (0.0015) | 0.0063 (0.0027) | 0.1366 (0.4136) |
| | Gaussian | 0.4354 (0.5421) | **0.0383** (0.0177) | **0.0224** (0.0100) | 0.0382 (0.0212) | 0.5035 (0.6067) | 0.1124 (0.0846) | **0.0087** (0.0025) | **0.0055** (0.0019) | **0.0097** (0.0053) | 0.1081 (0.1048) | 0.0678 (0.0622) | **0.0064** (0.0035) | **0.0034** (0.0010) | **0.0055** (0.0026) | 0.0625 (0.0422) |
| | Uniform | 0.4448 (0.5079) | **0.0385** (0.0209) | **0.0224** (0.0092) | 0.0328 (0.0119) | 0.3721 (0.3472) | 0.1079 (0.0833) | **0.0087** (0.0030) | **0.0059** (0.0020) | **0.0104** (0.0063) | 0.1312 (0.1633) | 0.0789 (0.0637) | **0.0058** (0.0022) | **0.0036** (0.0012) | **0.0057** (0.0028) | 0.0543 (0.0364) |
| | Epanechnikov | 0.6733 (1.5255) | **0.0390** (0.0153) | **0.0222** (0.0075) | 0.0373 (0.0193) | 0.5913 (0.7009) | 0.1251 (0.1590) | **0.0093** (0.0034) | **0.0055** (0.0017) | **0.0095** (0.0035) | 0.1169 (0.1493) | 0.0653 (0.0414) | **0.0061** (0.0031) | **0.0037** (0.0018) | **0.0053** (0.0023) | 0.0665 (0.0527) |
| **S1** Model B | Baseline | 0.3842 (0.4982) | 0.0527 (0.0255) | 0.0319 (0.0132) | 0.0475 (0.0224) | 0.4222 (0.5016) | 0.1143 (0.1129) | 0.0149 (0.0044) | 0.0107 (0.0028) | 0.0151 (0.0054) | 0.1277 (0.1658) | 0.1691 (0.4637) | 0.0099 (0.0043) | 0.0066 (0.0023) | 0.0082 (0.0027) | 0.3882 (2.2095) |
| | Gaussian | 0.4158 (0.2411) | 0.0665 (0.0302) | 0.0383 (0.0097) | 0.0633 (0.0169) | 0.5202 (0.3019) | 0.1172 (0.0797) | **0.0144** (0.0046) | **0.0097** (0.0031) | **0.0142** (0.0037) | **0.1066** (0.0862) | 0.1062 (0.2169) | **0.0095** (0.0035) | **0.0055** (0.0020) | **0.0086** (0.0036) | **0.0726** (0.0878) |
| | Uniform | 0.5652 (0.6474) | 0.0612 (0.0219) | 0.0378 (0.0108) | 0.0614 (0.0317) | 0.5685 (0.4267) | **0.1033** (0.0721) | **0.0145** (0.0038) | **0.0091** (0.0027) | 0.0154 (0.0043) | **0.1252** (0.1185) | **0.0974** (0.1867) | **0.0093** (0.0049) | **0.0055** (0.0018) | **0.0080** (0.0026) | **0.0736** (0.0521) |
| | Epanechnikov | 0.4275 (0.3151) | 0.0582 (0.0146) | 0.0383 (0.0109) | 0.0626 (0.0217) | 0.6050 (0.4719) | **0.1115** (0.0831) | **0.0146** (0.0036) | **0.0094** (0.0030) | **0.0145** (0.0038) | **0.1162** (0.0968) | **0.1294** (0.3764) | **0.0089** (0.0029) | **0.0057** (0.0019) | 0.0092 (0.0046) | **0.0663** (0.0429) |
| **S2** Model A | Baseline | 0.8292 (0.8274) | 0.0868 (0.1143) | 0.0619 (0.0325) | 0.0839 (0.0431) | 0.7874 (0.8127) | 0.2752 (0.1868) | 0.0275 (0.0146) | 0.0222 (0.0116) | 0.0308 (0.0144) | 0.2747 (0.1527) | 0.1704 (0.0853) | 0.0205 (0.0091) | 0.0145 (0.0046) | 0.0223 (0.0089) | 0.1670 (0.0537) |
| | Gaussian | **0.7994** (0.7103) | **0.0711** (0.0482) | **0.0587** (0.0622) | **0.0778** (0.0376) | 1.1466 (1.4345) | **0.2202** (0.1234) | 0.0276 (0.0177) | **0.0169** (0.0070) | **0.0273** (0.0123) | 0.2598 (0.2009) | **0.1316** (0.0642) | **0.0176** (0.0081) | **0.0128** (0.0052) | **0.0193** (0.0111) | 0.1537 (0.0718) |
| | Uniform | 0.8892 (0.8173) | **0.0721** (0.0432) | **0.0471** (0.0271) | 0.0964 (0.0855) | 0.8566 (0.7087) | 0.2308 (0.1131) | 0.0286 (0.0205) | **0.0181** (0.0099) | **0.0275** (0.0174) | 0.2586 (0.1715) | 0.1457 (0.0746) | **0.0180** (0.0114) | **0.0128** (0.0063) | **0.0191** (0.0091) | 0.1467 (0.0686) |
| | Epanechnikov | 0.9192 (1.2469) | **0.0787** (0.0596) | **0.0522** (0.0351) | 0.1048 (0.0934) | 0.9074 (0.9761) | 0.2415 (0.1555) | 0.0284 (0.0175) | **0.0177** (0.0086) | **0.0265** (0.0149) | 0.2492 (0.1881) | 0.1430 (0.0698) | **0.0162** (0.0079) | **0.0126** (0.0051) | **0.0191** (0.0090) | 0.1388 (0.0581) |
| **S2** Model B | Baseline | 0.4930 (0.5249) | 0.0583 (0.0268) | 0.0493 (0.0302) | 0.0840 (0.0439) | 0.5898 (0.5036) | 0.1732 (0.0803) | 0.0257 (0.0106) | 0.0178 (0.0062) | 0.0300 (0.0129) | 0.1966 (0.1159) | 0.1323 (0.1030) | 0.0202 (0.0118) | 0.0129 (0.0044) | 0.0220 (0.0095) | 0.1358 (0.0493) |
| | Gaussian | 0.7367 (1.1684) | 0.0639 (0.0401) | 0.0500 (0.0349) | **0.0759** (0.0375) | 0.6273 (0.6874) | 0.1800 (0.1209) | **0.0246** (0.0140) | **0.0155** (0.0063) | **0.0245** (0.0171) | 0.2157 (0.1665) | **0.1085** (0.0614) | **0.0160** (0.0120) | **0.0119** (0.0052) | **0.0178** (0.0090) | **0.1144** (0.0549) |
| | Uniform | **0.4515** (0.4592) | 0.0717 (0.0766) | 0.0503 (0.0351) | 0.0935 (0.0645) | 0.6034 (0.5697) | **0.1706** (0.1317) | 0.0216 (0.0131) | **0.0176** (0.0111) | **0.0241** (0.0136) | 0.2239 (0.1927) | **0.1172** (0.0872) | **0.0151** (0.0076) | **0.0118** (0.0061) | **0.0175** (0.0070) | **0.1342** (0.0682) |
| | Epanechnikov | 0.5800 (0.6180) | 0.0723 (0.0723) | 0.0546 (0.0480) | **0.0819** (0.0475) | 0.7484 (0.7563) | 0.2151 (0.1642) | 0.0263 (0.0184) | **0.0159** (0.0076) | **0.0289** (0.0268) | 0.1802 (0.0987) | **0.1038** (0.0635) | **0.0146** (0.0070) | **0.0107** (0.0033) | **0.0173** (0.0109) | **0.1302** (0.0686) |
| **S3** Model A | Baseline | 3.0766 (1.9288) | 0.7564 (0.2742) | 0.5350 (0.1755) | 0.7407 (0.2334) | 2.8969 (1.5409) | 1.2870 (0.5969) | 0.3854 (0.1527) | 0.2146 (0.0363) | 0.3387 (0.0768) | 1.3495 (0.6900) | 0.9232 (0.2460) | 0.2420 (0.0577) | 0.1459 (0.0195) | 0.2413 (0.0475) | 0.9960 (0.4499) |
| | Gaussian | **2.8841** (2.1654) | **0.7776** (0.5278) | **0.4788** (0.1456) | **0.7056** (0.3539) | 3.4222 (2.4064) | 1.3139 (0.8196) | **0.3379** (0.0955) | **0.1993** (0.0474) | **0.3269** (0.0706) | 1.1897 (0.5537) | **0.8395** (0.3993) | **0.2040** (0.0392) | **0.1295** (0.0188) | **0.2145** (0.0390) | **0.7477** (0.2841) |
| | Uniform | 3.3730 (2.1527) | **0.7735** (0.4435) | **0.4876** (0.1620) | 0.7832 (0.4280) | 3.3576 (3.1142) | **1.1577** (0.4404) | **0.3449** (0.0935) | **0.1912** (0.0351) | **0.3082** (0.0600) | 1.2392 (0.5797) | **0.8790** (0.3664) | **0.2064** (0.0454) | **0.1300** (0.0209) | **0.2146** (0.0384) | **0.7824** (0.2579) |
| | Epanechnikov | 3.4479 (2.3894) | **0.7308** (0.2561) | **0.4679** (0.1108) | 0.8153 (0.4724) | 3.3156 (1.8968) | **1.1229** (0.3912) | **0.3615** (0.1483) | **0.1980** (0.0497) | **0.3112** (0.0631) | 1.3312 (0.5874) | **0.8230** (0.3925) | **0.2129** (0.0437) | **0.1349** (0.0265) | **0.2206** (0.0517) | **0.7998** (0.4182) |
| **S3** Model B | Baseline | 2.8166 (2.0979) | 0.7558 (0.2711) | 0.5175 (0.1449) | 0.7665 (0.3596) | 2.2839 (1.8386) | 1.0061 (0.4096) | 0.3596 (0.0927) | 0.2196 (0.0384) | 0.3551 (0.0767) | 1.0990 (0.5519) | 0.7786 (0.2225) | 0.2306 (0.0531) | 0.1391 (0.0255) | 0.2380 (0.0597) | 0.7249 (0.2336) |
| | Gaussian | **2.3193** (1.7892) | 0.8405 (0.4052) | **0.5142** (0.1287) | **0.7543** (0.3117) | 2.6520 (3.5745) | 1.0602 (0.5594) | 0.4263 (0.1428) | 0.2304 (0.0494) | **0.3525** (0.0864) | **1.0681** (0.5810) | 0.8256 (0.3250) | 0.2518 (0.0931) | 0.1416 (0.0334) | 0.2444 (0.0599) | 0.7536 (0.3181) |
| | Uniform | **2.1946** (1.6520) | 0.8316 (0.5273) | 0.5193 (0.1790) | **0.7352** (0.2589) | 2.5765 (1.8300) | 1.2056 (0.7286) | 0.4038 (0.1408) | 0.2320 (0.0593) | **0.3373** (0.0812) | **1.0528** (0.6487) | 0.7167 (0.2505) | 0.2409 (0.0635) | **0.1372** (0.0285) | 0.2493 (0.0747) | **0.6920** (0.2713) |
| | Epanechnikov | 2.9702 (2.5447) | 0.9008 (0.7190) | **0.4892** (0.1324) | 0.8702 (0.5122) | **2.1331** (1.4889) | 1.0053 (0.5012) | 0.3750 (0.0798) | 0.2297 (0.0638) | **0.3473** (0.0978) | 1.0895 (0.6760) | 0.7502 (0.2473) | 0.2454 (0.0775) | **0.1390** (0.0302) | **0.2315** (0.0563) | **0.6683** (0.2145) |

### B.1.1 Sensitivity of the choice of bandwidth

In this section, we present MSE results for our ConquerNet with a sample size of 10000 and different bandwidth choices, as shown in Table 7. Bandwidths $h = \{0.001, 0.005, 0.01, 0.05, 0.1\}$ are considered. The results show that our ConquerNet outperform the baseline model over a wide range of bandwidths.

Table 6: MSE reduction of ConquerNet relative to the nonsmoothed baseline for scenarios 1–3, Models A and B, sample sizes, quantile levels, and smoothing kernels. Each entry is $\text{MSE}_{\text{baseline}} - \text{MSE}_{\text{ConquerNet}}$, so positive values indicate lower MSE for ConquerNet and are shown in bold. The second line gives the combined standard error in parentheses.

| | | Kernel | $n = 1000$ | | | | | $n = 5000$ | | | | | $n = 10000$ | | | | |
|---|---|---|---|---|---|---|---|---|---|---|---|---|---|---|---|---|---|
| | | | $\tau=0.05$ | $\tau=0.25$ | $\tau=0.5$ | $\tau=0.75$ | $\tau=0.95$ | $\tau=0.05$ | $\tau=0.25$ | $\tau=0.5$ | $\tau=0.75$ | $\tau=0.95$ | $\tau=0.05$ | $\tau=0.25$ | $\tau=0.5$ | $\tau=0.75$ | $\tau=0.95$ |
| **S1** Model A | Gaussian | | -0.0534 (0.1194) | **0.0019** (0.0036) | **0.0054** (0.0024) | -0.0008 (0.0038) | -0.1251 (0.1006) | -0.0128 (0.0209) | **0.0033** (0.0006) | **0.0031** (0.0008) | **0.0011** (0.0009) | -0.0142 (0.0255) | -0.0060 (0.0100) | **0.0003** (0.0006) | **0.0013** (0.0003) | **0.0008** (0.0005) | **0.0741** (0.0588) |
| | Uniform | | -0.0628 (0.1164) | **0.0017** (0.0039) | **0.0054** (0.0023) | **0.0046** (0.0029) | **0.0063** (0.0718) | -0.0083 (0.0208) | **0.0033** (0.0007) | **0.0027** (0.0008) | **0.0004** (0.0010) | -0.0373 (0.0310) | -0.0171 (0.0102) | **0.0009** (0.0005) | **0.0011** (0.0003) | **0.0006** (0.0005) | **0.0823** (0.0587) |
| | Epanechnikov | | -0.2913 (0.2344) | **0.0012** (0.0033) | **0.0056** (0.0022) | **0.0001** (0.0036) | -0.2129 (0.1121) | -0.0255 (0.0283) | **0.0027** (0.0007) | **0.0031** (0.0007) | **0.0013** (0.0007) | -0.0230 (0.0296) | -0.0035 (0.0076) | **0.0006** (0.0006) | **0.0010** (0.0003) | **0.0010** (0.0005) | **0.0701** (0.0590) |
| S1 Model B | Gaussian | | -0.0316 (0.0783) | -0.0138 (0.0056) | -0.0064 (0.0023) | -0.0158 (0.0040) | -0.0980 (0.0828) | -0.0029 (0.0195) | **0.0005** (0.0009) | **0.0010** (0.0006) | **0.0009** (0.0009) | **0.0211** (0.0264) | **0.0629** (0.0724) | **0.0004** (0.0008) | **0.0011** (0.0004) | -0.0004 (0.0006) | **0.3156** (0.3127) |
| | Uniform | | -0.1810 (0.1155) | -0.0085 (0.0048) | -0.0059 (0.0024) | -0.0139 (0.0055) | -0.1463 (0.0931) | **0.0110** (0.0189) | **0.0004** (0.0008) | **0.0016** (0.0006) | -0.0003 (0.0010) | **0.0025** (0.0288) | **0.0717** (0.0707) | **0.0006** (0.0009) | **0.0011** (0.0004) | **0.0002** (0.0005) | **0.3146** (0.3126) |
| | Epanechnikov | | -0.0433 (0.0834) | -0.0055 (0.0042) | -0.0064 (0.0024) | -0.0151 (0.0044) | -0.1828 (0.0974) | **0.0028** (0.0198) | **0.0003** (0.0008) | **0.0013** (0.0006) | **0.0006** (0.0009) | **0.0115** (0.0272) | **0.0397** (0.0845) | **0.0010** (0.0007) | **0.0009** (0.0004) | -0.0010 (0.0008) | **0.3219** (0.3125) |
| **S2** Model A | Gaussian | | **0.0298** (0.1542) | **0.0157** (0.0175) | **0.0032** (0.0099) | **0.0061** (0.0081) | -0.3592 (0.2332) | **0.0550** (0.0317) | -0.0001 (0.0032) | **0.0053** (0.0019) | **0.0035** (0.0027) | **0.0149** (0.0357) | **0.0388** (0.0151) | **0.0029** (0.0017) | **0.0017** (0.0010) | **0.0030** (0.0020) | **0.0133** (0.0127) |
| | Uniform | | -0.0600 (0.1645) | **0.0147** (0.0173) | **0.0148** (0.0060) | -0.0125 (0.0135) | -0.0692 (0.1525) | **0.0444** (0.0309) | -0.0011 (0.0036) | **0.0041** (0.0022) | **0.0033** (0.0032) | **0.0161** (0.0325) | **0.0247** (0.0160) | **0.0025** (0.0021) | **0.0017** (0.0011) | **0.0032** (0.0018) | **0.0203** (0.0123) |
| | Epanechnikov | | -0.0900 (0.2116) | **0.0081** (0.0182) | **0.0097** (0.0068) | -0.0209 (0.0145) | -0.1200 (0.1796) | **0.0337** (0.0344) | -0.0009 (0.0032) | **0.0045** (0.0020) | **0.0043** (0.0029) | **0.0255** (0.0343) | **0.0274** (0.0156) | **0.0043** (0.0017) | **0.0019** (0.0010) | **0.0032** (0.0018) | **0.0282** (0.0112) |
| S2 Model B | Gaussian | | -0.2437 (0.1811) | -0.0056 (0.0068) | -0.0007 (0.0065) | **0.0081** (0.0082) | -0.0375 (0.1205) | -0.0068 (0.0205) | **0.0011** (0.0025) | **0.0023** (0.0013) | **0.0055** (0.0030) | -0.0191 (0.0287) | **0.0238** (0.0170) | **0.0042** (0.0024) | **0.0010** (0.0010) | **0.0042** (0.0019) | **0.0214** (0.0104) |
| | Uniform | | **0.0415** (0.0986) | -0.0134 (0.0115) | -0.0010 (0.0065) | -0.0095 (0.0110) | -0.0136 (0.1075) | **0.0026** (0.0218) | **0.0041** (0.0024) | **0.0002** (0.0018) | **0.0059** (0.0027) | -0.0273 (0.0318) | **0.0151** (0.0191) | **0.0051** (0.0020) | **0.0011** (0.0011) | **0.0045** (0.0017) | **0.0016** (0.0119) |
| | Epanechnikov | | -0.0870 (0.1147) | -0.0140 (0.0109) | -0.0053 (0.0080) | **0.0021** (0.0091) | -0.1586 (0.1285) | -0.0419 (0.0258) | -0.0006 (0.0030) | **0.0019** (0.0014) | **0.0011** (0.0042) | **0.0164** (0.0215) | **0.0285** (0.0171) | **0.0056** (0.0019) | **0.0022** (0.0008) | **0.0047** (0.0020) | **0.0056** (0.0119) |
| **S3** Model A | Gaussian | | **0.1925** (0.4101) | -0.0212 (0.0841) | **0.0562** (0.0322) | **0.0351** (0.0600) | -0.5253 (0.4041) | -0.0269 (0.1434) | **0.0475** (0.0255) | **0.0153** (0.0084) | **0.0118** (0.0148) | **0.1598** (0.1251) | **0.0837** (0.0663) | **0.0380** (0.0099) | **0.0164** (0.0038) | **0.0268** (0.0087) | **0.2483** (0.0752) |
| | Uniform | | -0.2964 (0.4088) | -0.0171 (0.0737) | **0.0474** (0.0338) | -0.0425 (0.0689) | -0.4607 (0.4914) | **0.1293** (0.1049) | **0.0405** (0.0253) | **0.0234** (0.0071) | **0.0305** (0.0138) | **0.1103** (0.1274) | **0.0442** (0.0624) | **0.0356** (0.0104) | **0.0159** (0.0040) | **0.0267** (0.0086) | **0.2136** (0.0733) |
| | Epanechnikov | | -0.3713 (0.4343) | **0.0256** (0.0531) | **0.0671** (0.0294) | -0.0746 (0.0745) | -0.4187 (0.3456) | **0.1641** (0.1009) | **0.0239** (0.0301) | **0.0166** (0.0087) | **0.0275** (0.0141) | **0.0183** (0.1282) | **0.1002** (0.0655) | **0.0291** (0.0102) | **0.0110** (0.0047) | **0.0207** (0.0099) | **0.1962** (0.0869) |
| S3 Model B | Gaussian | | **0.4973** (0.3899) | -0.0847 (0.0689) | **0.0033** (0.0274) | **0.0122** (0.0673) | -0.3681 (0.5685) | -0.0541 (0.0981) | -0.0667 (0.0241) | -0.0108 (0.0088) | **0.0026** (0.0163) | **0.0309** (0.1133) | -0.0470 (0.0557) | -0.0212 (0.0152) | -0.0025 (0.0059) | -0.0064 (0.0120) | -0.0287 (0.0558) |
| | Uniform | | **0.6220** (0.3776) | -0.0758 (0.0838) | -0.0018 (0.0326) | **0.0313** (0.0627) | -0.2926 (0.3669) | -0.1995 (0.1182) | -0.0442 (0.0238) | -0.0124 (0.0100) | **0.0178** (0.0158) | **0.0462** (0.1204) | **0.0619** (0.0474) | -0.0103 (0.0117) | **0.0019** (0.0054) | -0.0113 (0.0135) | **0.0329** (0.0506) |
| | Epanechnikov | | -0.1536 (0.4664) | -0.1450 (0.1087) | **0.0283** (0.0278) | -0.1037 (0.0885) | **0.1508** (0.3346) | **0.0008** (0.0915) | -0.0154 (0.0173) | -0.0101 (0.0105) | **0.0078** (0.0176) | **0.0095** (0.1234) | **0.0284** (0.0470) | -0.0148 (0.0133) | **0.0001** (0.0056) | **0.0065** (0.0116) | **0.0566** (0.0449) |

### B.1.2 Residual-based neural networks

We also study the effect of residual-based neural networks. We add one residual block for every hidden layer in the baseline and ConquerNet to study the MSE performance. We compare the baseline model performance without residual blocks to that using residual-based structures. The results are shown in Table 9. The numbers with brackets in the table represent the standard error of the MSEs over the 50 trials, and the bold font means that the model has a smaller MSE. We find from the results that the residual-based neural networks have no advantage in reducing MSE in many scenarios. We also make the comparison for ConquerNet, as shown in Table 10 for the Gaussian kernel, for example. Residual blocks also fail to significantly reduce the MSE in the Gaussian ConquerNet. However, we find that for high ($\tau = 0.95$) or low ($\tau = 0.05$) quantile levels, the models with residual structures tend to have smaller MSEs and standard errors, which increases the training stability. We provide the table of the MSE performances for baseline models and ConquerNet with residual blocks, see Table 11. We can see that for residual-based networks, the ConquerNet still have better performance than the baseline models.

### B.1.3 Non-crossing constraints

We also implement the simulations for the joint estimation of multiple quantile levels under non-crossing constraints, see Padilla et al. (2022) for the estimation of baseline networks. For the ConquerNet, for multiple quantile levels are given in the set $\Gamma \subset (0, 1)$, we estimate the quantile functions by solving

$$
\begin{aligned}
\{\hat{f}_\tau\}_{\tau \in \Gamma} = &\underset{\{f_\tau\}_{\tau \in \Gamma}}{\arg \min} \sum_{\tau \in \Gamma} \sum_{i=1}^{n} \ell_\tau(y_i - f_\tau(\mathbf{x}_i)) \\
&\text{subject to} \quad f_\tau(\mathbf{x}_i) \leq f_{\tau'}(\mathbf{x}_i) \quad \forall \tau < \tau', \tau, \tau' \in \Gamma, i = 1, \ldots, n.
\end{aligned}
$$

Table 7: MSE performances for scenario 1-3, model A and B, sample size 10000 under different bandwidths, quantile levels, and smoothing kernels. The MSEs are averaged over 50 independent trials.

| $h$ | Method | Scenario 1, Model A | | | | | Scenario 1, Model B | | | | |
|---|---|---|---|---|---|---|---|---|---|---|---|
| | | $\tau=0.05$ | $\tau=0.25$ | $\tau=0.5$ | $\tau=0.75$ | $\tau=0.95$ | $\tau=0.05$ | $\tau=0.25$ | $\tau=0.5$ | $\tau=0.75$ | $\tau=0.95$ |
| | Baseline | 0.0618 | 0.0067 | 0.0047 | 0.0063 | 0.1366 | 0.1691 | 0.0099 | 0.0066 | 0.0082 | 0.3882 |
| 0.001 | Gaussian | 0.0678 | **0.0064** | **0.0034** | **0.0055** | **0.0625** | **0.0654** | 0.0101 | **0.0061** | 0.0083 | **0.0708** |
| | Uniform | 0.0789 | **0.0058** | **0.0036** | **0.0057** | **0.0543** | **0.1356** | **0.0094** | **0.0060** | 0.0089 | **0.0679** |
| | Epanechnikov | 0.0653 | **0.0061** | **0.0037** | **0.0053** | **0.0665** | **0.1225** | **0.0098** | **0.0056** | 0.0098 | **0.0699** |
| 0.005 | Gaussian | 0.0716 | **0.0061** | **0.0033** | **0.0054** | **0.0580** | **0.1062** | **0.0095** | **0.0055** | 0.0086 | **0.0726** |
| | Uniform | 0.0645 | **0.0056** | **0.0038** | **0.0056** | **0.0549** | **0.0974** | **0.0093** | **0.0055** | **0.0080** | **0.0736** |
| | Epanechnikov | 0.0676 | **0.0062** | **0.0036** | **0.0053** | **0.0587** | **0.1294** | **0.0089** | **0.0057** | 0.0092 | **0.0663** |
| 0.01 | Gaussian | 0.0624 | **0.0062** | **0.0037** | **0.0060** | **0.0662** | **0.0755** | **0.0091** | **0.0058** | 0.0087 | **0.0722** |
| | Uniform | 0.0687 | **0.0057** | **0.0037** | **0.0056** | **0.0805** | **0.0717** | **0.0090** | **0.0053** | 0.0088 | **0.0617** |
| | Epanechnikov | 0.0800 | **0.0059** | **0.0035** | **0.0056** | **0.0749** | **0.0846** | **0.0093** | **0.0056** | **0.0081** | **0.0737** |
| 0.05 | Gaussian | 0.0913 | 0.0081 | **0.0034** | 0.0074 | **0.0623** | **0.1055** | 0.0120 | **0.0059** | 0.0107 | **0.0748** |
| | Uniform | 0.0718 | **0.0066** | **0.0036** | **0.0060** | **0.0582** | **0.0746** | **0.0094** | **0.0055** | 0.0090 | **0.0878** |
| | Epanechnikov | 0.1112 | **0.0063** | **0.0033** | **0.0054** | **0.0744** | **0.0801** | **0.0093** | **0.0056** | 0.0096 | **0.0607** |
| 0.1 | Gaussian | 0.1153 | 0.0183 | **0.0035** | 0.0174 | **0.1076** | **0.1593** | 0.0215 | **0.0057** | 0.0203 | **0.1167** |
| | Uniform | 0.0687 | 0.0101 | **0.0037** | 0.0090 | **0.0758** | **0.1179** | 0.0135 | **0.0060** | 0.0130 | **0.0767** |
| | Epanechnikov | 0.0690 | 0.0085 | **0.0036** | 0.0071 | **0.0681** | **0.1006** | 0.0122 | **0.0054** | 0.0107 | **0.0872** |

| $h$ | Method | Scenario 2, Model A | | | | | Scenario 2, Model B | | | | |
|---|---|---|---|---|---|---|---|---|---|---|---|
| | | $\tau=0.05$ | $\tau=0.25$ | $\tau=0.5$ | $\tau=0.75$ | $\tau=0.95$ | $\tau=0.05$ | $\tau=0.25$ | $\tau=0.5$ | $\tau=0.75$ | $\tau=0.95$ |
| | Baseline | 0.1704 | 0.0205 | 0.0145 | 0.0223 | 0.1670 | 0.1323 | 0.0202 | 0.0129 | 0.0220 | 0.1358 |
| 0.001 | Gaussian | **0.1316** | **0.0176** | **0.0128** | **0.0193** | **0.1537** | **0.1085** | **0.0160** | **0.0119** | **0.0178** | **0.1144** |
| | Uniform | **0.1457** | **0.0180** | **0.0128** | **0.0191** | **0.1467** | **0.1172** | **0.0151** | **0.0118** | **0.0175** | **0.1342** |
| | Epanechnikov | **0.1430** | **0.0162** | **0.0126** | **0.0191** | **0.1388** | **0.1038** | **0.0146** | **0.0107** | **0.0173** | **0.1302** |
| 0.005 | Gaussian | **0.1428** | **0.0184** | **0.0134** | **0.0182** | **0.1371** | **0.1029** | **0.0148** | **0.0118** | **0.0164** | **0.1264** |
| | Uniform | **0.1457** | **0.0184** | **0.0128** | **0.0185** | **0.1423** | **0.1250** | **0.0151** | **0.0115** | **0.0171** | **0.1275** |
| | Epanechnikov | **0.1551** | **0.0167** | **0.0121** | **0.0185** | **0.1262** | **0.1262** | **0.0186** | 0.0129 | **0.0172** | **0.1326** |
| 0.01 | Gaussian | **0.1315** | **0.0165** | **0.0123** | **0.0172** | **0.1413** | **0.1166** | **0.0154** | **0.0112** | **0.0177** | **0.1117** |
| | Uniform | **0.1590** | **0.0169** | **0.0109** | **0.0180** | **0.1493** | **0.1079** | **0.0128** | **0.0108** | **0.0187** | **0.1142** |
| | Epanechnikov | **0.1364** | **0.0184** | **0.0118** | **0.0197** | **0.1366** | **0.1118** | **0.0152** | **0.0111** | **0.0184** | **0.1119** |
| 0.05 | Gaussian | **0.1478** | **0.0178** | **0.0122** | **0.0191** | **0.1553** | **0.1151** | **0.0146** | **0.0100** | **0.0193** | **0.1333** |
| | Uniform | **0.1328** | **0.0171** | **0.0131** | **0.0195** | **0.1400** | **0.1302** | **0.0135** | **0.0120** | **0.0174** | **0.1071** |
| | Epanechnikov | **0.1650** | **0.0162** | **0.0125** | **0.0202** | **0.1363** | **0.1093** | **0.0167** | **0.0109** | **0.0212** | **0.1247** |
| 0.1 | Gaussian | **0.1558** | **0.0179** | **0.0121** | **0.0193** | **0.1341** | **0.0980** | **0.0157** | **0.0106** | **0.0190** | **0.1256** |
| | Uniform | **0.1474** | **0.0176** | **0.0121** | **0.0188** | **0.1347** | **0.1198** | **0.0153** | **0.0126** | **0.0192** | **0.1169** |
| | Epanechnikov | **0.1533** | **0.0181** | **0.0120** | **0.0196** | **0.1421** | **0.1170** | **0.0162** | **0.0118** | **0.0201** | **0.1302** |

| $h$ | Method | Scenario 3, Model A | | | | | Scenario 3, Model B | | | | |
|---|---|---|---|---|---|---|---|---|---|---|---|
| | | $\tau=0.05$ | $\tau=0.25$ | $\tau=0.5$ | $\tau=0.75$ | $\tau=0.95$ | $\tau=0.05$ | $\tau=0.25$ | $\tau=0.5$ | $\tau=0.75$ | $\tau=0.95$ |
| | Baseline | 0.9232 | 0.2420 | 0.1459 | 0.2413 | 0.9960 | 0.7786 | 0.2306 | 0.1391 | 0.2380 | 0.7249 |
| 0.001 | Gaussian | **0.8395** | **0.2040** | **0.1295** | **0.2145** | **0.7477** | 0.8256 | 0.2518 | 0.1416 | 0.2444 | 0.7536 |
| | Uniform | **0.8790** | **0.2064** | **0.1300** | **0.2146** | **0.7824** | **0.7167** | 0.2409 | **0.1372** | 0.2493 | **0.6920** |
| | Epanechnikov | **0.8230** | **0.2129** | **0.1349** | **0.2206** | **0.7998** | **0.7502** | 0.2454 | **0.1390** | **0.2315** | **0.6683** |
| 0.005 | Gaussian | **0.8707** | **0.2077** | **0.1308** | **0.2263** | **0.8742** | **0.7740** | 0.2385 | 0.1413 | 0.2528 | **0.6860** |
| | Uniform | **0.8739** | **0.2021** | **0.1276** | **0.2249** | **0.8332** | **0.7142** | **0.2279** | **0.1374** | **0.2292** | 0.7574 |
| | Epanechnikov | **0.8138** | **0.2084** | **0.1307** | **0.2166** | **0.8117** | **0.7051** | 0.2376 | **0.1344** | 0.2529 | 0.7390 |
| 0.01 | Gaussian | **0.8995** | **0.1998** | **0.1275** | **0.2313** | **0.7526** | **0.7052** | 0.2473 | **0.1362** | 0.2381 | **0.7225** |
| | Uniform | **0.8739** | **0.2058** | **0.1289** | **0.2074** | **0.8256** | **0.7404** | 0.2411 | **0.1384** | 0.2390 | 0.7370 |
| | Epanechnikov | **0.8079** | **0.2167** | **0.1317** | **0.2294** | **0.7893** | 0.8089 | 0.2330 | 0.1393 | 0.2497 | **0.6786** |
| 0.05 | Gaussian | **0.8527** | **0.2083** | **0.1356** | **0.2211** | **0.7853** | **0.7434** | 0.2357 | **0.1332** | **0.2350** | **0.6987** |
| | Uniform | **0.8250** | **0.2164** | **0.1352** | **0.2280** | **0.7966** | **0.7513** | 0.2485 | **0.1349** | 0.2485 | **0.6703** |
| | Epanechnikov | **0.7879** | **0.2079** | **0.1310** | **0.2128** | **0.7526** | **0.6812** | 0.2372 | **0.1379** | 0.2439 | **0.7231** |
| 0.1 | Gaussian | **0.8220** | **0.2125** | **0.1348** | **0.2114** | **0.8293** | **0.7644** | **0.2302** | **0.1326** | 0.2467 | **0.6604** |
| | Uniform | **0.8100** | **0.2118** | **0.1371** | **0.2253** | **0.8218** | **0.7501** | **0.2295** | **0.1391** | 0.2393 | **0.7201** |
| | Epanechnikov | **0.9015** | **0.2132** | **0.1310** | **0.2204** | **0.7852** | **0.6904** | 0.2471 | **0.1302** | **0.2262** | **0.6741** |

Table 8: Mean squared error (MSE) performances for scenario 2, model A and B with 5-fold cross-validation under different sample sizes, quantile levels, and smoothing kernels. The MSEs are averaged over 50 independent trials.

| $n$ | Method | Scenario 2, Model A | | | | | Scenario 2, Model B | | | | |
|---|---|---|---|---|---|---|---|---|---|---|---|
| | | $\tau$=0.05 | $\tau$=0.25 | $\tau$=0.5 | $\tau$=0.75 | $\tau$=0.95 | $\tau$=0.05 | $\tau$=0.25 | $\tau$=0.5 | $\tau$=0.75 | $\tau$=0.95 |
| 1000 | Baseline | 0.9195 | [0.0714] | 0.0586 | 0.0894 | 0.8167 | 0.5480 | [0.0674] | [0.0462] | 0.0838 | [0.4792] |
| | Gaussian | **0.7859** | 0.0763 | **0.0498** | **0.0890** | 0.8510 | 0.5849 | 0.0751 | 0.0555 | **0.0814** | 0.6314 |
| | Uniform | 0.9437 | 0.0827 | **0.0569** | **0.0744** | **0.7308** | **0.5434** | 0.0740 | 0.0499 | **0.0802** | 1.0782 |
| | Epanechnikov | 1.0141 | 0.0759 | **0.0462** | 0.0919 | **0.6248** | 0.5938 | 0.0688 | 0.0561 | **0.0787** | 0.6151 |
| 5000 | Baseline | 0.3446 | 0.0258 | 0.0241 | 0.0335 | 0.3439 | [0.1629] | 0.0265 | 0.0188 | 0.0272 | 0.2205 |
| | Gaussian | **0.2358** | 0.0275 | **0.0180** | **0.0302** | **0.3064** | 0.1724 | **0.0199** | **0.0161** | 0.0261 | 0.2244 |
| | Uniform | **0.2477** | 0.0310 | **0.0184** | **0.0268** | **0.2732** | 0.1804 | **0.0256** | **0.0182** | 0.0258 | **0.2045** |
| | Epanechnikov | **0.2556** | **0.0237** | **0.0169** | **0.0273** | **0.2657** | 0.2104 | **0.0227** | **0.0161** | 0.0295 | 0.2357 |
| 10000 | Baseline | 0.1511 | 0.0193 | 0.0164 | 0.0229 | 0.1853 | 0.1218 | 0.0152 | 0.0128 | 0.0223 | 0.1522 |
| | Gaussian | 0.1577 | **0.0162** | **0.0139** | **0.0184** | **0.1401** | **0.1192** | **0.0138** | **0.0113** | **0.0189** | **0.1375** |
| | Uniform | **0.1489** | **0.0153** | **0.0120** | **0.0210** | **0.1508** | 0.1255 | 0.0164 | **0.0112** | **0.0171** | 0.1592 |
| | Epanechnikov | **0.1431** | **0.0165** | **0.0121** | **0.0187** | 0.1852 | **0.1091** | 0.0167 | **0.0113** | **0.0166** | **0.1122** |

Table 9: Mean squared error (MSE) and its standard error performances for baseline, scenario 1-3, model A and B with residual blocks under different sample sizes, quantile levels. "Original" means the original baseline network without residual blocks, and "+Res" means the baseline network with residual blocks. The MSEs are averaged over 50 independent trials. The bracket means the standard error of the MSEs over the trials.

| Baseline | | | n=1000 | | | | | n=5000 | | | | | n=10000 | | | | |
|---|---|---|---|---|---|---|---|---|---|---|---|---|---|---|---|---|---|
| | | | $\tau$=0.05 | $\tau$=0.25 | $\tau$=0.5 | $\tau$=0.75 | $\tau$=0.95 | $\tau$=0.05 | $\tau$=0.25 | $\tau$=0.5 | $\tau$=0.75 | $\tau$=0.95 | $\tau$=0.05 | $\tau$=0.25 | $\tau$=0.5 | $\tau$=0.75 | $\tau$=0.95 |
| S1 | Model A | Original | 0.3820 | **0.0402** | **0.0278** | **0.0374** | 0.3784 | **0.0996** | **0.0120** | **0.0086** | **0.0108** | 0.0939 | **0.0618** | **0.0067** | **0.0047** | **0.0063** | 0.1366 |
| | | | (0.0916) | (0.0025) | (0.0019) | (0.0024) | (0.0524) | (0.0172) | (0.0005) | (0.0007) | (0.0005) | (0.0207) | (0.0048) | (0.0003) | (0.0002) | (0.0004) | (0.0585) |
| | | + Res | **0.3316** | 0.0575 | 0.0324 | 0.0376 | **0.3508** | 0.1051 | 0.0221 | 0.0120 | 0.0121 | **0.0797** | 0.0661 | 0.0150 | 0.0088 | 0.0076 | **0.0596** |
| | | | (0.0343) | (0.0029) | (0.0018) | (0.0025) | (0.0729) | (0.0071) | (0.0010) | (0.0006) | (0.0008) | (0.0057) | (0.0037) | (0.0006) | (0.0004) | (0.0005) | (0.0070) |
| | Model B | Original | 0.3842 | **0.0527** | **0.0319** | 0.0475 | 0.4222 | **0.1143** | 0.0149 | **0.0107** | **0.0151** | 0.1277 | 0.1691 | **0.0099** | **0.0066** | **0.0082** | **0.3882** |
| | | | (0.0705) | (0.0036) | (0.0019) | (0.0032) | (0.0709) | (0.0160) | (0.0006) | (0.0004) | (0.0008) | (0.0234) | (0.0656) | (0.0006) | (0.0003) | (0.0004) | (0.3125) |
| | | + Res | **0.3457** | 0.0716 | 0.0373 | **0.0424** | **0.3231** | 0.1249 | 0.0293 | 0.0191 | 0.0162 | **0.0620** | **0.0620** | 0.0200 | 0.0121 | 0.0118 | 0.0455 |
| | | | (0.0365) | (0.0040) | (0.0017) | (0.0031) | (0.0515) | (0.0081) | (0.0011) | (0.0008) | (0.0011) | (0.0048) | (0.0057) | (0.0007) | (0.0005) | (0.0007) | (0.0037) |
| S2 | Model A | Original | 0.8292 | 0.0868 | **0.0619** | **0.0839** | 0.7874 | 0.2752 | **0.0275** | **0.0222** | **0.0308** | 0.2747 | 0.1704 | **0.0205** | **0.0145** | **0.0223** | **0.1670** |
| | | | (0.1170) | (0.0162) | (0.0046) | (0.0061) | (0.1149) | (0.0264) | (0.0021) | (0.0016) | (0.0020) | (0.0216) | (0.0121) | (0.0006) | (0.0006) | (0.0013) | (0.0076) |
| | | + Res | **0.7892** | **0.0801** | 0.0655 | 0.1017 | **0.7819** | **0.2290** | 0.0315 | 0.0244 | 0.0354 | **0.2551** | **0.1285** | 0.0213 | 0.0157 | 0.0231 | 0.1723 |
| | | | (0.1507) | (0.0061) | (0.0059) | (0.0060) | (0.1265) | (0.0220) | (0.0022) | (0.0017) | (0.0019) | (0.0164) | (0.0106) | (0.0015) | (0.0008) | (0.0012) | (0.0096) |
| | Model B | Original | **0.4930** | **0.0583** | **0.0493** | **0.0840** | **0.5898** | **0.1732** | **0.0257** | **0.0178** | **0.0300** | **0.1966** | 0.1323 | 0.0202 | **0.0129** | 0.0220 | 0.1358 |
| | | | (0.0742) | (0.0038) | (0.0043) | (0.0062) | (0.0712) | (0.0114) | (0.0009) | (0.0009) | (0.0018) | (0.0164) | (0.0146) | (0.0006) | (0.0006) | (0.0013) | (0.0070) |
| | | + Res | 0.5210 | 0.0683 | 0.0651 | 0.0869 | 0.7145 | 0.1847 | 0.0261 | 0.0205 | 0.0335 | 0.2108 | **0.1254** | **0.0174** | 0.0142 | **0.0204** | **0.1302** |
| | | | (0.1177) | (0.0056) | (0.0084) | (0.0054) | (0.1019) | (0.0214) | (0.0015) | (0.0008) | (0.0023) | (0.0160) | (0.0102) | (0.0010) | (0.0007) | (0.0009) | (0.0090) |
| S3 | Model A | Original | **3.0766** | **0.7564** | **0.5350** | **0.7407** | **2.8969** | 1.2870 | 0.3854 | **0.2146** | **0.3387** | 1.3495 | 0.9232 | **0.2420** | **0.1459** | **0.2413** | 0.9960 |
| | | | (0.2728) | (0.0388) | (0.0248) | (0.0330) | (0.2179) | (0.0844) | (0.0216) | (0.0051) | (0.0109) | (0.0976) | (0.0348) | (0.0082) | (0.0028) | (0.0067) | (0.0636) |
| | | + Res | 3.1729 | 0.8820 | 0.5587 | 0.8456 | 3.0478 | **1.1872** | **0.3741** | 0.2298 | 0.3626 | **1.1419** | **0.7821** | 0.2505 | 0.1708 | 0.2539 | **0.7933** |
| | | | (0.4222) | (0.0415) | (0.0139) | (0.0403) | (0.2550) | (0.0536) | (0.0106) | (0.0057) | (0.0107) | (0.0664) | (0.0382) | (0.0066) | (0.0035) | (0.0062) | (0.0389) |
| | Model B | Original | 2.8166 | **0.7558** | 0.5175 | 0.7665 | 2.2839 | 1.0061 | 0.3596 | 0.2196 | 0.3551 | 1.0990 | 0.7786 | **0.2306** | 0.1391 | 0.2380 | 0.7249 |
| | | | (0.2967) | (0.0383) | (0.0205) | (0.0509) | (0.2600) | (0.0579) | (0.0131) | (0.0054) | (0.0109) | (0.0780) | (0.0315) | (0.0075) | (0.0036) | (0.0084) | (0.0330) |
| | | + Res | **2.2566** | 0.7801 | **0.5042** | **0.7446** | **2.2770** | **0.9813** | **0.3330** | **0.2148** | **0.3146** | **0.9632** | **0.7292** | 0.2358 | 0.1554 | 0.2451 | **0.6422** |
| | | | (0.2357) | (0.0379) | (0.0171) | (0.0301) | (0.1769) | (0.0587) | (0.0103) | (0.0043) | (0.0083) | (0.0474) | (0.0358) | (0.0068) | (0.0025) | (0.0061) | (0.0267) |

Table 10: Mean squared error (MSE) and its standard error performances for Gaussian ConquerNet, scenario 1-3, model A and B with residual blocks under different sample sizes, quantile levels. "Original" means the original Gaussian ConquerNet without residual blocks, and "+Res" means the Gaussian ConquerNet with residual blocks. The MSEs are averaged over 50 independent trials. The bracket means the standard error of the MSEs over the trials.

| Gaussian | | | n=1000 | | | | | n=5000 | | | | | n=10000 | | | | |
|---|---|---|---|---|---|---|---|---|---|---|---|---|---|---|---|---|---|
| | | | τ=0.05 | τ=0.25 | τ=0.5 | τ=0.75 | τ=0.95 | τ=0.05 | τ=0.25 | τ=0.5 | τ=0.75 | τ=0.95 | τ=0.05 | τ=0.25 | τ=0.5 | τ=0.75 | τ=0.95 |
| S1 | Model A | Original | 0.4354 | **0.0383** | 0.0224 | 0.0382 | 0.5035 | 0.1124 | **0.0087** | **0.0055** | 0.0097 | 0.1081 | 0.0678 | **0.0064** | **0.0034** | **0.0055** | 0.0625 |
| | | | (0.0767) | (0.0025) | (0.0014) | (0.0030) | (0.0858) | (0.0120) | (0.0004) | (0.0003) | (0.0007) | (0.0148) | (0.0088) | (0.0005) | (0.0001) | (0.0004) | (0.0060) |
| | | + Res | **0.3787** | 0.0620 | **0.0217** | **0.0262** | **0.4817** | **0.1006** | 0.0249 | 0.0118 | **0.0090** | **0.0889** | **0.0641** | 0.0245 | 0.0122 | 0.0100 | **0.0465** |
| | | | (0.0573) | (0.0030) | (0.0014) | (0.0021) | (0.0825) | (0.0058) | (0.0011) | (0.0006) | (0.0006) | (0.0105) | (0.0037) | (0.0009) | (0.0006) | (0.0006) | (0.0044) |
| | Model B | Original | 0.4158 | **0.0665** | 0.0383 | 0.0633 | 0.5202 | **0.1172** | **0.0144** | **0.0097** | 0.0142 | 0.1066 | 0.1062 | **0.0095** | **0.0055** | **0.0086** | 0.0726 |
| | | | (0.0341) | (0.0043) | (0.0014) | (0.0024) | (0.0427) | (0.0113) | (0.0006) | (0.0004) | (0.0005) | (0.0122) | (0.0307) | (0.0005) | (0.0003) | (0.0005) | (0.0124) |
| | | + Res | **0.3525** | 0.0797 | **0.0306** | **0.0308** | **0.5173** | 0.1360 | 0.0386 | 0.0187 | **0.0131** | **0.0709** | **0.0829** | 0.0276 | 0.0156 | 0.0110 | **0.0443** |
| | | | (0.0380) | (0.0036) | (0.0018) | (0.0021) | (0.0915) | (0.0094) | (0.0016) | (0.0011) | (0.0009) | (0.0077) | (0.0052) | (0.0011) | (0.0006) | (0.0007) | (0.0044) |
| S2 | Model A | Original | 0.7994 | 0.0711 | 0.0587 | **0.0778** | 1.1466 | 0.2202 | 0.0276 | **0.0169** | **0.0273** | 0.2598 | 0.1316 | 0.0176 | **0.0128** | 0.0193 | 0.1537 |
| | | | (0.1005) | (0.0068) | (0.0088) | (0.0053) | (0.2029) | (0.0174) | (0.0025) | (0.0010) | (0.0017) | (0.0284) | (0.0091) | (0.0011) | (0.0007) | (0.0016) | (0.0102) |
| | | + Res | **0.7481** | **0.0704** | **0.0577** | 0.0906 | **0.7275** | **0.2076** | **0.0260** | 0.0223 | 0.0308 | **0.2394** | **0.1302** | **0.0152** | 0.0134 | **0.0187** | **0.1354** |
| | | | (0.1358) | (0.0082) | (0.0043) | (0.0064) | (0.0984) | (0.0219) | (0.0015) | (0.0028) | (0.0017) | (0.0315) | (0.0075) | (0.0009) | (0.0006) | (0.0010) | (0.0092) |
| | Model B | Original | 0.7367 | **0.0639** | **0.0500** | **0.0759** | 0.6273 | 0.1800 | 0.0246 | **0.0155** | **0.0245** | 0.2157 | **0.1085** | 0.0160 | **0.0119** | 0.0178 | 0.1144 |
| | | | (0.1652) | (0.0057) | (0.0049) | (0.0053) | (0.0972) | (0.0171) | (0.0020) | (0.0009) | (0.0024) | (0.0236) | (0.0087) | (0.0017) | (0.0007) | (0.0013) | (0.0078) |
| | | + Res | **0.7154** | 0.0700 | 0.0553 | 0.0827 | **0.5483** | **0.1430** | **0.0237** | 0.0195 | 0.0284 | **0.1835** | 0.1008 | **0.0152** | 0.0120 | 0.0187 | **0.1036** |
| | | | (0.1862) | (0.0083) | (0.0043) | (0.0053) | (0.1012) | (0.0134) | (0.0011) | (0.0013) | (0.0014) | (0.0138) | (0.0093) | (0.0010) | (0.0006) | (0.0011) | (0.0070) |
| S3 | Model A | Original | 2.8841 | **0.7776** | **0.4788** | **0.7056** | 3.4222 | 1.3139 | 0.3379 | **0.1993** | 0.3269 | 1.1897 | 0.8395 | **0.2040** | **0.1295** | **0.2145** | 0.7477 |
| | | | (0.3062) | (0.0746) | (0.0206) | (0.0441) | (0.3403) | (0.1159) | (0.0135) | (0.0067) | (0.0100) | (0.0783) | (0.0565) | (0.0055) | (0.0027) | (0.0055) | (0.0402) |
| | | + Res | **2.8543** | 0.9541 | 0.5828 | 0.7702 | **2.7395** | **1.1222** | **0.3339** | 0.2143 | **0.3045** | **0.9996** | **0.7556** | 0.2403 | 0.1515 | 0.2281 | **0.6931** |
| | | | (0.2822) | (0.0714) | (0.0455) | (0.0347) | (0.2494) | (0.0464) | (0.0117) | (0.0045) | (0.0084) | (0.0518) | (0.0360) | (0.0084) | (0.0027) | (0.0056) | (0.0286) |
| | Model B | Original | **2.3193** | 0.8405 | **0.5142** | **0.7543** | 2.6520 | 1.0602 | 0.4263 | 0.2304 | 0.3525 | 1.0681 | 0.8256 | 0.2518 | **0.1416** | 0.2444 | 0.7536 |
| | | | (0.2530) | (0.0573) | (0.0182) | (0.0441) | (0.5055) | (0.0791) | (0.0202) | (0.0070) | (0.0085) | (0.0822) | (0.0132) | (0.0047) | (0.0035) | (0.0085) | (0.0450) |
| | | + Res | 2.5481 | **0.7569** | 0.5322 | 0.8249 | **2.0108** | **0.9965** | **0.3196** | **0.2130** | **0.3027** | **0.8176** | **0.6785** | **0.2169** | 0.1470 | **0.2303** | **0.6120** |
| | | | (0.2880) | (0.0426) | (0.0230) | (0.0643) | (0.1294) | (0.0655) | (0.0112) | (0.0083) | (0.0108) | (0.0462) | (0.0374) | (0.0068) | (0.0035) | (0.0064) | (0.0310) |

Table 11: Mean squared error (MSE) performances for scenario 1-3, model A and B with residual blocks under different sample sizes, quantile levels, and smoothing kernels. The MSEs are averaged over 50 independent trials. The baseline model is boxed if it outperforms our ConquerNet.

| | Method | n=1000 | | | | | n=5000 | | | | | n=10000 | | | | |
|---|---|---|---|---|---|---|---|---|---|---|---|---|---|---|---|---|
| | | τ=0.05 | τ=0.25 | τ=0.5 | τ=0.75 | τ=0.95 | τ=0.05 | τ=0.25 | τ=0.5 | τ=0.75 | τ=0.95 | τ=0.05 | τ=0.25 | τ=0.5 | τ=0.75 | τ=0.95 |
| **S1 Model A** | Baseline | [0.3316] | [0.0575] | 0.0324 | 0.0376 | [0.3508] | 0.1051 | [0.0221] | 0.0120 | 0.0121 | [0.0797] | 0.0661 | [0.0150] | [0.0088] | [0.0076] | 0.0596 |
| | Gaussian | 0.3787 | 0.0620 | **0.0217** | **0.0262** | 0.4817 | **0.1006** | 0.0249 | 0.0118 | **0.0090** | 0.0889 | **0.0641** | 0.0245 | 0.0122 | 0.0100 | **0.0465** |
| | Uniform | 0.3719 | 0.0581 | **0.0218** | 0.0234 | 0.4008 | 0.1139 | 0.0255 | 0.0113 | **0.0092** | 0.0881 | 0.0689 | 0.0239 | 0.0127 | 0.0099 | 0.0483 |
| | Epanechnikov | 0.3356 | 0.0629 | **0.0225** | **0.0259** | 0.3973 | 0.1149 | 0.0259 | **0.0115** | **0.0090** | 0.0928 | **0.0627** | 0.0254 | 0.0127 | 0.0094 | **0.0434** |
| **S1 Model B** | Baseline | [0.3457] | [0.0716] | 0.0373 | 0.0424 | [0.3231] | 0.1249 | [0.0293] | 0.0191 | 0.0162 | [0.0620] | 0.0783 | [0.0200] | [0.0121] | 0.0118 | 0.0455 |
| | Gaussian | 0.3525 | 0.0797 | **0.0306** | 0.0308 | 0.5173 | 0.1360 | 0.0386 | **0.0187** | **0.0131** | 0.0709 | 0.0829 | 0.0276 | 0.0156 | **0.0110** | **0.0443** |
| | Uniform | 0.3596 | 0.0795 | **0.0322** | 0.0352 | 0.4551 | **0.1240** | 0.0389 | 0.0196 | **0.0137** | 0.0698 | **0.0769** | 0.0275 | 0.0153 | 0.0123 | **0.0440** |
| | Epanechnikov | 0.3680 | 0.0871 | **0.0348** | 0.0304 | 0.4675 | 0.1272 | 0.0379 | **0.0190** | **0.0131** | 0.0669 | 0.0810 | 0.0269 | 0.0157 | **0.0113** | **0.0379** |
| **S2 Model A** | Baseline | 0.7892 | 0.0801 | 0.0655 | 0.1017 | 0.7819 | 0.2290 | 0.0315 | 0.0244 | 0.0354 | 0.2551 | 0.1285 | 0.0213 | 0.0157 | 0.0231 | 0.1723 |
| | Gaussian | **0.7481** | **0.0704** | **0.0577** | **0.0906** | **0.7275** | **0.2076** | **0.0260** | 0.0223 | 0.0308 | **0.2394** | 0.1302 | **0.0152** | **0.0134** | **0.0187** | **0.1354** |
| | Uniform | 0.8190 | **0.0734** | **0.0647** | 0.1053 | 0.8890 | **0.2000** | 0.0240 | **0.0209** | 0.0353 | 0.2544 | **0.1136** | 0.0168 | **0.0137** | 0.0188 | **0.1168** |
| | Epanechnikov | **0.6887** | 0.0827 | 0.0716 | 0.1185 | **0.6827** | **0.1907** | 0.0265 | **0.0232** | **0.0302** | 0.2563 | 0.1227 | 0.0168 | **0.0137** | 0.0197 | **0.1233** |
| **S2 Model B** | Baseline | [0.5210] | 0.0683 | 0.0651 | 0.0869 | 0.7145 | 0.1847 | 0.0261 | 0.0205 | 0.0335 | 0.2108 | 0.1254 | 0.0174 | 0.0142 | 0.0204 | 0.1302 |
| | Gaussian | 0.7154 | 0.0700 | **0.0553** | 0.0827 | **0.5483** | **0.1430** | **0.0237** | **0.0195** | 0.0284 | **0.1835** | **0.1008** | **0.0152** | **0.0120** | **0.0187** | **0.1036** |
| | Uniform | 0.6440 | **0.0605** | 0.0523 | 0.0808 | **0.5217** | 0.1445 | **0.0215** | **0.0180** | 0.0318 | 0.1884 | 0.1043 | **0.0149** | **0.0113** | 0.0182 | 0.1074 |
| | Epanechnikov | 0.6414 | 0.0879 | **0.0580** | 0.0868 | **0.5577** | 0.1682 | 0.0235 | 0.0179 | 0.0268 | 0.1823 | 0.1049 | **0.0143** | **0.0116** | 0.0203 | 0.1154 |
| **S3 Model A** | Baseline | 3.1729 | 0.8820 | [0.5587] | 0.8456 | 3.0478 | 1.1872 | 0.3741 | 0.2298 | 0.3626 | 1.1419 | 0.7821 | 0.2505 | 0.1708 | 0.2539 | 0.7933 |
| | Gaussian | **2.8543** | 0.9541 | 0.5828 | **0.7702** | **2.7395** | **1.1222** | **0.3339** | 0.2143 | **0.3045** | **0.9996** | **0.7556** | 0.2403 | 0.1515 | 0.2281 | **0.6931** |
| | Uniform | 3.9049 | **0.8399** | 0.5665 | 0.7472 | 2.8420 | 1.1147 | 0.3521 | 0.2104 | 0.3096 | 1.1170 | 0.8147 | 0.2228 | 0.1468 | 0.2301 | 0.7264 |
| | Epanechnikov | **3.0467** | **0.8131** | 0.5638 | 0.7606 | 2.6823 | 1.2444 | 0.3520 | 0.2145 | 0.3247 | 1.2093 | **0.7679** | 0.2242 | 0.1531 | 0.2232 | 0.7798 |
| **S3 Model B** | Baseline | 2.2566 | 0.7801 | 0.5042 | 0.7446 | 2.2770 | 0.9813 | 0.3330 | 0.2148 | 0.3146 | 0.9632 | 0.7292 | 0.2358 | 0.1554 | 0.2451 | 0.6422 |
| | Gaussian | 2.5481 | **0.7569** | 0.5322 | 0.8249 | **2.0108** | 0.9965 | **0.3196** | 0.2130 | **0.3027** | 0.8176 | 0.6785 | 0.2169 | 0.1470 | **0.2303** | 0.6120 |
| | Uniform | **1.9425** | 0.8181 | **0.4824** | 0.7644 | **1.9375** | 0.9431 | 0.3394 | **0.2002** | 0.3013 | 0.8535 | 0.6307 | 0.2177 | 0.1462 | 0.2086 | 0.6415 |
| | Epanechnikov | 2.4130 | 0.7807 | 0.5403 | **0.7333** | 1.9167 | **0.8839** | 0.3262 | 0.2112 | 0.3211 | 0.9264 | 0.7278 | 0.2058 | 0.1471 | 0.2277 | **0.5911** |

To solve the problem, we let $\tau_0 < \cdots < \tau_m$ be the elements of $\Gamma$ and solve

$$\{\hat{g}_\tau\}_{\tau\in\Gamma} = \underset{\{g_\tau\}_{\tau\in\Gamma}}{\arg\min} \sum_{i=1}^{n} \ell_h(y_i - g_{\tau_0}(\mathbf{x}_i)) + \sum_{j=1}^{m}\sum_{i=1}^{n} \ell_h \left\{ y_i - g_{\tau_0}(\mathbf{x}_i) - \sum_{l=1}^{j} \log\left(1 + e^{g_{\tau_l}(\mathbf{x}_i)}\right) \right\}$$

and set

$$\hat{f}_{\tau_0}(\mathbf{x}) = \hat{g}_{\tau_0}(\mathbf{x}), \quad \text{and} \quad \hat{f}_{\tau_j}(\mathbf{x}) = \hat{g}_{\tau_0}(\mathbf{x}) + \sum_{l=1}^{j} \log\left(1 + e^{\hat{g}_{\tau_l}(\mathbf{x})}\right) \quad \text{for} \quad j = 1, \ldots, m,$$

where we recall that $\ell_h(\cdot)$ is the convolution-type smoothed quantile loss for the quantile $\tau$ in equation (4). For the simulation data, we study the MSE performance, see Table 12 below. In contrast to Table 1 in the paper, we can see that the joint estimation of multiple quantile levels performs much better for high ($\tau$=0.95) and low ($\tau$=0.05) quantiles, while the performance for $\tau = 0.25/0.5/0.75$ declines a little bit in exchange.

Table 12: Mean squared error (MSE) performances for scenario 1-3, model A and B under different sample sizes, quantile levels, and smoothing kernels. Multiple quantile levels are trained jointly under the non-crossing constraint. The MSEs are averaged over 50 independent trials.

| | Method | n=1000 | | | | | n=5000 | | | | | n=10000 | | | | |
|---|---|---|---|---|---|---|---|---|---|---|---|---|---|---|---|---|
| | | $\tau$=0.05 | $\tau$=0.25 | $\tau$=0.5 | $\tau$=0.75 | $\tau$=0.95 | $\tau$=0.05 | $\tau$=0.25 | $\tau$=0.5 | $\tau$=0.75 | $\tau$=0.95 | $\tau$=0.05 | $\tau$=0.25 | $\tau$=0.5 | $\tau$=0.75 | $\tau$=0.95 |
| **S1** Model A | Baseline | 0.1312 | 0.0514 | 0.0278 | 0.0417 | 0.1221 | 0.0494 | 0.0147 | 0.0100 | 0.0146 | 0.0397 | 0.0305 | 0.0088 | 0.0056 | 0.0078 | 0.0234 |
| | Gaussian | **0.1258** | **0.0427** | **0.0224** | **0.0395** | 0.1235 | **0.0477** | **0.0111** | **0.0070** | **0.0120** | **0.0361** | 0.0338 | **0.0076** | **0.0040** | **0.0060** | **0.0215** |
| | Uniform | **0.1212** | **0.0424** | **0.0240** | **0.0444** | 0.1331 | **0.0471** | **0.0111** | **0.0067** | **0.0113** | **0.0355** | 0.0340 | **0.0077** | **0.0039** | **0.0059** | **0.0222** |
| | Epanechnikov | **0.1204** | **0.0453** | **0.0235** | **0.0408** | 0.1289 | 0.0495 | **0.0123** | **0.0079** | 0.0128 | **0.0372** | 0.0361 | **0.0080** | **0.0043** | **0.0063** | **0.0217** |
| Model B | Baseline | 0.1911 | 0.0623 | 0.0352 | 0.0591 | 0.1821 | 0.0767 | 0.0192 | 0.0119 | 0.0191 | 0.0621 | 0.0547 | 0.0123 | 0.0074 | 0.0106 | 0.0362 |
| | Gaussian | 0.2105 | 0.0711 | 0.0378 | 0.0681 | 0.2041 | 0.0886 | **0.0178** | **0.0100** | 0.0169 | **0.0617** | 0.0674 | 0.0126 | **0.0063** | **0.0096** | **0.0362** |
| | Uniform | 0.2145 | 0.070 | 0.0378 | 0.0662 | 0.2066 | 0.0864 | **0.0171** | **0.0097** | 0.0159 | **0.0597** | 0.0711 | 0.0134 | **0.0067** | **0.0096** | 0.0375 |
| | Epanechnikov | 0.2136 | 0.073 | 0.0412 | 0.0685 | 0.2034 | 0.0883 | **0.0183** | **0.0103** | 0.0168 | **0.0605** | 0.0726 | 0.0126 | **0.0064** | 0.0099 | 0.0391 |
| **S2** Model A | Baseline | 0.1830 | 0.0829 | 0.0712 | 0.0910 | 0.2613 | 0.0650 | 0.0272 | 0.0248 | 0.0321 | 0.0888 | 0.0469 | 0.0199 | 0.0182 | 0.0234 | 0.0641 |
| | Gaussian | **0.1736** | **0.0718** | **0.0645** | 0.0948 | **0.2526** | 0.0542 | 0.0222 | 0.0202 | 0.0262 | 0.0810 | 0.0395 | 0.0140 | 0.0129 | 0.0181 | 0.0539 |
| | Uniform | **0.1669** | **0.0641** | **0.0592** | 0.0827 | 0.2362 | 0.0549 | 0.0233 | 0.0211 | 0.0283 | 0.0870 | 0.0377 | 0.0132 | 0.0122 | 0.0165 | 0.0477 |
| | Epanechnikov | 0.1901 | **0.0786** | 0.0749 | 0.1066 | 0.2634 | 0.0535 | 0.0222 | 0.0192 | 0.0267 | 0.0803 | 0.0396 | 0.0148 | 0.0130 | 0.0172 | 0.0524 |
| Model B | Baseline | 0.1358 | 0.0494 | 0.0481 | 0.0690 | 0.2026 | 0.0645 | 0.0222 | 0.0191 | 0.0276 | 0.0890 | 0.0499 | 0.0164 | 0.0143 | 0.0198 | 0.0560 |
| | Gaussian | 0.1449 | 0.0572 | 0.0537 | 0.0780 | 0.2060 | 0.0737 | **0.0207** | **0.0171** | **0.0261** | 0.0899 | **0.0480** | **0.0126** | **0.0108** | **0.0163** | **0.0540** |
| | Uniform | 0.1489 | 0.0559 | 0.0546 | 0.0835 | 0.2252 | 0.0728 | 0.0237 | **0.0190** | **0.0266** | 0.0906 | **0.0492** | **0.0132** | **0.0112** | **0.0165** | **0.0542** |
| | Epanechnikov | 0.1485 | 0.0567 | 0.0514 | 0.0800 | 0.2220 | 0.0777 | 0.0241 | 0.0208 | 0.0298 | 0.0937 | **0.0466** | **0.0135** | **0.0110** | **0.0159** | **0.0541** |
| **S3** Model A | Baseline | 0.9823 | 0.7126 | 0.6113 | 0.7058 | 1.1469 | 0.3581 | 0.2787 | 0.2524 | 0.2740 | 0.4018 | 0.2699 | 0.1865 | 0.1778 | 0.2012 | 0.2859 |
| | Gaussian | **0.8659** | **0.6814** | **0.5797** | **0.6803** | 1.0587 | **0.3368** | **0.2476** | **0.2292** | **0.2494** | **0.3801** | **0.2301** | **0.1597** | **0.1502** | **0.1654** | **0.2361** |
| | Uniform | **0.9403** | 0.7268 | 0.6202 | **0.6916** | 1.0572 | **0.3504** | **0.2589** | **0.2351** | **0.2550** | **0.3745** | **0.2297** | **0.1672** | **0.1572** | **0.1720** | **0.2371** |
| | Epanechnikov | **0.9368** | 0.7153 | **0.5725** | **0.6167** | 1.0125 | **0.3208** | **0.2507** | **0.2305** | **0.2518** | **0.3541** | **0.2305** | **0.1668** | **0.1554** | **0.1670** | **0.2398** |
| Model B | Baseline | 0.6933 | 0.5887 | 0.5255 | 0.5704 | 0.7379 | 0.2942 | 0.2617 | 0.2486 | 0.2605 | 0.3159 | 0.1990 | 0.1743 | 0.1704 | 0.1867 | 0.2247 |
| | Gaussian | 0.7480 | 0.6832 | 0.5910 | 0.6510 | 0.7794 | 0.3240 | 0.2964 | 0.2767 | 0.2867 | 0.3347 | 0.2008 | 0.1824 | 0.1748 | **0.1826** | **0.2063** |
| | Uniform | 0.7214 | 0.6326 | 0.5468 | 0.6127 | 0.8332 | 0.3078 | 0.2791 | 0.2623 | 0.2682 | 0.3108 | **0.1916** | **0.1700** | **0.1620** | **0.1698** | **0.1973** |
| | Epanechnikov | 0.8064 | 0.7124 | 0.6185 | 0.6936 | 0.8509 | 0.3292 | 0.2900 | 0.2717 | 0.2821 | 0.3409 | **0.1916** | **0.1712** | **0.1650** | **0.1725** | **0.2014** |

### B.1.4 Additional metrics and baselines

In addition, we implemented the MAE loss and the pinball loss in the experiment part. The MAE results for multiple quantile levels joint estimation are shown in Table 13. The MAE results for 5-fold cross-validation of Scenario 2 are shown in Table 14. The results for MAE show that our ConquerNet outperform the baseline networks in most cases, which remains consistent compared to the MSE results in Tables 12 and 8, respectively. The pinball loss results for real data analysis are presented in Table 15. These table results indicate the stability of our ConquerNet with respect to different loss metrics.

In Section 4, we studied the BMI data and presented the advantage of the ConquerNet for several single quantile levels. We also implement the pinball loss performance under multiple non-crossing quantile levels, see Table 15. Our ConquerNet maintain better performance under multiple quantile levels.

The BMI data are obtained from the Health and Lifestyle dataset. We retain observations without a family history of lifestyle diseases, group the samples by gender, and predict BMI using age, daily steps, sleep hours, water intake, and calories consumed. The resulting data contain 35,011 male and 35,074 female observations. The California Housing data are obtained through `fetch_california_housing` from scikit-learn. They

Table 13: Mean absolute error (MAE) performances for scenario 1-3, model A and B under different sample sizes, quantile levels, and smoothing kernels. Multiple quantile levels are trained jointly under the non-crossing constraint. The MAEs are averaged over 50 independent trials.

| | | Method | n=1000 | | | | | n=5000 | | | | | n=10000 | | | | |
|---|---|---|---|---|---|---|---|---|---|---|---|---|---|---|---|---|---|
| | | | $\tau$=0.05 | $\tau$=0.25 | $\tau$=0.5 | $\tau$=0.75 | $\tau$=0.95 | $\tau$=0.05 | $\tau$=0.25 | $\tau$=0.5 | $\tau$=0.75 | $\tau$=0.95 | $\tau$=0.05 | $\tau$=0.25 | $\tau$=0.5 | $\tau$=0.75 | $\tau$=0.95 |
| S1 | Model A | Baseline | 0.2805 | 0.1798 | 0.1310 | 0.1622 | 0.2792 | 0.1718 | 0.0960 | 0.0781 | 0.0948 | 0.1583 | 0.1314 | 0.0723 | 0.0573 | 0.0694 | 0.1202 |
| | | Gaussian | 0.2775 | 0.1645 | 0.1204 | 0.1587 | 0.2780 | 0.1678 | 0.0826 | 0.0647 | 0.0859 | 0.1516 | 0.1396 | 0.0675 | 0.0483 | 0.0607 | 0.1144 |
| | | Uniform | 0.2713 | 0.1654 | 0.1246 | 0.1671 | 0.2926 | 0.1663 | 0.0829 | 0.0633 | 0.0828 | 0.1487 | 0.1400 | 0.0679 | 0.0487 | 0.0605 | 0.1165 |
| | | Epanechnikov | 0.2707 | 0.1715 | 0.1230 | 0.1607 | 0.2870 | 0.1703 | 0.0849 | 0.0673 | 0.0883 | 0.1510 | 0.1456 | 0.0695 | 0.0507 | 0.0622 | 0.1157 |
| | Model B | Baseline | 0.3414 | 0.1974 | 0.1484 | 0.1948 | 0.3444 | 0.2151 | 0.1084 | 0.0856 | 0.1094 | 0.1983 | 0.1807 | 0.0876 | 0.0675 | 0.0816 | 0.1524 |
| | | Gaussian | 0.3552 | 0.2121 | 0.1551 | 0.2063 | 0.3620 | 0.2304 | 0.1039 | 0.0786 | 0.1032 | 0.1993 | 0.1971 | 0.0880 | 0.0617 | 0.0775 | 0.1516 |
| | | Uniform | 0.3616 | 0.2121 | 0.1546 | 0.2017 | 0.3659 | 0.2281 | 0.1016 | 0.0772 | 0.1003 | 0.1976 | 0.2027 | 0.0910 | 0.0639 | 0.0780 | 0.1550 |
| | | Epanechnikov | 0.3611 | 0.2133 | 0.1610 | 0.2073 | 0.3654 | 0.2312 | 0.1056 | 0.0797 | 0.1027 | 0.1975 | 0.2062 | 0.0876 | 0.0621 | 0.0791 | 0.1589 |
| S2 | Model A | Baseline | 0.3352 | 0.2137 | 0.1999 | 0.2265 | 0.3765 | 0.1955 | 0.1250 | 0.1182 | 0.1344 | 0.2236 | 0.1674 | 0.1060 | 0.1013 | 0.1159 | 0.1899 |
| | | Gaussian | 0.3271 | 0.1968 | 0.1882 | 0.2235 | 0.3686 | 0.1794 | 0.1141 | 0.1086 | 0.1240 | 0.2125 | 0.1534 | 0.0885 | 0.0843 | 0.0986 | 0.1724 |
| | | Uniform | 0.3256 | 0.1913 | 0.1851 | 0.2173 | 0.3611 | 0.1815 | 0.1143 | 0.1088 | 0.1256 | 0.2153 | 0.1494 | 0.0877 | 0.0832 | 0.0971 | 0.1661 |
| | | Epanechnikov | 0.3432 | 0.2054 | 0.2012 | 0.2380 | 0.3781 | 0.1798 | 0.1117 | 0.1057 | 0.1232 | 0.2117 | 0.1529 | 0.0900 | 0.0851 | 0.0987 | 0.1732 |
| | Model B | Baseline | 0.2947 | 0.1700 | 0.1680 | 0.2033 | 0.3517 | 0.1998 | 0.1162 | 0.1073 | 0.1289 | 0.2301 | 0.1733 | 0.0971 | 0.0919 | 0.1086 | 0.1814 |
| | | Gaussian | 0.3027 | 0.1820 | 0.1770 | 0.2153 | 0.3513 | 0.2124 | 0.1109 | 0.0990 | 0.1223 | 0.2294 | 0.1693 | 0.0869 | 0.0794 | 0.0977 | 0.1781 |
| | | Uniform | 0.3058 | 0.1780 | 0.1763 | 0.2184 | 0.3633 | 0.2112 | 0.1150 | 0.1022 | 0.1233 | 0.2320 | 0.1726 | 0.0890 | 0.0807 | 0.0986 | 0.1801 |
| | | Epanechnikov | 0.3035 | 0.1741 | 0.1713 | 0.2136 | 0.3596 | 0.2188 | 0.1194 | 0.1076 | 0.1304 | 0.2368 | 0.1672 | 0.0890 | 0.0801 | 0.0969 | 0.1791 |
| S3 | Model A | Baseline | 0.7794 | 0.6503 | 0.6017 | 0.6386 | 0.8088 | 0.4631 | 0.4050 | 0.3822 | 0.3952 | 0.4868 | 0.4030 | 0.3326 | 0.3220 | 0.3402 | 0.4076 |
| | | Gaussian | 0.7381 | 0.6396 | 0.5890 | 0.6228 | 0.7545 | 0.4453 | 0.3817 | 0.3651 | 0.3784 | 0.4728 | 0.3708 | 0.3071 | 0.2947 | 0.3067 | 0.3738 |
| | | Uniform | 0.7678 | 0.6559 | 0.6132 | 0.6444 | 0.7743 | 0.4575 | 0.3907 | 0.3680 | 0.3797 | 0.4688 | 0.3712 | 0.3133 | 0.3017 | 0.3134 | 0.3742 |
| | | Epanechnikov | 0.7650 | 0.6417 | 0.5790 | 0.6037 | 0.7491 | 0.4391 | 0.3849 | 0.3672 | 0.3805 | 0.4604 | 0.3700 | 0.3130 | 0.2988 | 0.3085 | 0.3771 |
| | Model B | Baseline | 0.6603 | 0.5983 | 0.5712 | 0.5957 | 0.6654 | 0.4162 | 0.3933 | 0.3816 | 0.3874 | 0.4271 | 0.3380 | 0.3167 | 0.3113 | 0.3227 | 0.3520 |
| | | Gaussian | 0.6923 | 0.6354 | 0.5995 | 0.6336 | 0.6862 | 0.4385 | 0.4147 | 0.3997 | 0.4032 | 0.4360 | 0.3422 | 0.3229 | 0.3138 | 0.3173 | 0.3404 |
| | | Uniform | 0.6805 | 0.6133 | 0.5824 | 0.6204 | 0.6978 | 0.4292 | 0.4047 | 0.3915 | 0.3917 | 0.4207 | 0.3353 | 0.3142 | 0.3030 | 0.3081 | 0.3357 |
| | | Epanechnikov | 0.7175 | 0.6433 | 0.6115 | 0.6542 | 0.7074 | 0.4433 | 0.4119 | 0.3987 | 0.4003 | 0.4336 | 0.3335 | 0.3143 | 0.3040 | 0.3074 | 0.3351 |

Table 14: Mean absolute error (MAE) performances for scenario 2, model A and B with 5-fold cross-validation under different sample sizes, quantile levels, and smoothing kernels. The MAEs are averaged over 50 independent trials.

| n | Method | Scenario 2, Model A | | | | | Scenario 2, Model B | | | | |
|---|---|---|---|---|---|---|---|---|---|---|---|
| | | $\tau$=0.05 | $\tau$=0.25 | $\tau$=0.5 | $\tau$=0.75 | $\tau$=0.95 | $\tau$=0.05 | $\tau$=0.25 | $\tau$=0.5 | $\tau$=0.75 | $\tau$=0.95 |
| 1000 | Baseline | 0.6325 | 0.2014 | 0.1820 | 0.2285 | 0.6348 | 0.5071 | 0.1994 | 0.1684 | 0.2264 | 0.5078 |
| | Gaussian | 0.5974 | 0.2043 | 0.1710 | 0.2199 | 0.6320 | 0.4969 | 0.2017 | 0.1770 | 0.2173 | 0.5303 |
| | Uniform | 0.6401 | 0.2062 | 0.1794 | 0.2024 | 0.5837 | 0.5030 | 0.1993 | 0.1694 | 0.2105 | 0.5735 |
| | Epanechnikov | 0.6348 | 0.1928 | 0.1663 | 0.2211 | 0.5642 | 0.5129 | 0.1922 | 0.1778 | 0.2142 | 0.5409 |
| 5000 | Baseline | 0.3797 | 0.1214 | 0.1158 | 0.1371 | 0.4034 | 0.2954 | 0.1237 | 0.1053 | 0.1275 | 0.3513 |
| | Gaussian | 0.3376 | 0.1220 | 0.1007 | 0.1302 | 0.4002 | 0.2905 | 0.1085 | 0.0973 | 0.1234 | 0.3410 |
| | Uniform | 0.3501 | 0.1276 | 0.1019 | 0.1249 | 0.3751 | 0.3059 | 0.1220 | 0.1020 | 0.1201 | 0.3337 |
| | Epanechnikov | 0.3513 | 0.1154 | 0.0982 | 0.1239 | 0.3680 | 0.3228 | 0.1151 | 0.0969 | 0.1297 | 0.3435 |
| 10000 | Baseline | 0.2858 | 0.1034 | 0.0951 | 0.1156 | 0.3194 | 0.2515 | 0.0949 | 0.0866 | 0.1124 | 0.2863 |
| | Gaussian | 0.2768 | 0.0937 | 0.0858 | 0.1014 | 0.2762 | 0.2373 | 0.0882 | 0.0800 | 0.1022 | 0.2774 |
| | Uniform | 0.2738 | 0.0937 | 0.0818 | 0.1064 | 0.2852 | 0.2579 | 0.0951 | 0.0802 | 0.0988 | 0.2916 |
| | Epanechnikov | 0.2744 | 0.0959 | 0.0818 | 0.1020 | 0.3157 | 0.2383 | 0.0943 | 0.0794 | 0.0974 | 0.2553 |

Table 15: Pinball loss performance for BMI (body mass index) prediction. "Single Quantile" represents that the models are trained with every single quantile level. "Multiple Quantiles" represents that the models are trained with multiple quantile levels jointly under non-crossing constraints.

| Gender | Method | Single Quantile | | | | | Multiple Quantiles | | | | |
|--------|--------|------|------|------|------|------|------|------|------|------|------|
| | | $\tau$=0.05 | $\tau$=0.25 | $\tau$=0.5 | $\tau$=0.75 | $\tau$=0.95 | $\tau$=0.05 | $\tau$=0.25 | $\tau$=0.5 | $\tau$=0.75 | $\tau$=0.95 |
| Male | Baseline | 0.5231 | 2.0638 | 2.7387 | 2.0534 | 0.5190 | 0.5218 | 2.0629 | 2.7415 | 2.0523 | 0.5198 |
| | Gaussian | **0.5221** | **2.0609** | **2.7384** | **2.0513** | **0.5189** | **0.5214** | **2.0625** | **2.7407** | **2.0520** | **0.5192** |
| | Uniform | **0.5217** | **2.0618** | 2.7403 | **2.0521** | 0.5192 | **0.5218** | **2.0619** | **2.7395** | **2.0517** | 0.5201 |
| | Epanechnikov | **0.5218** | **2.0636** | 2.7389 | **2.0518** | 0.5205 | **0.5215** | **2.0621** | **2.7399** | **2.0521** | **0.5197** |
| Female | Baseline | 0.5218 | 2.0596 | 2.7366 | 2.0555 | 0.5249 | 0.5204 | 2.0449 | 2.7291 | 2.0531 | 0.5251 |
| | Gaussian | **0.5202** | **2.0446** | **2.7323** | **2.0531** | 0.5259 | **0.5203** | 2.0450 | 2.7305 | **2.0529** | **0.5250** |
| | Uniform | **0.5211** | **2.0497** | **2.7311** | 2.0642 | 0.5251 | **0.5202** | **2.0449** | 2.7295 | **2.0526** | **0.5248** |
| | Epanechnikov | **0.5205** | **2.0454** | **2.7276** | **2.0523** | **0.5248** | 0.5212 | 2.0460 | 2.7325 | 2.0579 | 0.5290 |

contain 20,640 observations, and we predict median house value using the eight available predictors. For both datasets, 80% of the observations are used for training and the remaining 20% for testing.

Pinball loss provides a directly applicable evaluation criterion: its population expectation is minimized at the target conditional quantile. Therefore, it becomes a canonical proper scoring rule when the true conditional quantile is unobserved in real-data applications. Nevertheless, pinball loss alone does not reveal the coverage–sharpness trade-off of prediction intervals, coherence across separately fitted quantiles, or the severity of tail misses. We therefore additionally compute weighted interval score (WIS), 50% interval score, mean absolute quantile-calibration error, mean absolute interval-coverage error, quantile crossing, and conditional tail-miss distance. WIS combines the median error with the 50% and 90% interval scores, and the interval score jointly penalizes excessive width and undercoverage. The mean absolute quantile calibration error averages $|\widehat{P}(Y \leq \widehat{q}_\tau) - \tau|$ over the five fitted quantiles. The mean absolute interval-coverage error averages deviations from the nominal 50% and 90% coverage levels. The mean conditional tail-miss distance averages the lower- and upper-tail distances from the corresponding interval endpoint among observations outside the 50% or 90% interval.

For completeness, we define these diagnostics explicitly. Let $\mathcal{T} = \{0.05, 0.25, 0.5, 0.75, 0.95\}$, let $\widehat{q}_{\tau,i} = \widehat{q}_\tau(X_i)$ denote the fitted quantile for test observation $i$, and set $\widehat{C}_\tau = N_{\text{test}}^{-1} \sum_i \mathbf{1}\{Y_i \leq \widehat{q}_{\tau,i}\}$. The mean absolute quantile-calibration error is $\frac{1}{|\mathcal{T}|} \sum_{\tau \in \mathcal{T}} \left| \widehat{C}_\tau - \tau \right|$. For a central $(1-\alpha)$ prediction interval, where $\alpha \in \mathcal{A} = \{0.5, 0.1\}$, write $L_{\alpha,i} = \widehat{q}_{\alpha/2,i}$ and $U_{\alpha,i} = \widehat{q}_{1-\alpha/2,i}$. Its observation-wise interval score is $\text{IS}_{\alpha,i} = U_{\alpha,i} - L_{\alpha,i} + \frac{2}{\alpha}(L_{\alpha,i} - Y_i)_+ + \frac{2}{\alpha}(Y_i - U_{\alpha,i})_+$, and the reported interval score is its test-sample average. The weighted interval score is $\text{WIS} = \frac{1}{N_{\text{test}}(2+1/2)} \sum_{i=1}^{N_{\text{test}}} \left\{ \frac{1}{2}|Y_i - \widetilde{q}_{0.5,i}| + \sum_{\alpha \in \mathcal{A}} \frac{\alpha}{2}\text{IS}_{\alpha,i} \right\}$. The mean absolute interval-coverage error is $\frac{1}{|\mathcal{A}|} \sum_{\alpha \in \mathcal{A}} \left| \frac{1}{N_{\text{test}}} \sum_i \mathbf{1}\{L_{\alpha,i} \leq Y_i \leq U_{\alpha,i}\} - (1-\alpha) \right|$. The any-crossing rate, computed from the raw fitted quantiles, is $\frac{1}{N_{\text{test}}} \sum_i \mathbf{1}\{\exists j : \widehat{q}_{\tau_j,i} > \widehat{q}_{\tau_{j+1},i}\}$, $\tau_1 < \cdots < \tau_5$. Finally, conditional on at least one miss in each tail, define the $1-\alpha$ mean tail-miss distance $(D_{U,\alpha} + D_{L,\alpha})/2$ where $D_{L,\alpha} = \frac{\sum_i (L_{\alpha,i} - Y_i)_+}{\sum_i \mathbf{1}\{Y_i < L_{\alpha,i}\}}$ and $D_{U,\alpha} = \frac{\sum_i (Y_i - U_{\alpha,i})_+}{\sum_i \mathbf{1}\{Y_i > U_{\alpha,i}\}}$.

On BMI, all three kernels reduce WIS, the 50% interval score, the mean interval-coverage error, and both tail-miss distances; calibration is also improved by Gaussian and Epanechnikov, while Uniform is worse on this metric. All BMI methods have zero crossings. On California Housing, all three kernels reduce WIS, the 50% interval score, mean calibration error, mean coverage error, 50% tail-miss distance, and the raw crossing rate; the latter falls from 23.26% to 1.43–2.47%. Uniform and Epanechnikov also reduce the 90% tail-miss distance, whereas Gaussian is slightly higher than the baseline.

We also evaluate the empirical-coverage absolute bias $|\widehat{C}_\tau - \tau|$, where $\widehat{C}_\tau = N_{\text{test}}^{-1} \sum_i \mathbf{1}\{Y_i \leq \widehat{f}_\tau(X_i)\}$. As shown in Table 17, the Gaussian, Uniform, and Epanechnikov kernels outperform the nonsmoothed baseline

Table 16: Additional quantile-regression metrics on BMI and California Housing datasets. Lower is better. Bold entries improve on the corresponding baseline for each metric.

| Dataset | Metric | Baseline | Gaussian | Uniform | Epanechnikov |
|---|---|---|---|---|---|
| | WIS | 3.1593 | **3.1535** | **3.1573** | **3.1534** |
| | 50% interval score | 16.4646 | **16.4198** | **16.4554** | **16.4261** |
| | Mean quantile-calibration error | 0.0062 | **0.0057** | 0.0092 | **0.0033** |
| BMI | Mean interval-coverage error | 0.0073 | **0.0028** | **0.0065** | **0.0051** |
| | Any-crossing rate | 0.00% | 0.00% | 0.00% | 0.00% |
| | 50% mean tail-miss distance | 2.7759 | **2.6989** | **2.6886** | **2.6962** |
| | 90% mean tail-miss distance | 0.5688 | **0.5622** | **0.5528** | **0.5539** |
| | WIS | 0.2422 | **0.2347** | **0.2338** | **0.2355** |
| | 50% interval score | 1.2355 | **1.1858** | **1.1660** | **1.1973** |
| | Mean quantile-calibration error | 0.0348 | **0.0272** | **0.0241** | **0.0138** |
| California Housing | Mean interval-coverage error | 0.0717 | **0.0229** | **0.0460** | **0.0149** |
| | Any-crossing rate | 23.26% | **1.43%** | **1.70%** | **2.47%** |
| | 50% mean tail-miss distance | 0.3375 | **0.3070** | **0.2947** | **0.2982** |
| | 90% mean tail-miss distance | 0.2834 | 0.2863 | **0.2812** | **0.2782** |

in 80/90, 79/90, and 74/90 settings, respectively. Averaged over all settings, the absolute bias decreases from 0.0161 to 0.0124, a 23.3% relative reduction. Thus, the MSE improvements are accompanied by improved empirical quantile calibration.

Table 17: Empirical-coverage absolute bias for scenarios 1–3, Models A and B, sample sizes, quantile levels, and smoothing kernels. Entries are means over 50 independent trials.

| | Method | $n = 1000$ | | | | | $n = 5000$ | | | | | $n = 10000$ | | | | |
|---|---|---|---|---|---|---|---|---|---|---|---|---|---|---|---|---|
| | | $\tau=0.05$ | $\tau=0.25$ | $\tau=0.5$ | $\tau=0.75$ | $\tau=0.95$ | $\tau=0.05$ | $\tau=0.25$ | $\tau=0.5$ | $\tau=0.75$ | $\tau=0.95$ | $\tau=0.05$ | $\tau=0.25$ | $\tau=0.5$ | $\tau=0.75$ | $\tau=0.95$ |
| **S1** Model A | Baseline | 0.0109 | 0.0379 | 0.0468 | 0.0330 | 0.0132 | 0.0057 | 0.0236 | 0.0280 | 0.0189 | 0.0063 | 0.0043 | 0.0140 | 0.0196 | 0.0168 | 0.0046 |
| | Gaussian | **0.0098** | **0.0314** | **0.0341** | **0.0311** | **0.0099** | **0.0054** | **0.0167** | **0.0184** | **0.0141** | **0.0060** | **0.0052** | **0.0131** | **0.0127** | **0.0122** | **0.0048** |
| | Uniform | **0.0096** | **0.0346** | **0.0344** | **0.0267** | **0.0096** | 0.0068 | **0.0154** | **0.0193** | **0.0168** | **0.0060** | 0.0061 | 0.0148 | **0.0160** | **0.0125** | **0.0045** |
| | Epanechnikov | **0.0086** | **0.0363** | **0.0361** | **0.0295** | **0.0104** | **0.0047** | **0.0182** | **0.0166** | **0.0161** | 0.0063 | 0.0058 | 0.0164 | **0.0161** | **0.0120** | 0.0052 |
| Model B | Baseline | 0.0126 | 0.0408 | 0.0439 | 0.0350 | 0.0109 | 0.0058 | 0.0235 | 0.0262 | 0.0206 | 0.0057 | 0.0052 | 0.0154 | 0.0223 | 0.0166 | 0.0048 |
| | Gaussian | **0.0100** | **0.0368** | **0.0345** | **0.0307** | **0.0094** | **0.0055** | **0.0199** | **0.0208** | **0.0168** | 0.0060 | 0.0063 | 0.0190 | **0.0116** | **0.0144** | **0.0048** |
| | Uniform | **0.0110** | **0.0367** | **0.0350** | **0.0317** | **0.0105** | 0.0059 | **0.0205** | **0.0192** | **0.0172** | 0.0066 | 0.0058 | 0.0191 | **0.0146** | **0.0111** | 0.0050 |
| | Epanechnikov | **0.0099** | **0.0305** | **0.0385** | **0.0259** | **0.0106** | 0.0060 | **0.0212** | **0.0202** | **0.0169** | 0.0068 | 0.0059 | 0.0184 | **0.0135** | **0.0129** | 0.0050 |
| **S2** Model A | Baseline | 0.0136 | 0.0277 | 0.0315 | 0.0338 | 0.0162 | 0.0057 | 0.0135 | 0.0188 | 0.0132 | 0.0069 | 0.0051 | 0.0107 | 0.0148 | 0.0132 | 0.0043 |
| | Gaussian | **0.0095** | **0.0246** | **0.0389** | **0.0274** | **0.0114** | **0.0045** | **0.0106** | **0.0168** | 0.0146 | **0.0038** | **0.0034** | **0.0095** | **0.0116** | **0.0096** | **0.0036** |
| | Uniform | **0.0099** | **0.0229** | **0.0309** | **0.0271** | **0.0119** | **0.0045** | 0.0134 | **0.0167** | 0.0166 | **0.0052** | **0.0038** | **0.0091** | **0.0100** | **0.0082** | **0.0039** |
| | Epanechnikov | 0.0123 | 0.0291 | 0.0330 | **0.0316** | **0.0113** | **0.0045** | 0.0128 | **0.0214** | **0.0125** | **0.0053** | **0.0035** | **0.0089** | **0.0110** | **0.0088** | **0.0029** |
| Model B | Baseline | 0.0106 | 0.0227 | 0.0340 | 0.0328 | 0.0127 | 0.0057 | 0.0130 | 0.0158 | 0.0179 | 0.0052 | 0.0047 | 0.0123 | 0.0147 | 0.0134 | 0.0046 |
| | Gaussian | **0.0079** | **0.0210** | **0.0304** | **0.0182** | **0.0092** | **0.0048** | **0.0102** | **0.0157** | **0.0125** | **0.0030** | **0.0042** | **0.0092** | **0.0118** | **0.0099** | **0.0039** |
| | Uniform | **0.0083** | **0.0195** | **0.0292** | **0.0257** | **0.0113** | **0.0051** | 0.0138 | 0.0162 | **0.0127** | **0.0043** | **0.0045** | **0.0092** | 0.0123 | **0.0080** | **0.0035** |
| | Epanechnikov | **0.0082** | **0.0219** | **0.0288** | **0.0235** | **0.0118** | **0.0049** | 0.0139 | **0.0136** | **0.0125** | **0.0038** | **0.0040** | **0.0076** | **0.0113** | **0.0085** | **0.0029** |
| **S3** Model A | Baseline | 0.0133 | 0.0223 | 0.0329 | 0.0269 | 0.0106 | 0.0067 | 0.0157 | 0.0179 | 0.0163 | 0.0068 | 0.0051 | 0.0093 | 0.0159 | 0.0122 | 0.0037 |
| | Gaussian | **0.0097** | 0.0245 | **0.0270** | **0.0213** | **0.0082** | **0.0055** | **0.0144** | **0.0144** | 0.0146 | **0.0049** | **0.0036** | **0.0074** | **0.0104** | **0.0082** | **0.0035** |
| | Uniform | 0.0117 | **0.0211** | **0.0283** | **0.0241** | **0.0082** | **0.0051** | **0.0102** | **0.0163** | **0.0118** | **0.0052** | **0.0033** | **0.0071** | 0.0121 | **0.0087** | **0.0028** |
| | Epanechnikov | **0.0094** | 0.0231 | **0.0263** | **0.0230** | **0.0097** | **0.0048** | **0.0128** | **0.0137** | **0.0129** | **0.0058** | **0.0041** | **0.0073** | **0.0128** | **0.0087** | **0.0033** |
| Model B | Baseline | 0.0122 | 0.0258 | 0.0274 | 0.0324 | 0.0120 | 0.0057 | 0.0159 | 0.0181 | 0.0139 | 0.0063 | 0.0043 | 0.0118 | 0.0117 | 0.0115 | 0.0037 |
| | Gaussian | **0.0080** | **0.0248** | **0.0259** | **0.0240** | **0.0089** | **0.0051** | **0.0136** | **0.0137** | **0.0122** | **0.0054** | **0.0029** | **0.0068** | **0.0107** | **0.0074** | **0.0039** |
| | Uniform | **0.0083** | **0.0237** | **0.0269** | **0.0243** | **0.0100** | **0.0056** | **0.0127** | **0.0136** | **0.0124** | **0.0051** | **0.0032** | **0.0084** | **0.0105** | **0.0083** | **0.0036** |
| | Epanechnikov | **0.0087** | 0.0316 | **0.0233** | **0.0261** | **0.0076** | **0.0047** | **0.0111** | **0.0115** | **0.0116** | **0.0047** | **0.0035** | **0.0075** | **0.0098** | **0.0080** | 0.0040 |

Besides the original baseline (Nonsmooth), we have added two complementary quantile-regression baselines to the simulations in Scenarios S1–S3: quantile regression forest (QRF), a tree-based nonparametric estimator, and a Huberized quantile-loss neural network (Huber). The complete MSE results are reported in Table 18.

The original conclusion remains unchanged after adding these two baselines: the gains are not uniform in every finite-sample configuration, but become substantially more consistent as the sample size grows. Across all 90 settings in S1–S3, Gaussian, Uniform, and Epanechnikov ConquerNet have lower MSE in 56/90, 59/90, and 60/90 comparisons with Nonsmooth; 83/90, 85/90, and 84/90 comparisons with QRF; and 61/90, 61/90, and 64/90 comparisons with Huber, respectively. Thus, the original Table 1 comparison

with Nonsmooth is unchanged, while the additional results show that the conclusion is not specific to that baseline. More importantly for our sample-size claim, at $n = 10000$ the three kernels win 23/30, 27/30, and 27/30 comparisons with Nonsmooth; 30/30, 30/30, and 30/30 comparisons with QRF; and 27/30, 27/30, and 28/30 comparisons with Huber. These counts increase from 13/30, 10/30, and 10/30 against Nonsmooth, 23/30, 25/30, and 24/30 against QRF, and 10/30, 13/30, and 14/30 against Huber at small sample $n = 1000$. For the original matched comparison with Nonsmooth, the pooled counts at the fixed low and high quantiles increase particularly sharply, from 7/36 at $n = 1000$ to 22/36 at $n = 5000$ and 31/36 at $n = 10000$. Hence, adding the tree-based and alternative smooth-loss baselines preserves the original conclusion that the gains become increasingly consistent as the sample size grows.

Table 18: MSE comparison with the additional quantile-regression baselines for Scenarios S1–S3 (means over 50 trials; smaller is better). Nonsmooth denotes the original pinball-loss neural-network estimator, QRF denotes quantile regression forest, and Huber uses the Huberized quantile loss. Since QRF is independent of the neural-network architecture, its values are repeated in the two architecture blocks (Model A and B). Bold ConquerNet entries have lower MSE than at least one corresponding baseline. Superscripts $N$, $Q$, and $H$ indicate lower MSE than Nonsmooth, QRF, and Huber, respectively.

| | | Method | n=1000 | | | | | n=5000 | | | | | n=10000 | | | | |
|---|---|---|---|---|---|---|---|---|---|---|---|---|---|---|---|---|---|
| | | | $\tau{=}0.05$ | $\tau{=}0.25$ | $\tau{=}0.5$ | $\tau{=}0.75$ | $\tau{=}0.95$ | $\tau{=}0.05$ | $\tau{=}0.25$ | $\tau{=}0.5$ | $\tau{=}0.75$ | $\tau{=}0.95$ | $\tau{=}0.05$ | $\tau{=}0.25$ | $\tau{=}0.5$ | $\tau{=}0.75$ | $\tau{=}0.95$ |
| S1 | Model A | Nonsmooth | 0.3820 | 0.0402 | 0.0278 | 0.0374 | 0.3784 | 0.0996 | 0.0120 | 0.0086 | 0.0108 | 0.0939 | 0.0618 | 0.0067 | 0.0047 | 0.0063 | 0.1366 |
| | | QRF | 4.2466 | 0.0663 | 0.0386 | 0.0585 | 4.9542 | 10.5445 | 0.0529 | 0.0305 | 0.0515 | 3.3888 | 4.4791 | 0.0530 | 0.0281 | 0.0482 | 4.7380 |
| | | Huber | 0.2867 | 0.0440 | 0.0222 | 0.0368 | 0.3496 | 0.1067 | 0.0156 | 0.0091 | 0.0133 | 0.1247 | 0.0935 | 0.0108 | 0.0056 | 0.0096 | 0.0939 |
| | | Gaussian | $\mathbf{0.4354}^{Q}$ | $\mathbf{0.0383}^{NQH}$ | $\mathbf{0.0224}^{Q}$ | $\mathbf{0.0382}^{Q}$ | $\mathbf{0.5035}^{Q}$ | $\mathbf{0.1124}^{Q}$ | $\mathbf{0.0087}^{NQH}$ | $\mathbf{0.0055}^{NQH}$ | $\mathbf{0.0097}^{NQH}$ | $\mathbf{0.1081}^{QH}$ | $\mathbf{0.0678}^{QH}$ | $\mathbf{0.0064}^{NQH}$ | $\mathbf{0.0034}^{NQH}$ | $\mathbf{0.0055}^{NQH}$ | $\mathbf{0.0625}^{NQH}$ |
| | | Uniform | $\mathbf{0.4448}^{Q}$ | $\mathbf{0.0385}^{NQH}$ | $\mathbf{0.0224}^{NQ}$ | $\mathbf{0.0328}^{NQH}$ | $\mathbf{0.3721}^{NQ}$ | $\mathbf{0.1079}^{Q}$ | $\mathbf{0.0087}^{NQH}$ | $\mathbf{0.0059}^{NQH}$ | $\mathbf{0.0104}^{NQH}$ | $\mathbf{0.1312}^{Q}$ | $\mathbf{0.0789}^{QH}$ | $\mathbf{0.0058}^{NQH}$ | $\mathbf{0.0036}^{NQH}$ | $\mathbf{0.0057}^{NQH}$ | $\mathbf{0.0543}^{NQH}$ |
| | | Epanechnikov | $\mathbf{0.6733}^{Q}$ | $\mathbf{0.0390}^{NQH}$ | $\mathbf{0.0222}^{NQ}$ | $\mathbf{0.0373}^{NQ}$ | $\mathbf{0.5913}^{Q}$ | $\mathbf{0.1251}^{Q}$ | $\mathbf{0.0093}^{NQH}$ | $\mathbf{0.0055}^{NQH}$ | $\mathbf{0.0095}^{NQH}$ | $\mathbf{0.1169}^{QH}$ | $\mathbf{0.0653}^{QH}$ | $\mathbf{0.0061}^{NQH}$ | $\mathbf{0.0037}^{NQH}$ | $\mathbf{0.0053}^{NQH}$ | $\mathbf{0.0665}^{NQH}$ |
| | Model B | Nonsmooth | 0.3842 | 0.0527 | 0.0319 | 0.0475 | 0.4222 | 0.1143 | 0.0149 | 0.0107 | 0.0151 | 0.1277 | 0.1691 | 0.0099 | 0.0066 | 0.0082 | 0.3882 |
| | | QRF | 4.2466 | 0.0663 | 0.0386 | 0.0585 | 4.9542 | 10.5445 | 0.0529 | 0.0305 | 0.0515 | 3.3888 | 4.4791 | 0.0530 | 0.0281 | 0.0482 | 4.7380 |
| | | Huber | 0.3393 | 0.0458 | 0.0326 | 0.0448 | 0.4705 | 0.1090 | 0.0193 | 0.0105 | 0.0152 | 0.1155 | 0.0900 | 0.0120 | 0.0069 | 0.0106 | 0.0828 |
| | | Gaussian | $\mathbf{0.4158}^{Q}$ | 0.0665 | $\mathbf{0.0383}^{Q}$ | 0.0633 | $\mathbf{0.5202}^{Q}$ | $\mathbf{0.1172}^{Q}$ | $\mathbf{0.0144}^{NQH}$ | $\mathbf{0.0097}^{NQH}$ | $\mathbf{0.0142}^{NQH}$ | $\mathbf{0.1066}^{NQH}$ | $\mathbf{0.1062}^{NQ}$ | $\mathbf{0.0095}^{NQH}$ | $\mathbf{0.0055}^{NQH}$ | $\mathbf{0.0086}^{QH}$ | $\mathbf{0.0726}^{NQH}$ |
| | | Uniform | $\mathbf{0.5652}^{Q}$ | $\mathbf{0.0612}^{Q}$ | $\mathbf{0.0378}^{Q}$ | 0.0614 | $\mathbf{0.5685}^{Q}$ | $\mathbf{0.1033}^{NQH}$ | $\mathbf{0.0145}^{NQH}$ | $\mathbf{0.0091}^{NQH}$ | $\mathbf{0.0154}^{Q}$ | $\mathbf{0.1252}^{NQ}$ | $\mathbf{0.0974}^{NQ}$ | $\mathbf{0.0093}^{NQH}$ | $\mathbf{0.0055}^{NQH}$ | $\mathbf{0.0080}^{NQH}$ | $\mathbf{0.0736}^{NQH}$ |
| | | Epanechnikov | $\mathbf{0.4275}^{Q}$ | $\mathbf{0.0582}^{Q}$ | $\mathbf{0.0383}^{Q}$ | 0.0626 | $\mathbf{0.6650}^{Q}$ | $\mathbf{0.1115}^{NQ}$ | $\mathbf{0.0146}^{NQH}$ | $\mathbf{0.0094}^{NQH}$ | $\mathbf{0.0145}^{NQH}$ | $\mathbf{0.1162}^{NQ}$ | $\mathbf{0.1294}^{NQ}$ | $\mathbf{0.0089}^{NQH}$ | $\mathbf{0.0057}^{NQH}$ | $\mathbf{0.0092}^{QH}$ | $\mathbf{0.0663}^{NQH}$ |
| S2 | Model A | Nonsmooth | 0.8292 | 0.0868 | 0.0619 | 0.0839 | 0.7874 | 0.2752 | 0.0275 | 0.0222 | 0.0308 | 0.2747 | 0.1704 | 0.0205 | 0.0145 | 0.0223 | 0.1670 |
| | | QRF | 1.2012 | 0.0892 | 0.0683 | 0.1079 | 1.0185 | 0.7314 | 0.0667 | 0.0489 | 0.0749 | 0.6401 | 0.6289 | 0.0588 | 0.0443 | 0.0665 | 0.6294 |
| | | Huber | 0.7087 | 0.1019 | 0.0597 | 0.1062 | 0.9352 | 0.2953 | 0.0423 | 0.0201 | 0.0475 | 0.3395 | 0.2191 | 0.0333 | 0.0157 | 0.0352 | 0.2128 |
| | | Gaussian | $\mathbf{0.7994}^{NQ}$ | $\mathbf{0.0711}^{NQH}$ | $\mathbf{0.0587}^{NQH}$ | $\mathbf{0.0778}^{NQH}$ | 1.1466 | $\mathbf{0.2202}^{NQH}$ | $\mathbf{0.0276}^{QH}$ | $\mathbf{0.0169}^{NQH}$ | $\mathbf{0.0273}^{NQH}$ | $\mathbf{0.2598}^{NQH}$ | $\mathbf{0.1316}^{NQH}$ | $\mathbf{0.0176}^{NQH}$ | $\mathbf{0.0128}^{NQ}$ | $\mathbf{0.0193}^{NQH}$ | $\mathbf{0.1537}^{NQH}$ |
| | | Uniform | $\mathbf{0.8892}^{Q}$ | $\mathbf{0.0721}^{NQH}$ | $\mathbf{0.0471}^{NQH}$ | $\mathbf{0.0964}^{QH}$ | $\mathbf{0.8566}^{QH}$ | $\mathbf{0.2308}^{NQH}$ | $\mathbf{0.0286}^{QH}$ | $\mathbf{0.0181}^{NQH}$ | $\mathbf{0.0275}^{NQH}$ | $\mathbf{0.2586}^{NQH}$ | $\mathbf{0.1457}^{NQH}$ | $\mathbf{0.0180}^{NQH}$ | $\mathbf{0.0128}^{NQH}$ | $\mathbf{0.0191}^{NQH}$ | $\mathbf{0.1467}^{NQH}$ |
| | | Epanechnikov | $\mathbf{0.9192}^{Q}$ | $\mathbf{0.0787}^{NQH}$ | $\mathbf{0.0522}^{NQH}$ | $\mathbf{0.1048}^{QH}$ | $\mathbf{0.9074}^{QH}$ | $\mathbf{0.2415}^{NQH}$ | $\mathbf{0.0284}^{QH}$ | $\mathbf{0.0177}^{NQH}$ | $\mathbf{0.0265}^{NQH}$ | $\mathbf{0.2492}^{NQH}$ | $\mathbf{0.1430}^{NQH}$ | $\mathbf{0.0162}^{NQH}$ | $\mathbf{0.0126}^{NQH}$ | $\mathbf{0.0191}^{NQH}$ | $\mathbf{0.1388}^{NQH}$ |
| | Model B | Nonsmooth | 0.4930 | 0.0583 | 0.0493 | 0.0840 | 0.5898 | 0.1732 | 0.0257 | 0.0178 | 0.0300 | 0.1966 | 0.1323 | 0.0202 | 0.0129 | 0.0220 | 0.1358 |
| | | QRF | 1.2012 | 0.0892 | 0.0683 | 0.1079 | 1.0185 | 0.7314 | 0.0667 | 0.0489 | 0.0749 | 0.6401 | 0.6289 | 0.0588 | 0.0443 | 0.0665 | 0.6294 |
| | | Huber | 0.5992 | 0.0811 | 0.0490 | 0.1077 | 0.5570 | 0.2344 | 0.0396 | 0.0180 | 0.0415 | 0.2608 | 0.1940 | 0.0306 | 0.0134 | 0.0334 | 0.1889 |
| | | Gaussian | $\mathbf{0.7367}^{Q}$ | $\mathbf{0.0639}^{QH}$ | $\mathbf{0.0500}^{Q}$ | $\mathbf{0.0759}^{NQH}$ | $\mathbf{0.6273}^{Q}$ | $\mathbf{0.1800}^{QH}$ | $\mathbf{0.0246}^{NQH}$ | $\mathbf{0.0155}^{NQH}$ | $\mathbf{0.0245}^{NQH}$ | $\mathbf{0.2157}^{QH}$ | $\mathbf{0.1085}^{NQ}$ | $\mathbf{0.0160}^{NQH}$ | $\mathbf{0.0119}^{NQH}$ | $\mathbf{0.0178}^{NQH}$ | $\mathbf{0.1144}^{NQH}$ |
| | | Uniform | $\mathbf{0.4515}^{NQH}$ | $\mathbf{0.0717}^{QH}$ | $\mathbf{0.0503}^{Q}$ | $\mathbf{0.0935}^{QH}$ | $\mathbf{0.6034}^{Q}$ | $\mathbf{0.1706}^{NQH}$ | $\mathbf{0.0216}^{NQH}$ | $\mathbf{0.0176}^{NQH}$ | $\mathbf{0.0241}^{NQH}$ | $\mathbf{0.2239}^{QH}$ | $\mathbf{0.1172}^{NQH}$ | $\mathbf{0.0151}^{NQH}$ | $\mathbf{0.0118}^{NQH}$ | $\mathbf{0.0175}^{NQH}$ | $\mathbf{0.1342}^{NQH}$ |
| | | Epanechnikov | $\mathbf{0.5800}^{QH}$ | $\mathbf{0.0723}^{QH}$ | $\mathbf{0.0546}^{Q}$ | $\mathbf{0.0819}^{NQH}$ | $\mathbf{0.7484}^{Q}$ | $\mathbf{0.2151}^{QH}$ | $\mathbf{0.0263}^{QH}$ | $\mathbf{0.0159}^{NQH}$ | $\mathbf{0.0289}^{NQH}$ | $\mathbf{0.1802}^{NQH}$ | $\mathbf{0.1038}^{NQH}$ | $\mathbf{0.0146}^{NQH}$ | $\mathbf{0.0107}^{NQH}$ | $\mathbf{0.0173}^{NQH}$ | $\mathbf{0.1302}^{NQH}$ |
| S3 | Model A | Nonsmooth | 3.0766 | 0.7564 | 0.5350 | 0.7407 | 2.8969 | 1.2870 | 0.3854 | 0.2146 | 0.3387 | 1.3495 | 0.9232 | 0.2420 | 0.1459 | 0.2413 | 0.9960 |
| | | QRF | 2.5323 | 0.8312 | 0.6050 | 0.8245 | 2.4859 | 1.7064 | 0.5855 | 0.4149 | 0.5844 | 1.7286 | 1.4937 | 0.5158 | 0.3589 | 0.5141 | 1.4810 |
| | | Huber | 3.0916 | 0.7454 | 0.5762 | 0.8738 | 3.3923 | 1.3614 | 0.3267 | 0.2202 | 0.3386 | 1.2768 | 1.0265 | 0.2406 | 0.1536 | 0.2477 | 0.9829 |
| | | Gaussian | $\mathbf{2.8841}^{NH}$ | $\mathbf{0.7776}^{Q}$ | $\mathbf{0.4788}^{NQH}$ | $\mathbf{0.7056}^{NQH}$ | 3.4222 | $\mathbf{1.3139}^{QH}$ | $\mathbf{0.3379}^{NQ}$ | $\mathbf{0.1993}^{NQH}$ | $\mathbf{0.3269}^{NQH}$ | $\mathbf{1.1897}^{NQH}$ | $\mathbf{0.8395}^{NQH}$ | $\mathbf{0.2040}^{NQH}$ | $\mathbf{0.1295}^{NQH}$ | $\mathbf{0.2145}^{NQH}$ | $\mathbf{0.7477}^{NQH}$ |
| | | Uniform | 3.3730 | $\mathbf{0.7735}^{Q}$ | $\mathbf{0.4876}^{NQH}$ | $\mathbf{0.7832}^{QH}$ | $\mathbf{3.3576}^{H}$ | $\mathbf{1.1577}^{NQH}$ | $\mathbf{0.3449}^{NQ}$ | $\mathbf{0.1912}^{NQH}$ | $\mathbf{0.3082}^{NQH}$ | $\mathbf{1.2392}^{NQH}$ | $\mathbf{0.8790}^{NQH}$ | $\mathbf{0.2064}^{NQH}$ | $\mathbf{0.1300}^{NQH}$ | $\mathbf{0.2146}^{NQH}$ | $\mathbf{0.7824}^{NQH}$ |
| | | Epanechnikov | 3.4479 | $\mathbf{0.7308}^{NQH}$ | $\mathbf{0.4679}^{NQH}$ | $\mathbf{0.8153}^{QH}$ | $\mathbf{3.3156}^{H}$ | $\mathbf{1.1229}^{NQH}$ | $\mathbf{0.3615}^{NQ}$ | $\mathbf{0.1980}^{NQH}$ | $\mathbf{0.3112}^{NQH}$ | $\mathbf{1.3312}^{NQ}$ | $\mathbf{0.8230}^{NQH}$ | $\mathbf{0.2129}^{NQH}$ | $\mathbf{0.1349}^{NQH}$ | $\mathbf{0.2206}^{NQH}$ | $\mathbf{0.7998}^{NQH}$ |
| | Model B | Nonsmooth | 2.8166 | 0.7558 | 0.5175 | 0.7665 | 2.8839 | 1.0061 | 0.3596 | 0.2196 | 0.3551 | 1.0990 | 0.7786 | 0.2306 | 0.1391 | 0.2380 | 0.7249 |
| | | QRF | 2.5323 | 0.8312 | 0.6050 | 0.8245 | 2.4859 | 1.7064 | 0.5855 | 0.4149 | 0.5844 | 1.7286 | 1.4937 | 0.5158 | 0.3589 | 0.5141 | 1.4810 |
| | | Huber | 2.0365 | 0.6887 | 0.5285 | 0.7115 | 2.2202 | 1.1677 | 0.3329 | 0.2195 | 0.3264 | 1.1466 | 0.9326 | 0.2292 | 0.1469 | 0.2400 | 0.9807 |
| | | Gaussian | $\mathbf{2.3193}^{NQ}$ | 0.8405 | $\mathbf{0.5142}^{NQ}$ | $\mathbf{0.7543}^{NQ}$ | 2.6520 | $\mathbf{1.0602}^{QH}$ | $\mathbf{0.4263}^{Q}$ | $\mathbf{0.2304}^{Q}$ | $\mathbf{0.3525}^{NQ}$ | $\mathbf{1.0681}^{NQH}$ | $\mathbf{0.8256}^{QH}$ | $\mathbf{0.2518}^{Q}$ | $\mathbf{0.1416}^{QH}$ | $\mathbf{0.2444}^{Q}$ | $\mathbf{0.7536}^{QH}$ |
| | | Uniform | $\mathbf{2.1946}^{NQ}$ | 0.8316 | $\mathbf{0.5193}^{QH}$ | $\mathbf{0.7352}^{NQ}$ | 2.5765 | $\mathbf{1.2056}^{Q}$ | $\mathbf{0.4038}^{Q}$ | $\mathbf{0.2320}^{Q}$ | $\mathbf{0.3373}^{NQ}$ | $\mathbf{1.0528}^{NQH}$ | $\mathbf{0.7167}^{NQH}$ | $\mathbf{0.2409}^{Q}$ | $\mathbf{0.1372}^{NQH}$ | $\mathbf{0.2493}^{Q}$ | $\mathbf{0.6920}^{NQH}$ |
| | | Epanechnikov | 2.9702 | 0.9008 | $\mathbf{0.4892}^{NQH}$ | 0.8702 | $\mathbf{2.1331}^{NQH}$ | $\mathbf{1.0053}^{NQH}$ | $\mathbf{0.3750}^{Q}$ | $\mathbf{0.2297}^{Q}$ | $\mathbf{0.3473}^{NQ}$ | $\mathbf{1.0895}^{NQH}$ | $\mathbf{0.7502}^{NQH}$ | $\mathbf{0.2454}^{Q}$ | $\mathbf{0.1390}^{NQH}$ | $\mathbf{0.2315}^{NQH}$ | $\mathbf{0.6683}^{NQH}$ |

### B.2 Plots of MSEs by sample sizes

We study the plot of MSEs as a function of sample sizes to directly corroborate Theorem 3.2. We plotted the curves of log MSEs by log sample sizes. For scenario 2 and model A, we trained the baseline model and the ConquerNet with the Gaussian smoothing kernel. We took sample sizes of $\{1000, 3000, 5000, 7000, 10000\}$ and evaluated the mean MSEs of test sets over 50 trials. The plots are shown in Figure 3. The curves have linear shapes, and in equation (15) of Theorem 3.2, the log MSE also has a nearly linear upper bound with respect to log sample size, which corroborates the theoretical results in Theorem 3.2. Besides, the slopes of both baseline and ConquerNet are larger than $-1$, which is the asymptotic slope of the minimax rate $n^{-2s/(2s+d)}$. We also fit linear models for the curves in the figure and find that the slope of the ConquerNet is smaller than that of the baseline models, indicating that our upper bound is better than the baseline from the simulation.

### B.3 Plots of training time in Table 1

We also record the training time for each setting and for 50 trials. We can conclude that our ConquerNet is generally faster than the baseline, with only 80% of the training time consumed, and is stable enough.

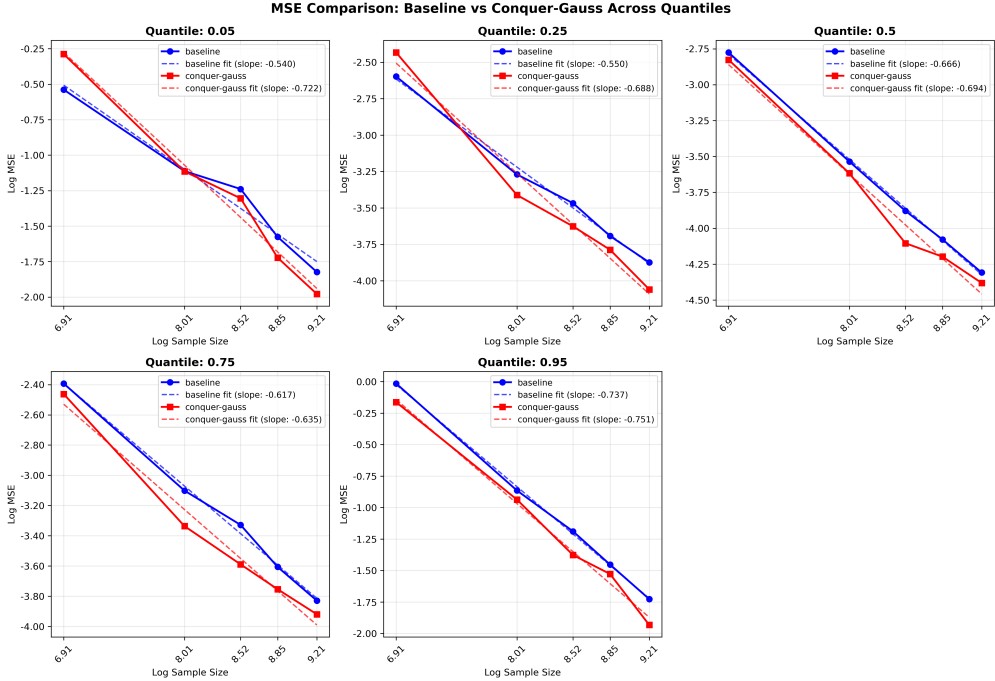

Figure 3: Plots of log MSEs by log sample sizes for scenario 2, model A. The red lines represent the ConquerNet smoothed by the Gaussian kernel with $h = 0.1$. The blue lines represent the baseline network. The solid lines with points represent the line charts of simulation results. The dashed lines represent the fitted linear models. The MSEs are averaged over 50 independent trials.

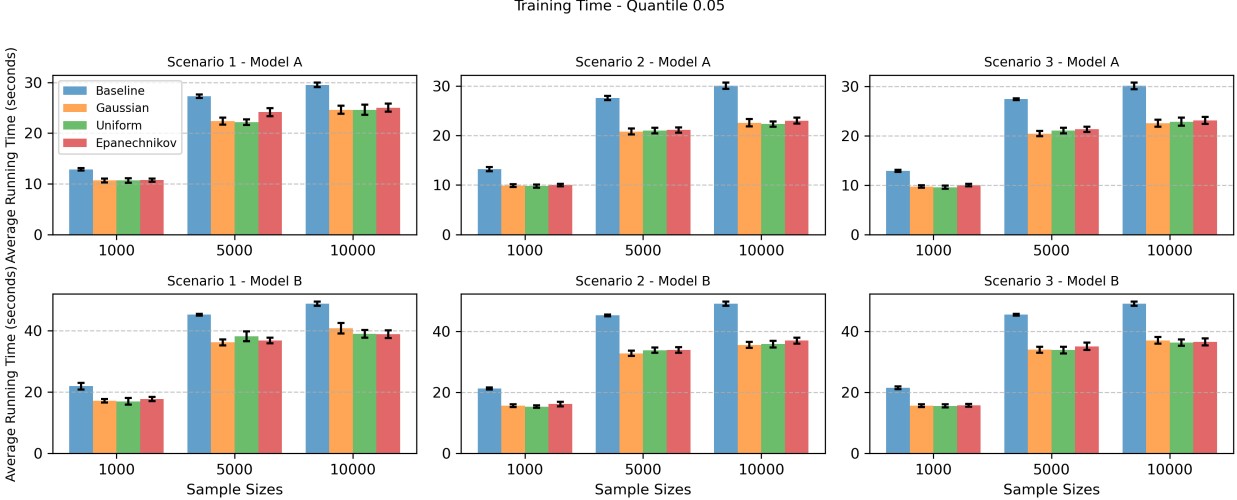

Figure 4: Bar chart with error bars with average training time over 50 trials under quantile level $\tau = 0.05$. The error bars represent 95% confidence intervals of training time for each setting.

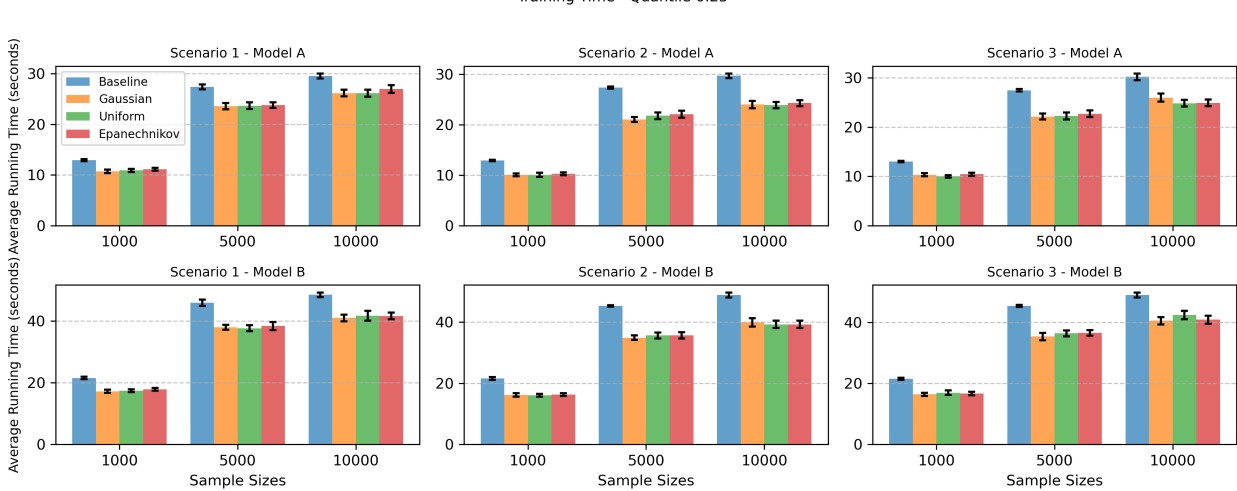

Figure 5: Bar chart with error bars with average training time over 50 trials under quantile level $\tau = 0.25$. The error bars represent 95% confidence intervals of training time for each setting.

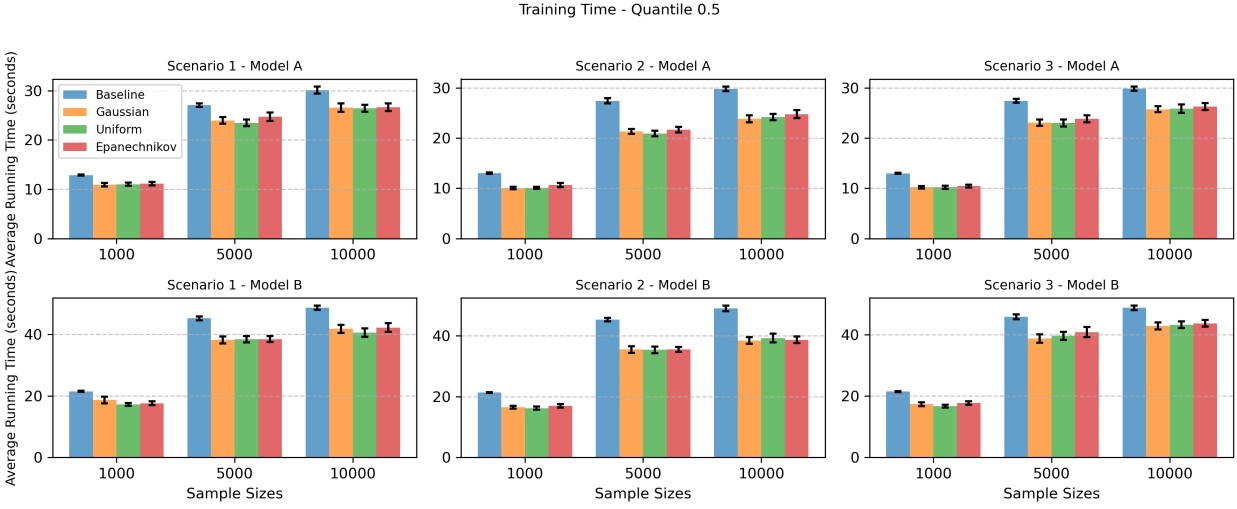

Figure 6: Bar chart with error bars with average training time over 50 trials under quantile level $\tau = 0.5$. The error bars represent 95% confidence intervals of training time for each setting.

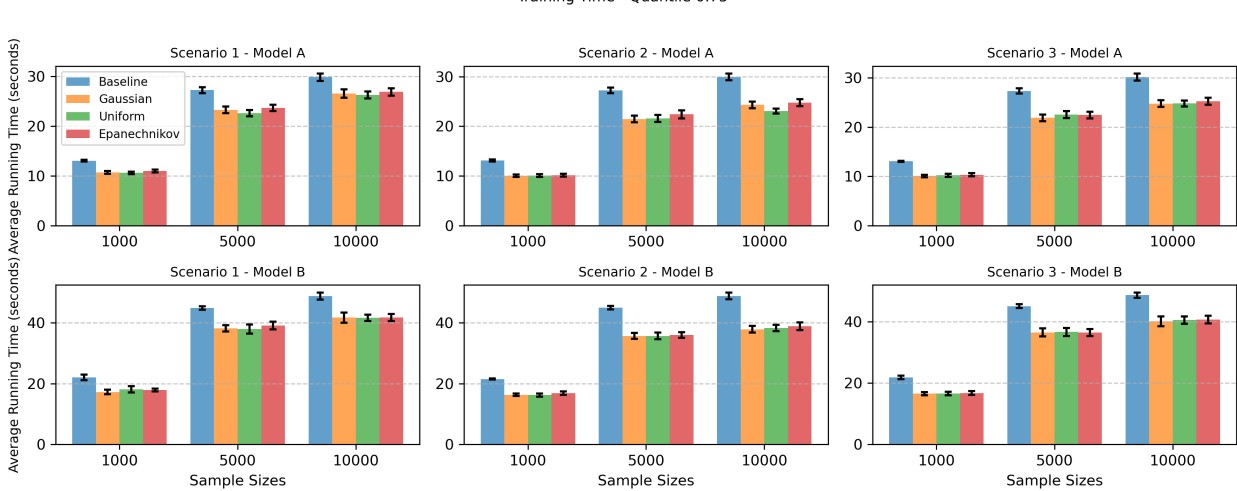

Figure 7: Bar chart with error bars with average training time over 50 trials under quantile level $\tau = 0.75$. The error bars represent 95% confidence intervals of training time for each setting.

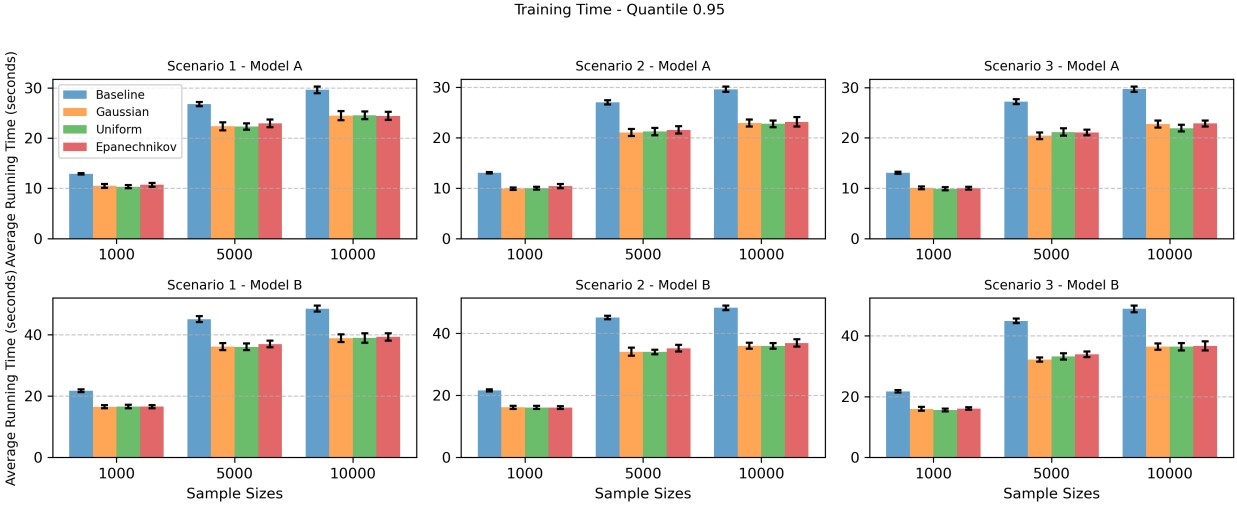

Figure 8: Bar chart with error bars with average training time over 50 trials under quantile level $\tau = 0.95$. The error bars represent 95% confidence intervals of training time for each setting.

Figures 4-8 show that our ConquerNet requires 20% less training time compared to the baseline method. Within the same scenario, model and sample size, the training times between 3 different kernels and between 5 different quantile levels are close to each other. We can also conclude that shallower networks have shorter training times when the number of parameters is similar.

