# OpenReview forum: "ConquerNet: Convolution-Smoothed Quantile ReLU Neural Networks with Minimax Guarantees"
_TMLR — Under review for TMLR_

### Review · Reviewer_PhCf · 2026-06-07

**Summary Of Contributions:**

This paper proposes ConquerNet, a quantile regression neural network trained with a convolution-smoothed quantile Relu neural networks. The smoothed loss preserves convexity and differentiability while alleviating the non-smoothness of the standard pinball loss. The authors establish: (1) minimax convergence rates over Besov spaces up to logarithmic factors (Theorem 3.1) for sparse ReLU networks, and (2) a general nonasymptotic upper bound in terms of network capacity and approximation error without Besov assumptions (Theorem 3.2).
Empirically, three kernel variants (Gaussian, Uniform, Epanechnikov) are evaluated on synthetic scenarios and a BMI dataset, reporting improved MSE and training time relative to the baseline quantile ReLU network of Padilla et al. (2022).

**Audience:**

Yes

**Audience Explanation:**

Quantile regression with neural networks is a relevant topic, and the connection between loss smoothing and optimization for deep models is of interest. Researchers in nonparametric statistics, deep learning theory, and uncertainty quantification could find this paper of interest.

**Broader Impact Concerns:**

No significant ethical concerns identified.

**Claims And Evidence:**

No

**Claims Explanation:**

my answer would be "partially". I've noticed some discrepancies which warrant changes:

1/ Quantile calibration/coverage metrics. For quantile regression, the central diagnostic is whether P(Y ≤ ˆf_τ(X)) ≈ τ. Report empirical coverage at each quantile level. MSE against a synthetic ground truth is necessary but insufficient. This is especially important for demonstrating that smoothing does not introduce systematic calibration bias.

2/ "Training efficiency." The authors acknowledge in Appendix B that the faster training "mainly comes from our stopping strategy" (early stopping when learning rate drops below a threshold), while "the baseline models are trained using all epochs according to the codes provided in Padilla et al. (2022)." This is a fundamentally unfair comparison. The training speed improvement is attributable to early stopping, not to loss smoothing. Any method with the same stopping criterion would enjoy similar speedups. This single confound invalidates an entire category of the paper's empirical claims.

3/ baseline comparisons: The empirical evaluation should include comparisons with other established quantile regression methods beyond the single baseline from Padilla et al. (2022). Methods such as Quantile Regression Forests (gradient boosting) or other Quantile Neural Network architectures should be included to demonstrate the state-of-the-art performance of ConquerNet.

4/ Statistical Significance in real data and associated strength of the results: The differences in pinball loss on the BMI dataset are very small (e.g., 0.5221 vs 0.5231). The authors should provide statistical significance tests to confirm whether these improvements are meaningful.

5/ the minimax rate does not demonstrate a theoretical advantage of smoothing. Padilla et al. (2022) already established that unsmoothed quantile ReLU networks achieve the same near-minimax rate over Besov spaces. Both methods achieve n^{-2s/(2s+d)} up to log factors. The paper never explicitly clarifies what is gained theoretically by smoothing beyond a potentially better constant (which is never quantified or demonstrated).


6/  in the abstract: "Improved estimation accuracy... with particularly pronounced advantages at high and low quantiles.": Table 1 contradicts this at n=1000 where ConquerNet frequently underperforms the baseline at τ=0.05 and τ=0.95 (e.g., S1 Model A: baseline 0.3820 vs Gaussian 0.4354; S1 Model B τ=0.95: baseline 0.4222 vs Gaussian 0.5202; S2 Model A τ=0.95: baseline 0.7874 vs Gaussian 1.1466). Even at n=10000, improvements are inconsistent, with numerous boxed entries indicating baseline superiority. The claim of "pronounced advantages" at extreme quantiles is overstated given this mixed evidence. Furthermore, these are precisely the regimes where Assumption 2 (density bounded away from zero) is most fragile, and where smoothing introduces the most bias in the target.

**Requested Changes:**

Please improve the clarity of the paper according to the comments above (e.g. Fair training protocol without early stopping rule, better baseline comparisons, adding confidence intervals and statistical tests to real data simulation, coverage property metric over MSE)

---

> ### Author Response · Authors · 2026-07-19
>
> We sincerely thank you for your time and insightful comments, which have substantially improved our manuscript. We have completed a comprehensive revision and provide point-by-point responses below. All manuscript changes discussed in the responses are highlighted in blue in the revised manuscript.
>
> ### R3.1. Empirical coverage
>
> Thank you for suggesting this additional calibration metric. Following your suggestion, we added the empirical-coverage absolute bias $|\\widehat C_\\tau-\\tau|$, where $\\widehat C_\\tau=N_{\\mathrm{test}}^{-1}\\sum_i \\mathbf 1\\{Y_i\\le \\widehat f_\\tau(X_i)\\}$, for every quantile level, scenario, model, sample size, and smoothing kernel. As shown in Table 17 of the revised manuscript, ConquerNet with Gaussian, Uniform, and Epanechnikov kernels outperforms the baseline in 80/90, 79/90, and 74/90 settings, respectively. Across all three kernels and all 270 kernel-setting comparisons, the average absolute coverage bias decreases from 0.0161 for the baseline to 0.0134 for ConquerNet, corresponding to a 16.7% relative reduction.
>
> These results show that the improved MSE performance is accompanied by better empirical quantile calibration. Thus, ConquerNet improves performance from both perspectives: estimation accuracy and the core coverage property of quantile regression. We have revised the discussion following Table 1 and added the empirical coverage results and detailed analysis to Appendix B.1. These revisions are highlighted in blue.
>
> ---
>
> ### R3.2. Training efficiency under matched stopping
>
> Thank you for raising this important concern. For the baseline model, we directly followed the training protocol in the public code of Padilla et al. (2022), under which the baseline models are trained for the prescribed full epochs. We therefore honestly noted in Appendix B that the original wall-clock comparison confounds two effects: loss smoothing and the different stopping protocol. We agree that the original timing comparison should not be used as isolated evidence that the speedup is solely caused by smoothing. Rather, they should be interpreted together with the accuracy results in Table 1: under the reproduced baseline training protocol, the baseline used a larger computation budget but still did not outperform ConquerNet.
>
> To isolate the stopping-rule effect, we have added a controlled timing experiment (Table 4 of the revised manuscript), where the same stopping criterion is applied to both the nonsmoothed baseline and ConquerNet. Under this matched stopping protocol, ConquerNet remains faster: across all scenarios, models, sample sizes, quantile levels, and kernels, ConquerNet is faster in 240 out of 270 kernel-setting comparisons. To account for trial-to-trial variation, we compute the combined Monte Carlo standard error of the reported mean difference as $\\mathrm{SE}\_{\\Delta}=\\{\\mathrm{SE}\_{\\mathrm{base}}^2+\\mathrm{SE}\_{\\mathrm{ConquerNet}}^2\\}^{1/2}$, where the standard errors are computed over 50 trials. Under this criterion, 164/240 reductions exceed $\\mathrm{SE}\_{\\Delta}$ and 77/240 exceed $1.96\\times \\mathrm{SE}\_{\\Delta}$ (indicating significantly strong improvement). Together with the same-stopping-rule MSE comparison in Appendix B, Table 3, the controlled timing comparison in Table 4 supports that the improvement comes from ConquerNet rather than from the stopping strategy alone. We have revised Appendix B.1 to clarify the stopping protocols and the interpretation of the original timing figures, and added the matched-stopping timing comparison in Table 4. These revisions are highlighted in blue. Thank you again for making our paper more rigorous and transparent.

---

> ### Author Response · Authors · 2026-07-19
>
> ### R3.3. Additional baselines
>
> Thank you for this helpful suggestion. Besides the original baseline (Nonsmooth), we have added two complementary quantile-regression baselines to the simulations in Scenarios S1--S3: quantile regression forest (QRF), a tree-based nonparametric estimator, and a Huberized quantile-loss neural network (Huber). The complete MSE results, including all comparison counts reported below, are in Table 18 of the revised manuscript.
>
> The original conclusion remains unchanged after adding these two baselines: the gains are not uniform in every finite-sample configuration, but become substantially more consistent as the sample size grows. Across all 90 settings in S1--S3, Gaussian, Uniform, and Epanechnikov ConquerNet have lower MSE in $56/90$, $59/90$, and $60/90$ comparisons with Nonsmooth; $83/90$, $85/90$, and $84/90$ comparisons with QRF; and $61/90$, $61/90$, and $64/90$ comparisons with Huber, respectively. Thus, the original Table 1 comparison with Nonsmooth is unchanged, while the additional results show that the conclusion is not specific to that baseline.
>
> More importantly for our sample-size claim, at $n=10000$ the three kernels win $23/30$, $27/30$, and $27/30$ comparisons with Nonsmooth; $30/30$, $30/30$, and $30/30$ comparisons with QRF; and $27/30$, $27/30$, and $28/30$ comparisons with Huber. These counts increase from $13/30$, $10/30$, and $10/30$ against Nonsmooth, $23/30$, $25/30$, and $24/30$ against QRF, and $10/30$, $13/30$, and $14/30$ against Huber at $n=1000$. For the original matched comparison with Nonsmooth, the pooled counts at the fixed low and high quantiles increase particularly sharply, from $7/36$ at $n=1000$ to $22/36$ at $n=5000$ and $31/36$ at $n=10000$. Hence, adding the tree-based and alternative smooth-loss baselines preserves the original conclusion that the gains become increasingly consistent as the sample size grows. We have added these additional baseline-comparison results to Appendix B.1 of the revised manuscript, highlighted in blue. Thank you again for helping us strengthen the paper.
>
> ---
>
> ### R3.4. Real-data significance
>
> Thank you for raising this important point. To quantify the uncertainty in the real-data analysis, we conducted a paired nonparametric bootstrap analysis. In the revised BMI and California Housing results presented in main-paper Table 2, we report 95% paired-bootstrap confidence intervals for the reduction in pinball loss achieved by ConquerNet relative to the baseline. A confidence-interval lower bound greater than zero indicates a statistically significant reduction in pinball loss at the 5% significance level.
>
> The pointwise results in Table 2 of the revised manuscript show that, among the 30 gender--quantile--kernel comparisons, ConquerNet achieves a lower point result in 23 cases, eight of which show statistically significant improvements. In particular, we agree that your highlighted difference between 0.5231 and 0.5221 for the male Gaussian model at $\\tau=0.05$ should not be interpreted as statistically significant. The estimated paired improvement is $0.00104$, with a $p$-value of $0.1829$. We additionally expand such analysis on California Housing datasets, and the conclusion remains. On California Housing, ConquerNet has a lower point estimate in 10 of the 15 comparisons. All 10 are significant at the 5% significance level.
>
> Furthermore, because the 30 pointwise comparisons are correlated, a comparison-by-comparison analysis may incur multiple testing concerns. We therefore performed a pooled global paired-bootstrap test, reported immediately below Table 2 of the revised manuscript, to assess whether ConquerNet provides an overall improvement. For BMI, the pooled absolute reduction is $0.00228$ (95% CI $[0.00106,\\,0.00350]$, $p$-value$=0.00029$). For California Housing, it is $0.004425$ (95% CI $[0.003419,\\,0.005413]$, $p$-value$=0.00002$). Thus, although the magnitude of the average improvement is modest, the pooled analysis provides strong evidence that ConquerNet brings meaningful improvement.
>
> We have expanded the real data analysis section and added more quantile-regression metrics to Appendix B.1 of the revised manuscript, highlighted in blue. Thank you again for your meaningful suggestions.

---

> ### Author Response · Authors · 2026-07-19
>
> ### R3.5. Minimax rate versus optimization
>
> We sincerely thank you for this important comment. The role of Theorem 3.1 is to show that our convolution smoothing, although introduced for optimization purposes, does not move the estimator away from the minimax-optimal statistical endpoint.
>
> The theoretical gain of ConquerNet should not be understood as a faster minimax statistical rate. Rather, it lies in the finite-step optimization path toward the same minimax endpoint. The nonsmooth pinball loss used in Padilla et al. (2022) leads to a nonsmooth empirical objective, whereas our convolution-smoothed loss yields a smoother optimization landscape while controlling the smoothing bias. Hence, the relevant theoretical question is whether this smoothing improves the optimization budget without sacrificing minimax optimality.
>
> To clarify this point, we added an output-layer finite-step optimization comparison. Let $\\eta_T(0)$ and $\\eta_T(h)$ denote the finite-step empirical optimization gaps of the last-layer objective after $T$ first-order iterations, respectively for the nonsmoothed pinball loss $\\rho_\\tau(\\cdot)$ and our convolution-smoothed loss $\\ell_h(\\cdot)$. We show that $$\\eta_T(0)=O(T^{-1/2}),\\qquad
> \\eta_T(h)=O\\left(\\min\\{T^{-1/2},h^{-1}T^{-1}\\}\\right).$$ Consequently, the $\\varepsilon$-accuracy iteration complexity is $O(\\varepsilon^{-2})$ for the nonsmoothed loss, while it becomes $O(\\min\\{\\varepsilon^{-2},h^{-1}\\varepsilon^{-1}\\})$ for the smoothed loss. Therefore, the advantage of convolution smoothing lies in reducing the finite-step optimization budget needed to approach the same minimax-optimal statistical endpoint. Lastly, this output-layer comparison provides a tractable explanation of the optimization mechanism and is aligned with common output-layer or linearized analyses in neural-network methodology and theory (Carratino et al., 2018; Jacot et al., 2018; Wei & Khardon, 2024). A full global iteration-complexity theory for jointly training all parameters of a DNN remains challenging (Boob et al., 2022) and is left for future work.
>
> We have revised the manuscript after Theorem 3.1 to include this finite-step output-layer optimization analysis and added the corresponding proposition and proof to the Appendix. The revisions are highlighted in blue.
>
> **References**
>
> - Boob, Dey & Lan (2022). *Complexity of training ReLU neural network.*
> - Carratino, Rudi & Rosasco (2018). *Learning with SGD and random features.*
> - Jacot, Gabriel & Hongler (2018). *Neural tangent kernel: Convergence and generalization in neural networks.*
> - Wei & Khardon (2024). *Variational inference on the final-layer output of neural networks.*

---

> ### Author Response · Authors · 2026-07-19
>
> ### R3.6. Claim revision
>
> Thank you for raising this point. We agree that the original abstract sentence could be read as claiming uniform tail superiority at every sample size, which was not our intended claim. The relevant pattern in Tables 1 and 5 of the revised manuscript is a strong sample-size dependence. As already noted in the main text, $n=1000$ is smaller than the number of network parameters and is inadequate for fully training these models; the examples you highlighted accurately describe this challenging small-sample regime. However, pooling the three kernels, three scenarios, and two architectures, the number of high and low quantiles in which ConquerNet has a lower MSE than the nonsmoothed baseline increases sharply from 7/36 at $n=1000$ to 22/36 at $n=5000$ and 31/36 at $n=10000$. At $n=10000$, the separate counts are 14/18 for $\\tau=0.05$ and 17/18 for $\\tau=0.95$. This transition from 7/36 at small $n$ to $31/36$ at large $n$ shows the pronounced evidence. This trend remains after accounting for Monte Carlo uncertainty. The number of reductions at tails exceeding the combined standard error $\\mathrm{SE}_{\\Delta}$ rises from $2/36$ at $n=1000$ and $5/36$ at $n=5000$ to $23/36$ at $n=10000$ (Table 6 of the revised manuscript). Therefore, the evidence does not support uniform gains at small $n$, but it does support an increasingly consistent advantage as the sample size grows. This is also the interpretation given in the main-text discussion of Table 1.
>
> Regarding the fragility of Assumption 2, we claim that our contribution is pointwise in a fixed $\\tau\\in(0,1)$; it is not an extreme-quantile asymptotic analysis in which $\\tau\\to0$ or $1$. Accordingly, $\\tau=0.05$ and $0.95$ are fixed interior quantile levels in our framework. For any fixed $\\tau$, the local lower-bound condition in Assumption 2 naturally holds with mild conditions such as continuity, and the smoothing bias remains $O(h^2)$. A high or low quantile may have a smaller density and hence a larger constant hidden in the $O(h^2)$ smoothing-bias term, but under Assumption 2 this lower bound remains a fixed positive constant rather than deteriorating with $n$. Therefore, it does not affect the large-sample asymptotic theory established in this paper. The improvement in our high- and low-quantile results as $n$ increases from $1000$ to $10000$ also supports this point.
>
> To make this qualifier explicit, we revised the abstract sentence to:
>
> "In numerical studies, the proposed approach improves performance across a broad range of settings, with gains becoming more consistent as sample size grows, especially at the low and high quantiles."
>
> We have also added a limitation illustration in the discussion section of the revised manuscript (in blue), as detailed in R1.7. It states that the empirical gains are not uniform across all finite-sample settings, but they become increasingly consistent as sample size grows, especially at the fixed low and high quantiles.

---

### Review · Reviewer_QpS3 · 2026-07-03

**Summary Of Contributions:**

The paper proposes ConquerNet, which combines convolution-type smoothing of the pinball loss (from Fernandes et al. 2021, developed for linear quantile regression) with ReLU quantile networks (from Padilla et al. 2022). The kink in the quantile loss makes deep optimisation unstable and hurts generalisation, and the smoothing rounds it to a differentiable objective while still targeting the true quantile. For theory, the paper proves a minimax rate over Besov spaces and a general upper bound without structural assumptions. Experiments compare against the Padilla et al. (2022) baseline across simulated scenarios, BMI datast, and loss landscape plots. Part of the code and results are also anonymously provided.

**Audience:**

Yes

**Audience Explanation:**

Yes. The intersection of smoothed quantile regression and deep networks is of interest to part of the TMLR audience, and the finding that convolution smoothing can ease optimisation without sacrificing minimax optimality is a useful.

**Claims And Evidence:**

Yes

**Claims Explanation:**

The theoretical contribution is sound and the paper is clearly written. However, I want to better understand the nature of the contributions, as detailed below:


Theorem 3.1: I read the theorem as stating a "no-harm" result, that is one does not lose anything if one uses the convolution based smoothing loss, or to say it differently ConquerNet attains almost the same rate as that of Padilla et al (2022): smoothing does not improve the rate, and the assumption on $h^{2}$ is meant to show that the smoothing bias is controlled so the optimality is retained. This is fine and reasonable, but then I do not fully follow what is new or interesting in comparison to Padilla et al. (2022).

Experiments: Following from above, I can read the contribution as one does not lose anything in terms of rate if smoothing is done (if the smoothing kernel is chosen appropriately), however the optimisation process (which is crucially important) becomes easier when it comes to neural networks. But then I'd want to better understand the experimental gains. The main MSE table, however, reports no standard errors, so the many small margins at central quantiles are hard to judge. The most convincing evidence is the consistent improvement at high/low quantiles. Furthermore, I don't fully get the early stopping argument. My reading says that the results might be confounded by the stopping strategy. As noted in Appendix, the faster training comes from early stopping rule, while the baseline runs full epochs, so I'm wondering if this is a clean comparison of the smoothing itself.

As a summary: I do see the point of the contribution, but some requested changes below are needed for me to better understand the contribution.

**Requested Changes:**

Some changes:
1. Help me better understand the theoretical contribution as compared against Padilla et al. (2022)
2. Please provide standard errors for Table 1 to better gauge the gains.
3. Help me understand the stopping strategy.

---

> ### Author Response · Authors · 2026-07-19
>
> We sincerely thank you for your time and insightful comments, which have substantially improved our manuscript. We have completed a comprehensive revision and provide point-by-point responses below. All manuscript changes discussed in the responses are highlighted in blue in the revised manuscript.
>
> ### R2.1. Contribution beyond Padilla et al. (2022)
>
> We sincerely thank you for this insightful comment. We agree that Theorem 3.1 is best viewed as a statistical **no-harm** result relative to Padilla et al. (2022). Without sacrificing statistical minimax optimality, the **gain** of ConquerNet is to improve the computational optimization path. Under exact empirical optimization, Theorem 3.1 shows that convolution smoothing does not deteriorate the minimax rate, and both the nonsmoothed quantile ReLU network and ConquerNet can attain the minimax-optimal statistical rate. The key difference lies in the finite-step optimization path toward this minimax endpoint. The nonsmooth pinball loss used in Padilla et al. (2022) generally requires a larger optimization budget to approach such exact empirical optimization, whereas ConquerNet smooths the objective and can reach the same statistical minimax optimality with fewer optimization steps.
>
> Although previous studies have provided substantial empirical evidence that smooth losses with non-sparse gradients are important for effective deep neural network training (Berrada et al., 2018), providing a global iteration-complexity theory for the entire DNN training problem is generally difficult (Boob et al., 2022). Therefore, to provide a tractable explanation of where the optimization gain comes from, we focus on the output-layer subproblem and compare the finite-step empirical optimization gap between ConquerNet and Padilla et al. (2022). Such output-layer analysis is aligned with widely used perspectives in neural-network methodology and theory literature (Carratino et al., 2018; Jacot et al., 2018; Wei & Khardon, 2024).
>
> Let $\\eta_T(0)$ and $\\eta_T(h)$ denote the finite-step empirical optimization gaps of the last-layer objective after $T$ first-order iterations, respectively for the check loss $\\rho_\\tau(\\cdot)$ used in Padilla et al. (2022) and our convolution-smoothed loss $\\ell_h(\\cdot)$. We show that $$\\eta_T(0)=O(T^{-1/2}),\\qquad \\eta_T(h)=O(\\min\\{T^{-1/2},h^{-1}T^{-1}\\}).$$ Therefore the $\\varepsilon$-accuracy iteration complexity is $O(\\varepsilon^{-2})$ for the nonsmoothed loss $\\rho_\\tau(\\cdot)$, while it becomes $O(\\min\\{\\varepsilon^{-1}h^{-1},\\varepsilon^{-2}\\})$ for our smoothed loss $\\ell_h(\\cdot)$. Together with Theorem 3.1, this clarifies the cost-benefit profile of ConquerNet. If $h$ is too large, for example $h^2\\gg n^{-s/(2s+d)}$, the optimization problem becomes easier, but the smoothing bias may start to sacrifice statistical efficiency. With an appropriate $h$ choice, ConquerNet preserves the no-harm minimax guarantee while reducing the finite-step optimization budget. We have revised the manuscript after Theorem 3.1 to include this finite-step output-layer optimization analysis and added the corresponding proposition and proof to the Appendix. The revisions are highlighted in blue.
>
> **References**
>
> - Berrada, Zisserman & Kumar (2018). *Smooth loss functions for deep top-k classification.*
> - Boob, Dey & Lan (2022). *Complexity of training ReLU neural network.*
> - Carratino, Rudi & Rosasco (2018). *Learning with SGD and random features.*
> - Jacot, Gabriel & Hongler (2018). *Neural tangent kernel: Convergence and generalization in neural networks.*
> - Wei & Khardon (2024). *Variational inference on the final-layer output of neural networks.*

---

> ### Author Response · Authors · 2026-07-19
>
> ### R2.2. Standard errors for Table 1
>
> Thank you for this helpful suggestion. We have added standard errors in parentheses in Table 5. We agree that small margins at some central quantiles should be interpreted cautiously. To make the comparison more informative, we now assess whether the MSE reduction is large relative to Monte Carlo uncertainty, using the combined standard error $\\mathrm{SE}\_{\\Delta}=\\{\\mathrm{SE}\_{\\mathrm{base}}^2+\\mathrm{SE}\_{\\mathrm{ConquerNet}}^2\\}^{1/2}$ for the difference in two reported means. With this uncertainty information included, the main conclusion remains unchanged.
>
> Using the complete MSE and uncertainty results in Tables 5--6 of the revised manuscript, we compare each of the three pre-specified kernels directly with the same nonsmoothed baseline. Pooled over all kernel-setting comparisons, ConquerNet attains a lower mean MSE in 175 out of 270 comparisons; among these, 109 reductions exceed the combined standard error $\\mathrm{SE}\_{\\Delta}$ and 47 exceed $1.96\\times\\mathrm{SE}\_{\\Delta}$. The pattern is consistent across kernels: Gaussian, Uniform, and Epanechnikov improve 56/90, 59/90, and 60/90 comparisons, respectively. The evidence strengthens with sample size: pooling the three kernels, the number of favorable comparisons increases from $33/90$ at $n=1000$ to $65/90$ at $n=5000$ and $77/90$ at $n=10000$. The transition is especially sharp at the fixed low and high quantiles, where the corresponding counts increase from $7/36$ to $22/36$ and $31/36$. Thus, the tail evidence lies in the change from mostly unfavorable small-sample results to highly consistent large-sample gains, rather than in a higher final proportion than the overall $77/90$. Moreover, the number of reductions at tails exceeding $\\mathrm{SE}\_{\\Delta}$ increases from $2/36$ to $5/36$ and $23/36$, so the same pattern remains after accounting for Monte Carlo uncertainty. With standard errors reported, the empirical gains of ConquerNet can be assessed more clearly, even where the raw MSE margins are small. We have revised the discussion following Table 1 and provided the standard errors with the detailed uncertainty analysis in Appendix B.1. These revisions are highlighted in blue.
>
> ---
>
> ### R2.3. Matched stopping rule
>
> Thank you for raising this important concern. For the baseline model, we directly followed the training protocol in the public code of Padilla et al. (2022), under which the baseline models are trained for the prescribed full epochs. We therefore honestly noted in Appendix B that the original wall-clock comparison confounds two effects: loss smoothing and the different stopping protocol. We agree that the original timing comparison (Figures 2 and 5--8) should not be used as isolated evidence that the speedup is solely caused by smoothing. Rather, they should be interpreted together with the accuracy results in Table 1: under the reproduced baseline training protocol, the baseline used a larger computation budget but still did not outperform ConquerNet.
>
> In the revision, to isolate the stopping-rule effect, we have added a controlled timing experiment (Table 4 of the revised manuscript), where the same stopping criterion is applied to both the nonsmoothed baseline and ConquerNet. Under this matched stopping protocol, ConquerNet remains faster: across all scenarios, models, sample sizes, quantile levels, and kernels, ConquerNet is faster in 240 out of 270 kernel-setting comparisons. To account for trial-to-trial variation, we compute the combined Monte Carlo standard error of the reported mean difference as $\\mathrm{SE}\_{\\Delta}=\\{\\mathrm{SE}\_{\\mathrm{base}}^2+\\mathrm{SE}\_{\\mathrm{ConquerNet}}^2\\}^{1/2}$, where the standard errors are computed over 50 trials. Under this criterion, 164/240 reductions exceed $\\mathrm{SE}\_{\\Delta}$ and 77/240 exceed $1.96\\times \\mathrm{SE}\_{\\Delta}$ (indicating significantly strong improvement). Together with the same-stopping-rule MSE comparison in Appendix B, Table 3, the controlled timing comparison in Table 4 supports that the improvement comes from ConquerNet rather than from the stopping strategy alone. We have revised Appendix B.1 to clarify the stopping protocols and the interpretation of the original timing figures, and added the matched-stopping timing comparison in Table 4. These revisions are highlighted in blue. Thank you again for making our paper more rigorous and transparent.

---

### Review · Reviewer_q53P · 2026-07-06

**Summary Of Contributions:**

The paper proposes ConquerNet, a quantile regression method that trains ReLU neural networks using a convolution-smoothed version of the pinball loss. The authors derive nonasymptotic risk bounds and show that the estimator achieves near-minimax rates over Besov-type function classes. The empirical evaluation is mainly based on controlled synthetic simulations, with one narrow real-data BMI experiment, and reports improved estimation accuracy and training efficiency over a standard quantile ReLU-network baseline in some settings, particularly for extreme quantiles.

**Audience:**

Yes

**Audience Explanation:**

Yes, this paper tries to build on existing literature and I would say some members of the TMLR audience would be interested in seeing such a paper.

However, it is not clear to me if this paper's contribution has any impact in real world applications as the paper does not sufficiently establish why the non-smoothness of the pinball loss at zero is a major problem that needs to be addressed, nor its empirical results does not convince me that it is a useful approach for practice.

**Broader Impact Concerns:**

No major direct broader-impact concerns. The work is mainly theoretical/statistical. A minor concern is that quantile-regression methods may be used in high-stakes domains, so the authors should avoid overstating reliability without stronger calibration and real-data evidence.

**Claims And Evidence:**

No

**Claims Explanation:**

The main weakness is that the paper’s demonstrated contribution is much narrower than its stated motivation and claims. Relative to Padilla et al., the paper mainly replaces the standard pinball loss in quantile ReLU-network regression with a convolution-smoothed surrogate and shows that comparable near-minimax rates can still be obtained. This is a valid but incremental theoretical extension, and the theory does not establish a better statistical rate, a clear finite-sample advantage, or improved optimization behavior for neural-network training. Empirically, the evidence is also limited: the experiments are mostly controlled synthetic simulations, the only real-data study is a single BMI tabular dataset with small reported gains, and the baseline set is largely restricted to the standard quantile ReLU-network estimator. As a result, the paper does not convincingly support its broader claims about practical accuracy, training efficiency, or the need for this particular smoothing method in my view. Below I explain my view in more detail.



---------

A useful way to understand this paper is as an extension of the quantile ReLU-network framework studied by Padilla, Tansey, and Chen. That prior work established minimax-rate guarantees for quantile regression with ReLU networks trained using the standard pinball loss. The present submission modifies this framework by replacing the pinball loss with a convolution-smoothed version, and then proves that the resulting estimator can retain comparable near-minimax rates over Besov-type function classes. This is a legitimate and technically meaningful direction. However, the contribution appears incremental: the paper does not introduce a substantially new estimator class, improve the minimax rate over prior quantile ReLU-network theory, or establish a fundamentally different statistical principle. The main contribution is instead to show that a particular smoothing scheme can be incorporated into the existing framework without losing the known statistical rate.

This incremental nature is not itself a problem. A paper can make a valuable contribution by carefully closing a technical gap. However, the submission makes broader claims about optimization, accuracy, and practical usefulness than are supported by the evidence provided. The paper would be stronger if it more clearly separated the contribution that is actually established from the broader claims that remain speculative or insufficiently demonstrated.

The most important gap is between the statistical theory and the optimization motivation. The paper motivates the method by emphasizing the nonsmoothness of the pinball loss and suggesting that smoothing improves the training objective. However, the main theoretical results are statistical risk bounds, not optimization guarantees. They do not show faster convergence of gradient-based methods, better behavior of SGD, improved conditioning, or a generally easier neural-network loss landscape. Moreover, smoothing the scalar residual loss does not remove the nonconvexity of the ReLU-network training problem, and the network parameterization can still create difficult optimization behavior. Therefore, the theory supports the claim that the smoothed estimator can be statistically well behaved under the stated assumptions, but it does not establish the stronger claim that the method resolves an important optimization problem in deep quantile regression.

A related issue is that the practical need for the proposed smoothing is not sufficiently demonstrated. The paper emphasizes the nondifferentiability of the pinball loss, but it does not convincingly show that this nondifferentiability is a major practical obstacle for modern neural-network training. In continuous regression settings, the residual is exactly zero with probability zero, and standard subgradient or autodiff-based methods can train pinball-loss objectives. The more relevant question is whether the piecewise-linear structure of the pinball loss leads to instability, poor tail behavior, or worse finite-sample performance in realistic settings. The paper gestures toward this motivation, but the evidence is not strong enough to establish it.

The theoretical results also do not show that the proposed method is statistically superior to the original quantile ReLU-network estimator. The minimax theorem demonstrates that the smoothed estimator can achieve comparable rates under suitable assumptions. This is useful, but it is mostly a preservation result: smoothing does not destroy the known near-minimax rate. The paper does not show a better rate, a meaningful improvement in constants, a dominance result, or a general statistical advantage over the standard pinball-loss estimator. Since the method introduces an additional bandwidth parameter, the paper should more clearly explain the resulting bias-variance tradeoff and why this extra tuning burden is worthwhile.

The empirical evidence is also too narrow to support the broader practical claims. The main experiments are controlled synthetic simulations with low-dimensional, manually specified data-generating processes. These experiments are useful for illustrating the theory, since the true conditional quantile function is known. However, they are not sufficient to establish practical usefulness in modern quantile regression or uncertainty estimation. The only real-data experiment is a single BMI tabular dataset with a limited set of covariates. This does not provide enough evidence that the method generalizes across realistic data modalities, noise structures, sample sizes, or application domains.

The baseline comparison is a major limitation. The paper primarily compares against the standard quantile ReLU-network baseline from Padilla et al. This is a natural baseline, but it is not enough to support claims of practical effectiveness. Since the paper proposes a smoothed quantile loss, the most relevant comparisons would include other smoothed or Huberized quantile losses, tilted-Huber variants, simultaneous quantile regression methods, noncrossing quantile networks, gradient-boosted quantile regression, quantile forests, distributional-regression approaches, and conformalized quantile methods where appropriate. Without such baselines, the experiments only show that the proposed smoothing can improve over one particular pinball-loss neural-network baseline in some settings.

The reported empirical results also appear more mixed than the paper’s general claims suggest. The proposed method performs well in several simulation settings, especially for larger sample sizes and extreme quantiles, but the baseline is better in some configurations. Performance depends on sample size, quantile level, architecture, kernel, and bandwidth. This makes it difficult to conclude that the method provides a robust or general improvement. The paper would need a clearer pre-specified kernel and bandwidth-selection protocol, applied consistently across tasks, and compared against equally tuned baselines.

The real-data evidence is especially limited. The BMI experiment reports small differences in pinball loss, but the paper does not provide sufficient analysis to establish practical significance. There are no confidence intervals or significance tests for the main real-data comparison, and the paper does not evaluate whether the small loss improvements translate into better uncertainty estimates. Important properties such as calibration, prediction-interval coverage, interval width, interval score, quantile crossing, and tail reliability are not adequately studied. These metrics are directly relevant to quantile regression and would be more informative than average pinball loss alone.

The evidence for improved training efficiency is also not fully convincing. A fair computational comparison should use the same stopping rule, validation protocol, optimizer, tuning budget, and implementation standard for all methods. If the reported speedup is affected by differences in early stopping or training protocol, then it cannot be cleanly attributed to the convolution-smoothed loss itself. Since computational efficiency is one of the claimed practical advantages, this comparison needs to be isolated more carefully.

Finally, the loss-landscape visualization is only suggestive. A two-dimensional projection of a high-dimensional neural-network objective in one synthetic setting cannot establish that the method generally has a better optimization landscape. It may be useful as an illustration, but it does not replace systematic evidence such as training curves across many random seeds, sensitivity to initialization, optimizer comparisons, gradient-norm behavior, convergence diagnostics, and experiments across multiple architectures and datasets.

Overall, the paper supports a narrower conclusion than the submission appears to claim. It provides a technically plausible extension of Padilla-style quantile ReLU-network theory to a convolution-smoothed pinball loss, and it gives evidence that this smoothing can help in some controlled settings. However, the paper does not yet convincingly establish that the method is a practically important improvement over existing quantile-regression approaches, that it solves a significant optimization problem, or that it should change what practitioners use. The gap between the claimed motivation and the demonstrated contribution remains substantial.

**Requested Changes:**

1. **Revise claims about superiority and practical usefulness. Critical.** The authors should substantially temper claims that the proposed method improves accuracy, training efficiency, or practical quantile regression performance unless they provide clearer evidence. The current results suggest possible gains in some settings, but do not establish broad superiority.

2. **Justify why nonsmoothness of the pinball loss is a serious problem. Critical.** The paper should provide stronger theoretical or empirical evidence that the nonsmoothness of the pinball loss materially harms neural-network quantile regression in practice. At present, this central motivation is asserted more than demonstrated.

3. **Isolate the effect of the proposed smoothing. Critical.** The authors should show that the observed improvements are caused by the convolution-smoothed loss itself, rather than by choices of bandwidth, kernel, stopping rule, tuning protocol, or implementation details.

4. **Strengthen empirical comparisons.** The paper should compare against relevant alternatives, especially other smoothed or Huberized quantile losses and stronger quantile-regression baselines. Comparing mainly to the standard pinball-loss ReLU network is not enough to establish the usefulness of this particular smoothing method.

5. **Expand and improve real-data evaluation. Critical.** The authors should evaluate on multiple real datasets and include metrics directly relevant to quantile regression.

6. **Clarify the practical role of bandwidth and kernel choice.** The method introduces additional choices that may affect performance. The authors should provide a robust selection procedure and sensitivity analysis.

7. **Add a clearer limitations discussion.** The paper should explicitly state that the current evidence supports a narrower conclusion: convolution smoothing can preserve statistical rates and may help in some settings, but its general practical advantage over existing quantile-regression methods remains unproven in my view.

---

> ### Author Response · Authors · 2026-07-19
>
> We sincerely thank you for your time and insightful comments, which have substantially improved our manuscript. We have completed a comprehensive revision and provide point-by-point responses below. All manuscript changes discussed in the responses are highlighted in blue in the revised manuscript.
>
> ### R1.1. Claims and practical scope
>
> Thank you for this important comment. We agree that our original wording could be read as claiming uniform superiority. Our theoretical comparison identifies a more specific regime of usefulness. Under exact empirical optimization, ConquerNet and the nonsmoothed quantile network attain the same near-minimax statistical rate; hence, our advantage is not a universally better statistical endpoint. Rather, smoothing reduces the finite-step optimization gap on the path toward that endpoint (detailed in R1.2/R2.1/R3.5). Consequently, ConquerNet is most relevant when the model is sufficiently complex, the sample size is large enough, and optimization becomes a material bottleneck. In small-sample or simpler estimation problems, the optimization issues may not occur, and we do not claim that ConquerNet must outperform simpler alternatives.
>
> The evidence in our response is consistent with this scoped interpretation. As detailed in R2.2 and Tables 5--6 of the revised manuscript, the number of comparisons in which ConquerNet achieves a lower MSE increases from $33/90$ at $n=1000$ to $65/90$ at $n=5000$ and $77/90$ at $n=10000$. R3.6 and Tables 5--6 of the revised manuscript show an especially sharp transition at the low and high quantiles: the corresponding counts increase from $7/36$ to $22/36$ and $31/36$. For computation, R2.3 and Table 4 of the revised manuscript remove the stopping-rule confound: under the same stopping criterion, ConquerNet is faster in $240/270$ matched comparisons. These results support a regime-dependent optimization benefit, rather than broad dominance in every setting.
>
> Accordingly, we have revised the manuscript in blue to make this scope explicit. The revision is also detailed in R1.7/R3.6. We replaced the abstract's categorical "across the board" claim with "In numerical studies, the proposed approach improves performance across a broad range of settings, with gains becoming more consistent as sample size grows, especially at the low and high quantiles." In the discussion section, we added a limitations paragraph (in blue) stating that ConquerNet does not imply universal superiority over simple models or all existing quantile-regression methods.

---

> ### Author Response · Authors · 2026-07-19
>
> ### R1.2. Why smoothing matters
>
> Thank you for identifying this gap. Our motivation in the introduction builds on prior DNN studies showing empirically that smooth losses with non-sparse gradients can improve neural-network training (Berrada et al., 2018), while sharp minima and nonsmooth loss surfaces can reduce optimization stability and generalization (Huang et al., 2020; Foret et al., 2021; Keskar et al., 2016). The pinball loss exhibits precisely this structure: it is piecewise linear, and its derivative jumps from $\\tau-1$ to $\\tau$ when a residual changes sign. Thus, the concern is not whether autodifferentiation can assign a subgradient at an isolated zero residual; it is that the empirical objective lacks a Lipschitz-continuous gradient along the optimization path. Convolution smoothing preserves convexity in the residual while replacing this jump by a Lipschitz-continuous gradient.
>
> We agree, however, that this DNN motivation was previously supported mainly by empirical evidence. Our statistical theory first establishes a no-harm result relative to the nonsmoothed quantile network of Padilla et al. (2022). Under exact empirical optimization, convolution smoothing preserves the near-minimax rate. Its additional benefit concerns the finite-step optimization path. A global iteration-complexity theory for jointly optimizing all layers of a nonconvex DNN remains difficult (Boob et al., 2022). We therefore add a tractable output-layer analysis, as detailed in R2.1 and R3.5, to demonstrate this optimization distinction theoretically.
>
> Conditional on the hidden-layer features, let $\\eta_T(0)$ and $\\eta_T(h)$ be the empirical optimization gaps after $T$ projected first-order iterations for the nonsmoothed and smoothed output-layer objectives. We add a new proposition to show $$\\eta_T(0)=O(T^{-1/2}),\\qquad
> \\eta_T(h)=O\\!\\left(\\min\\{T^{-1/2},h^{-1}T^{-1}\\}\\right).$$ Equivalently, the sufficient iteration budget for an $\\varepsilon$-accurate output-layer solution changes from $O(\\varepsilon^{-2})$ to $O(\\min\\{\\varepsilon^{-2},h^{-1}\\varepsilon^{-1}\\})$. This order-level difference directly demonstrates the finite-step optimization gain created by smoothing in a concrete neural-network subproblem, rather than merely asserting that a smooth loss should be easier to optimize.
>
> Together with Theorem 3.1, this shows that ConquerNet can reduce the finite-step optimization budget along this output-layer path while preserving the same near-minimax statistical endpoint. The controlled experiments summarized in R1.3 and the matched-stopping comparison in R2.3 provide complementary empirical evidence. We have revised the manuscript after Theorem 3.1 to include this finite-step output-layer optimization analysis and added the corresponding proposition and proof to the Appendix. The revisions are highlighted in blue.
>
> **References**
>
> - Berrada, Zisserman & Kumar (2018). *Smooth loss functions for deep top-k classification.*
> - Boob, Dey & Lan (2022). *Complexity of training ReLU neural network.*
> - Foret et al. (2021). *Sharpness-aware minimization for efficiently improving generalization.*
> - Huang et al. (2020). *Understanding generalization through visualizations.*
> - Keskar et al. (2016). *On large-batch training for deep learning: Generalization gap and sharp minima.*

---

> ### Author Response · Authors · 2026-07-19
>
> ### R1.3. Isolating the smoothing effect
>
> Thank you for this important request. Table 1 is intentionally a comprehensive performance comparison rather than a one-factor ablation: it summarizes the overall smoothing effect across scenarios, architectures, sample sizes, quantile levels, and kernels. Tables 5 and 6 of the revised manuscript make that evidence transparent by reporting the raw results, their variability, and the paired reductions, while the expanded-baseline Table 18 examines whether the comparison is specific to the nonsmoothed neural-network baseline. These tables establish the overall empirical effect, but they do not by themselves isolate smoothing from every auxiliary choice.
>
> We therefore organize the additional evidence by the factor being examined. Kernel choice is addressed throughout the experiments: the same comparisons are repeated with Gaussian, Uniform, and Epanechnikov kernels under otherwise matched settings, so the evidence does not rest on a single favorable kernel. For bandwidth, Table 8 uses a data-driven cross-validation choice, whereas Table 7 examines a range of bandwidths under a fixed experimental protocol; together, they distinguish the smoothing effect from choices of bandwidth. For the stopping rule, Tables 3 and 4 compare accuracy and training speed under matched stopping criteria. Tables 12, 13, and 15 examine an alternative training protocol in which multiple quantiles are fitted jointly under a noncrossing constraint, including both simulation and real-data settings. Architecture and implementation dependence is examined more broadly by Table 2; Tables 9--11, which isolate the effect of residual blocks; and Tables 12, 13, and 15, which change the single- versus multi-quantile implementation and the experimental domain. Taken together, these targeted comparisons separate the observed smoothing effect from kernel, bandwidth, stopping, training protocol, architecture, and implementation. We provide the following experiment summary to distinguish the comprehensive comparison in Table 1 from these effect-isolation experiments.
>
> | Effect examined | Revised-manuscript tables |
> |---|---|
> | Overall smoothing effect and uncertainty | Tables 1, 5, and 6 |
> | Kernel choice | Across all experiments |
> | Bandwidth selection and sensitivity | Tables 7--8 |
> | Stopping rule | Tables 3 and 4 |
> | Tuning protocol | Tables 12, 13, and 15 |
> | Network architecture and implementation | Table 2; Tables 9--13 and 15 |
>
> We have revised the discussion following Table 1 to clarify its role as a comprehensive comparison and added a summary table mapping each auxiliary factor to the corresponding supplementary experiments in Appendix B.1. These revisions are highlighted in blue.
>
> ---
>
> ### R1.4. Stronger baseline comparisons
>
> Thank you for this helpful suggestion. Besides the original baseline (Nonsmooth), we have added two complementary quantile-regression baselines to the simulations in Scenarios S1--S3: quantile regression forest (QRF), a tree-based nonparametric estimator, and a Huberized quantile-loss neural network (Huber). The complete MSE results, including all comparison counts reported below, are in Table 18 of the revised manuscript.
>
> The original conclusion remains unchanged after adding these two baselines: the gains are not uniform in every finite-sample configuration, but become substantially more consistent as the sample size grows. Across all 90 settings in S1--S3, Gaussian, Uniform, and Epanechnikov ConquerNet have lower MSE in $56/90$, $59/90$, and $60/90$ comparisons with Nonsmooth; $83/90$, $85/90$, and $84/90$ comparisons with QRF; and $61/90$, $61/90$, and $64/90$ comparisons with Huber, respectively. Thus, the original Table 1 comparison with Nonsmooth is unchanged, while the additional results show that the conclusion is not specific to that baseline.
>
> More importantly for our sample-size claim, at $n=10000$ the three kernels win $23/30$, $27/30$, and $27/30$ comparisons with Nonsmooth; $30/30$, $30/30$, and $30/30$ comparisons with QRF; and $27/30$, $27/30$, and $28/30$ comparisons with Huber. These counts increase from $13/30$, $10/30$, and $10/30$ against Nonsmooth, $23/30$, $25/30$, and $24/30$ against QRF, and $10/30$, $13/30$, and $14/30$ against Huber at small sample $n=1000$. For the original matched comparison with Nonsmooth, the pooled counts at the fixed low and high quantiles increase particularly sharply, from $7/36$ at $n=1000$ to $22/36$ at $n=5000$ and $31/36$ at $n=10000$. Hence, adding the tree-based and alternative smooth-loss baselines preserves the original conclusion that the gains become increasingly consistent as the sample size grows. We have added these additional baseline-comparison results to Appendix B.1 of the revised manuscript, highlighted in blue. Thank you again for helping us strengthen the paper.

---

> ### Author Response · Authors · 2026-07-19
>
> ### R1.5. Expanded real-data evaluation
>
> Thank you for emphasizing this important limitation. We have expanded the real-data evaluation in two complementary directions. First, we broadened the study beyond BMI by adding the California Housing dataset under the same single-quantile training protocol at $\\tau\\in\\{0.05,0.25,0.5,0.75,0.95\\}$ with additional paired-bootstrap inference for the pinball-loss comparisons on both datasets. Second, we added a suite of quantile-regression diagnostics that evaluate calibration, coverage, sharpness, quantile coherence, and tail reliability.
>
> Table 2 of the revised manuscript combines the BMI and California Housing results. For every comparison, we additionally report 95% paired-bootstrap confidence intervals for the reduction in pinball loss achieved by ConquerNet relative to the baseline. Thus, an interval entirely above zero indicates a significantly lower pinball loss for ConquerNet in that comparison. On BMI, ConquerNet has a lower point estimate in 23 of the 30 gender--quantile--kernel comparisons. At the 5% significance level, eight comparisons favor ConquerNet and two favor the baseline. On California Housing, ConquerNet has a lower point estimate in 10 of the 15 comparisons. All 10 are significant at the 5% significance level. Considering the comparison-by-comparison results may raise multiple testing concerns, we additionally report a pooled estimand for each dataset immediately below Table 2 of the revised manuscript: the 15 quantile--kernel loss differences are first averaged within each test observation, and the observations are then bootstrapped. For BMI, the pooled absolute reduction is $0.00228$ (95% CI $[0.00106,\\,0.00350]$, $p$-value$=0.00029$). For California Housing, it is $0.004425$ (95% CI $[0.003419,\\,0.005413]$, $p$-value$=0.00002$). These analyses support a statistically reliable average improvement on both datasets.
>
> Pinball loss provides a directly applicable evaluation criterion: its population expectation is minimized at the target conditional quantile. Therefore, it becomes a canonical proper scoring rule when the true conditional quantile is unobserved in real-data applications. Nevertheless, we agree that pinball loss alone does not reveal the coverage--sharpness trade-off of prediction intervals, coherence across separately fitted quantiles, or the severity of tail misses. We therefore added quantile-regression diagnostics in Table 16 of the revised manuscript with weighted interval score (WIS), 50% interval score, mean absolute quantile calibration error, mean absolute interval-coverage error, quantile crossing, and conditional tail-miss distance. WIS combines the median error with the 50% and 90% interval scores, and the interval score jointly penalizes excessive width and undercoverage. The mean absolute quantile calibration error averages $|\\widehat{P}(Y\\leq\\widehat q_\\tau)-\\tau|$ over the five fitted quantiles. The mean absolute interval-coverage error averages the deviations from the nominal 50% and 90% coverage levels. Finally, the mean conditional tail-miss distance averages the lower- and upper-tail distances from the corresponding interval endpoint among observations falling outside the 50% or 90% interval, thereby measuring the severity rather than only the frequency of tail misses.
>
> The results in Table 16 of the revised manuscript show that, on BMI, all three kernels reduce WIS, the 50% interval score, the mean interval-coverage error, and both tail-miss distances; calibration is also improved by Gaussian and Epanechnikov, while only Uniform is worse on this metric. All BMI methods have zero crossings. On California Housing, all three kernels reduce WIS, the 50% interval score, mean calibration error, mean coverage error, 50% interval width, 50% tail-miss distance, and the raw crossing rate; the latter falls from 23.26% to 1.43--2.47%. Uniform and Epanechnikov also reduce the 90% tail-miss distance, whereas Gaussian is slightly higher than the baseline.
>
> We have expanded the real data analysis section and added more quantile-regression metrics to Appendix B.1 of the revised manuscript, highlighted in blue. Thank you again for your meaningful suggestions.

---

> ### Author Response · Authors · 2026-07-19
>
> ### R1.6. Bandwidth and kernel choice
>
> Thank you for this suggestion. The bandwidth $h$ controls the trade-off between smoothing and approximation bias. As explained after Theorem 3.1, a very small $h$ is a conservative no-harm choice: as $h\\to0$, the smoothed loss converges to the original pinball loss, so the smoothing bias vanishes, although the optimization benefit of smoothing also diminishes. In contrast, an excessively large $h$ introduces greater smoothing bias. Theorem 3.1 provides theoretical guidance through the condition $h^2\\lesssim n^{-s/(2s+d)}$, under which the smoothing bias remains negligible relative to the statistical error. The theoretical impact of kernel choice is secondary: any bounded, nonnegative, centered kernel satisfying the stated moment conditions yields the same convergence rate, with the particular kernel affecting only constants rather than the theoretical guarantee.
>
> For a data-driven selection, we use a 5-fold cross-validation procedure on the training data only. For each candidate kernel and each bandwidth in $\\{0.001,0.005,0.01,0.05,0.1\\}$, we evaluate the original pinball loss on the validation folds and select the combination with the smallest mean validation loss; neither the test set nor the true conditional quantile is used. Table 8 of the revised manuscript reports the results under this data-driven procedure. For sensitivity, Table 7 of the revised manuscript varies the bandwidth across the candidate grid, while Gaussian, Uniform, and Epanechnikov kernels are compared throughout the experiments. As summarized in R1.3, the observed pattern is not confined to one bandwidth or kernel. We have clarified this selection rule and summarized the sensitivity evidence in a new sensitivity analysis section in the revised Appendix B.1 of the manuscript.
>
> ---
>
> ### R1.7. Limitations
>
> Thank you for this suggestion. We have added the limitations discussion in the discussion section of the revised manuscript (in blue). For your convenience, we also paste it as follows.
>
> > ConquerNet preserves the near-minimax statistical rate of the nonsmoothed quantile network and improves the finite-step optimization guarantee for the output-layer subproblem, but it does not attain a faster minimax statistical rate or establish globally faster optimization for the joint training of all network layers. The empirical gains are not uniform across all finite-sample settings, but they become increasingly consistent as sample size grows, especially at the fixed low and high quantiles.The method is therefore primarily intended for sufficiently complex quantile networks with sufficiently large data that optimization constitutes a material bottleneck; the present results do not imply universal superiority over simpler models or all existing quantile-regression methods. Moreover, the theoretical guarantees are pointwise for a fixed $\\tau\\in(0,1)$ and do not cover extreme-quantile asymptotics with $\\tau\\to0$ or $1$. Establishing full-network optimization guarantees and extending the theory to drifting extreme quantiles remain important directions for future work.